# Calibrate and Boost Logical Expressiveness of GNN Over Multi-Relational and Temporal Graphs

**Dingmin Wang**[*]
University of Oxford
`dingmin.wang@cs.ox.ac.uk`

**Yeyuan Chen**[*]
University of Michigan
`yeyuanch@umich.edu`

## Abstract

As a powerful framework for graph representation learning, Graph Neural Networks (GNNs) have garnered significant attention in recent years. However, to the best of our knowledge, there has been no formal analysis of the logical expressiveness of GNNs as Boolean node classifiers over multi-relational graphs, where each edge carries a specific relation type. In this paper, we investigate $\mathcal{FOC}_2$, a fragment of first-order logic with two variables and counting quantifiers. On the negative side, we demonstrate that the $R^2$-GNN architecture, which extends the local message passing GNN by incorporating global readout, fails to capture $\mathcal{FOC}_2$ classifiers in the general case. Nevertheless, on the positive side, we establish that $R^2$-GNN models are equivalent to $\mathcal{FOC}_2$ classifiers under certain restricted yet reasonable scenarios. To address the limitations of $R^2$-GNN regarding expressiveness, we propose a simple *graph transformation* technique, akin to a preprocessing step, which can be executed in linear time. This transformation enables $R^2$-GNN to effectively capture any $\mathcal{FOC}_2$ classifiers when applied to the "transformed" input graph. Moreover, we extend our analysis of expressiveness and *graph transformation* to temporal graphs, exploring several temporal GNN architectures and providing an expressiveness hierarchy for them. To validate our findings, we implement $R^2$-GNN and the *graph transformation* technique and conduct empirical tests in node classification tasks against various well-known GNN architectures that support multi-relational or temporal graphs. Our experimental results consistently demonstrate that $R^2$-GNN with the graph transformation outperform the baseline methods on both synthetic and real-world datasets. The code is available at `https://github.com/hdmmblz/multi-graph`.

## 1 Introduction

Graph Neural Networks (GNNs) have become a standard paradigm for learning with graph structured data, such as knowledge graphs Park et al. [2019], Tena Cucala et al. [2021], Wang et al. [2023] and molecules Hao et al. [2020], Gasteiger et al. [2021], Guo et al. [2021]. GNNs take as input a graph where each node is labelled by a feature vector, and then they recursively update the feature vector of each node by processing a subset of the feature vectors from the previous layer. For example, many GNNs update a node's feature vector by combining its value in the previous layer with the output of some aggregation function applied to its *neighbours*' feature vectors in the previous layer; in this case, after $k$ iterations, a node's feature vector can capture structural information about the node's $k$-hop neighborhood. GNNs have proved to be very efficient in many applications like knowledge graph completion and recommender systems. Most previous work on GNNs mainly revolves around finding

---

[*]Equal contribution, listed in alphabetical order.

37th Conference on Neural Information Processing Systems (NeurIPS 2023).

GNN architectures (e.g. using different aggregation functions or graph-level pooling schemes) which offer good empirical performance Kipf and Welling [2016], Xu et al. [2018], Corso et al. [2020]. The theoretical properties of different architectures, however, are not yet well understood.

In Xu et al. [2018], the authors first proposed a theoretical framework to analyze the expressive power of GNNs by establishing a close connection between GNNs and the Weisfeiler-Lehman (1-WL) test for checking graph isomorphism. Similarly, Geerts and Reutter [2022] provides an elegant way to easily obtain bounds on the separation power of GNNs in terms of the Weisfeiler-Leman (k-WL) tests. However, the characterization in terms of the Weisfeiler-Lehman test only calibrates distinguishing ability. It cannot answer *which Boolean node classifier can be expressed by GNNs*. To this end, Barceló et al. [2020] consider a class of GNNs named ACR-GNNs proposed in Battaglia et al. [2018], where the update function uses a "global" aggregation of the features of all nodes in the graph in addition to the typical aggregation of feature vectors of neighbour nodes. Then, the authors of the paper prove that in the single-relational [2] scenario, ACR-GNNs can capture every Boolean node classifier expressible in the logic $\mathcal{FOC}_2$.

However, most knowledge graphs need multiple relation types. For example, in a family tree, there are multiple different relation types such as "father" and "spouse". In this paper, we consider the abstraction of a widely used GNN architecture called R-GCN Schlichtkrull et al. [2018], which is applicable to multi-relational graphs. Following Barceló et al. [2020], we define R$^2$-GNN as a generalization of R-GCN by adding readout functions to the neighborhood aggregation scheme. We show that although adding readout functions enables GNNs to aggregate information of isolated nodes that can not be collected by the neighborhood-based aggregation mechanism, R$^2$-GNN are still unable to capture all Boolean node classifiers expressible as formulas in logic $\mathcal{FOC}_2$ in multi-relational scenarios if applied "directly" to the input. This leaves us with the following questions: (1) Are there reasonable and practical sub-classes of multi-relational graphs for which $\mathcal{FOC}_2$ can be captured by R$^2$-GNN? (2) Is there some simple way to encode input graphs, so that all $\mathcal{FOC}_2$ node classifiers can be captured by R$^2$-GNN for all multi-relational graphs?

In this paper, we provide answers to the above questions. Moreover, we show that our theoretical findings also transfer to temporal knowledge graphs, which are studied extensively in Park et al. [2022] and Gao and Ribeiro [2022]. In particular, we leverage the findings from Gao and Ribeiro [2022] which shows that a temporal graph can be transformed into an "equivalent" static multi-relational graph. Consequently, our results, originally formulated for static multi-relational graphs, naturally extend to the domain of temporal knowledge graphs. Our contributions are as follows:

- We calibrate the logic expressiveness of R$^2$-GNN as node classifiers over different sub-classes of multi-relational graphs.
- In light of some negative results about the expressiveness of R$^2$-GNN found in the multi-relational scenario, there is a compelling need to boost the power of R$^2$-GNN. To address this challenge, we propose a *graph transformation* and show that such a transformation enables R$^2$-GNN to capture each classifier expressible as a $\mathcal{FOC}_2$ formula in all multi-relational graphs.
- We expand the scope of expressiveness results and graph transformation from static multi-relational graphs to *temporal* settings. Within this context, we propose several temporal GNN architectures and subject them to a comparative analysis with frameworks outlined in Gao and Ribeiro [2022]. Ultimately, we derive an expressiveness hierarchy.

## 2 Preliminaries

### 2.1 Multi-relational Graphs

A *multi-relational* graph is a 4-tuple $G = (V, \mathcal{E}, P_1, P_2)$, where $V$, $P_1$, $P_2$ are finite sets of *nodes*, *types* and *relations* (a.k.a, unary/binary predicates)[3], respectively, and $\mathcal{E}$ is a set of triples of the form $(v_1, p_2, v_2)$ or $(v, type, p_1)$, where $p_1 \in P_1$, $p_2 \in P_2$, $v_1, v_2, v \in V$, and $type$ is a special symbol.

Next, given arbitrary (but fixed) finite sets $P_1$ and $P_2$ of unary and binary predicates, respectively, we define the following three kinds of graph classes:

---

[2]The "single-relational" means there is only one type of edges in the graph.

[3]For directed graphs, we assume $P_2$ contains relations both in two directions (with inverse-predicates). Moreover, we assume there exists an "equality relation" $\mathsf{EQ} \in P_2$ such that $\forall x, y \in V, x = y \Leftrightarrow \mathsf{EQ}(x,y) = 1$.

- a *universal* graph class can be any set of graphs of the form $(V, \mathcal{E}, P_1, P_2)$.
- a *bounded* graph class is a universal graph class for which there exists $n \in \mathbb{N}$ such that each graph in the class has no more than $n$ nodes;
- a *simple* graph class is a universal graph class where for each graph $(V, \mathcal{E}, P_1, P_2)$ in the class, and for each pair of nodes $v_1, v_2 \in V$, there exists at most one triple in $\mathcal{E}$ of the form $(v_1, p_2, v_2)$, where $p_2 \in P_2$.

We typically use symbols $\mathcal{G}_u$, $\mathcal{G}_b$, and $\mathcal{G}_s$ to denote universal, bounded, and simple graph classes, respectively.

**Definition 1.** *For a given graph class over predicates $P_1$ and $P_2$, a* Boolean node classifier *is a function $\mathcal{C}$ such that for each graph $G = (V, \mathcal{E}, P_1, P_2)$ in that graph class, and each $v \in V$, $\mathcal{C}$ classifies $v$ as $true$ or $false$.*

## 2.2 Graph Neural Networks

**Node Encoding**   We leverage a GNN as a Boolean node classifier for multi-relational graphs, which cannot be directly processed by GNN architectures, requiring graphs where each node is labelled by an initial feature vector. Therefore, we require some form of *encoding* to map a multi-relational graph to a suitable input for a GNN. Such an encoding should keep graph permutation invariance Geerts and Reutter [2021] since we don't want a GNN to have different outputs for isomorphic graphs. Inspired by Liu et al. [2021] for a multi-relational graph $G = (V, \mathcal{E}, P_1, P_2)$ and an ordering $p_1, p_2, \cdots, p_k$ of the predicates in $P_1$, we define an initialization function $I(\cdot)$ which maps each node $v \in V$ to a Boolean feature vector $I(v) = \mathbf{x}_v$ with a fixed dimension $|P_1|$, where the $i$th component of the vector is set to 1 if and only if the node $v$ is of the type $p_i$, that is, $(\mathbf{x}_v)_i = 1$ if and only if $(v, type, p_i) \in \mathcal{E}$. If $P_1$ is an empty set, we specify that each node has a 1-dimension feature vector whose value is 1. Clearly, this encoding is permutation invariant.

**R-GNN**   R-GCN Schlichtkrull et al. [2018] is a widely-used GNN architecture that can be applied to multi-relational graphs. By allowing different aggregation and combination functions, we extend R-GCN to a more general form which we call R-GNN. Formally, let $\left\{ \{A_j^{(i)}\}_{j=1}^{|P_2|} \right\}_{i=1}^{L}$ and $\{C^{(i)}\}_{i=1}^{L}$ be two sets of *aggregation* and *combination* functions. An R-GNN computes vectors $\mathbf{x}_v^{(i)}$ for every node $v$ of the *multi-relational* graph $G = (V, \mathcal{E}, P_1, P_2)$ on each layer $i$, via the recursive formula

$$\mathbf{x}_v^{(i)} = C^{(i)} \left( \mathbf{x}_v^{(i-1)}, \left( A_j^{(i)}(\{\!\{\mathbf{x}_u^{(i-1)} | u \in \mathcal{N}_{G,j}(v)\}\!\}) \right)_{j=1}^{|P_2|} \right) \tag{1}$$

where $x_v^{(0)}$ is the initial feature vector as encoded by $I(\cdot)$, $\{\!\{\cdot\}\!\}$ denotes a *multiset*, $(\cdot)_{j=1}^{|P_2|}$ denotes a tuple of size $|P_2|$, $\mathcal{N}_{G,j}(v)$ denotes the neighbours of $v$ via a binary relation $p_j \in P_2$, that is, nodes $w \in V$ such that $(v, p_j, w) \in \mathcal{E}$.

**$R^2$-GNN**   $R^2$-GNN extends R-GNN by specifying readout functions $\{R^{(i)}\}_{i=1}^{L}$, which aggregates the feature vectors of all the nodes in a graph. The vector $\mathbf{x}_v^{(i)}$ of each node $v$ in $G$ on each layer $i$, is computed by the following formula

$$\mathbf{x}_v^{(i)} = C^{(i)} \left( \mathbf{x}_v^{(i-1)}, \left( A_j^{(i)}(\{\!\{\mathbf{x}_u^{(i-1)} | u \in \mathcal{N}_{G,j}(v)\}\!\}) \right)_{j=1}^{|P_2|}, R^{(i)}(\{\!\{\mathbf{x}_u^{(i-1)} | u \in V\}\!\}) \right) \tag{2}$$

Every layer in an $R^2$-GNN first computes the aggregation over all the nodes in $G$; then, for every node $v$, it computes the aggregation over the neighbors of $v$; and finally, it combines the features of $v$ with the two aggregation vectors; the result of this operation is the new feature vector for $v$. Please note that an R-GNN can be seen as a special type of $R^2$-GNN where the combination function simply ignores the output of the readout function.

It is worth noting that R-GNN as well as $R^2$-GNN is not a specific model architecture; it is a framework that contains a bunch of different GNN architectures. In the paper, we mentioned it's generalized from R-GCN (Schlichtkrull et al. [2018]), but our primary objective is to establish a comprehensive framework that serves as an abstraction of most Message-Passing GNNs (MPGNN).

In the definitions (Equations (1) and (2)), the functions can be set as any functions, such as matrix multiplications or QKV-attentions. Most commonly used GNN such as R-GCN (Schlichtkrull et al. [2018]) and R-GAT (Busbridge et al. [2019]) are captured (upper-bounded) within our R-GNN frameworks. Other related works, such as (Barceló et al. [2020], Huang et al. [2023], Qiu et al. [2023]) also use intrinsically the same framework as our R-GNN/$R^2$-GNN, which has been widely adopted and studied within the GNN community. We believe that analyzing these frameworks can yield common insights applicable to numerous existing GNNs

**GNN-based Boolean node classifier** In order to translate the output of a GNN to a Boolean value, we apply a Boolean classification function $CLS : \mathbb{R}^d \to \{true, false\}$, where $d$ is the dimension of the feature vectors $\mathbf{x}_v^L$. Hence, a Boolean node classifier based on an $R^2$-GNN $\mathcal{M}$ proceeds in three steps: (1) encode the input multi-relational graph $G$ as described above, (2) apply the $R^2$-GNN, and (3) apply $CLS$ to the output of the $R^2$-GNN. This produces a $true$ or $false$ value for each node of $G$. In what follows, we abuse the language and represent a family of GNN-based Boolean node classifiers by the name of the corresponding GNN architecture; for example, $R^2$-GNN is the set of all $R^2$-GNN-based Boolean node classifiers.

## 2.3 Logic $\mathcal{FOC}_2$ Formulas

In this paper, we focus on the logic $\mathcal{FOC}_2$, a fragment of first-order logic that only allows formulas with at most two variables, but in turn permits to use *counting quantifiers*. Formally, given two finite sets $P_1$ and $P_2$ of *unary* and *binary* predicates, respectively, a $\mathcal{FOC}_2$ formula $\varphi$ is inductively defined according to the following grammar:

$$\varphi ::= A(x) \mid r(x,y) \mid \varphi \wedge \varphi \mid \varphi \vee \varphi \mid \neg\varphi \mid \exists^{\geq n} y(\varphi) \text{ where } A \in P_1 \text{ and } r \in P_2 \tag{3}$$

where $x/y$ in the above rules can be replaced by one another. But please note that $x$ and $y$ are the only variable names we are allowed to use (Though we can reuse these two names). In particular, a $\mathcal{FOC}_2$ formula $\varphi$ with exactly one free variable $x$ represents a Boolean node classifier for multi-relational graphs as follows: a node $v$ is assigned to $true$ iff the formula $\varphi_v$ obtained by substituting $x$ by $v$ is satisfied by the (logical) model represented by the multi-relational graph. Similarly as the GNN-based Boolean node classifiers, in what follows, we abuse the language and represent the family of $\mathcal{FOC}_2$ Boolean node classifiers by its name $\mathcal{FOC}_2$.

## 2.4 Inclusion and Equality Relationships

In this paper, we will mainly talk about inclusion/non-inclusion/equality/strict-inclusion relationships between different node classifier families on certain graph classes. To avoid ambiguity, we give formal definitions of these relationships here. These definitions are all quite natural.

**Definition 2.** *For any two sets of node classifier A,B, and graph class $\mathcal{G}$, We say:*

- *$A \subseteq B$ on $\mathcal{G}$, iff for any node classifier $a \in A$, there exists some node classifier $b \in B$ such that for all graph $G \in \mathcal{G}$ and $v \in V(G)$, it satisfies $a(G,v) = b(G,v)$ (Namely, $a$ and $b$ evaluate the same for all instances in $\mathcal{G}$). It implies $B$ is more expressive than $A$ on $\mathcal{G}$.*
- *$A \nsubseteq B$ on $\mathcal{G}$, iff the above condition in item 1 doesn't hold.*
- *$A \subsetneq B$ on $\mathcal{G}$, iff $A \subseteq B$ but $B \nsubseteq A$. It implies $B$ is **strictly** more expressive than $A$ on $\mathcal{G}$.*
- *$A = B$ on $\mathcal{G}$, iff $A \subseteq B$ and $B \subseteq A$. It implies $A$ and $B$ has the same expressivity on $\mathcal{G}$.*

# 3 Related Work

The relationship between first-order logic and the Weisfeiler-Lehman test was initially established by Cai et al. [1989]. Subsequently, more recent works such as Xu et al. [2018], have connected the Weisfeiler-Lehman test with expressivity of GNN. This line of research has been followed by numerous studies, including Maron et al. [2020], which explore the distinguishability of GNNs using the Weisfeiler-Lehman test technique. In particular, Barceló et al. [2020] introduced the calibration of logical expressivity in GNN-based classifiers and proposed a connection between $\mathcal{FOC}_2$ and $R^2$-GNN in single-relational scenario. This led to the emergence of related works, such as Huang et al. [2023], Geerts and Reutter [2021], and Qiu et al. [2023], all of which delve into the logical expressivity of GNNs. Moreover, the theoretical analysis provided in Gao and Ribeiro [2022] has inspired us to extend our results to temporal graph scenarios.

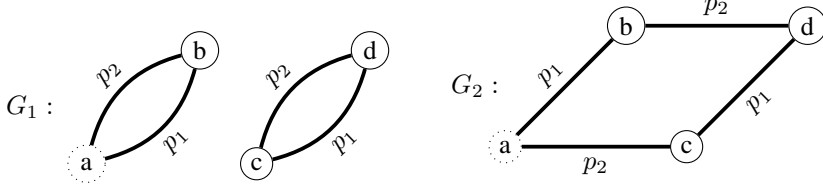

Node classifier: $\varphi(x) := \exists^{\geq 1}y(p_1(x,y) \wedge p_2(x,y))$.

Figure 1: Multi-edge graphs $G_1$ and $G_2$, and a $\mathcal{FOC}_2$ formula $\varphi(x)$ that distinguishes them; $\varphi(x)$ evaluates node $a$ in $G_1$ to $true$ and node $a$ in $G_2$ to $false$.

## 4    Logic expressiveness of R$^2$-GNN in multi-relational graphs

Our analysis begins with the observation that certain Boolean classifiers can be represented as $\mathcal{FOC}_2$ formulas, but remain beyond the expressiveness of any R$^2$-GNN (and consequently, any R-GNN or R-GCN). An illustrative example of this distinction is provided in Figure 1. In this example, we make the assumption that $P_1$ is empty, thereby ensuring that all nodes in both $G_1$ and $G_2$ possess identical initial feature vectors. Additionally, $P_2$ is defined to comprise precisely two relations, namely, $p_1$ and $p_2$. It is evident that no R$^2$-GNN can distinguish the node $a$ in $G_1$ from node $a$ in $G_2$ – that is, when an R$^2$-GNN performs the neighbour-based aggregation, it cannot distinguish whether the $p_1$-neighbour of $a$ and the $p_2$-neighbour of $a$ are the same. Moreover, the global readout aggregation cannot help in distinguishing those nodes because all nodes have the same feature vector.

We proceed to formalize this intuition and, in the reverse direction, offer a corresponding result. We demonstrate that there exist Boolean classifiers that fall within the scope of R$^2$-GNN but elude capture by any $\mathcal{FOC}_2$ formula.

**Proposition 3.** $\mathcal{FOC}_2 \not\subseteq R^2\text{-}GNN$ and $R^2\text{-}GNN \not\subseteq \mathcal{FOC}_2$ on some universal graph class.

We prove Proposition 3 in the Appendix. Here, we give some intuition about the proof. The first result is proved using the example shown in Figure 1, which we have already discussed. To show R$^2$-GNN $\not\subseteq \mathcal{FOC}_2$, we construct a classifier $c$ which classifies a node into true iff *the node has a larger number of $r_1$-type neighbors than that of $r_2$-type neighbors.* We can prove that we can easily construct an R$^2$-GNN to capture $c$. However, for $\mathcal{FOC}_2$, this cannot be done, since we can only use counting quantifiers expressing that there exist at most or at least a specific number of neighbours connected via a particular relation, but our target classifier requires comparing indefinite numbers of neighbours via two relations. Thus, we proceed by contradiction, assume that there exists a $\mathcal{FOC}_2$ classifier equivalent to $c$, and then find two large enough graphs with nodes that cannot be distinguished by the classifier (but can be distinguished by $c$).

In some real-world applications, it is often possible to find an upper bound on the size of any possible input graph or to ensure that any input graph will contain at most one relation between every two nodes. For this reason, we next present restricted but positive&practical expressiveness results on bounded and simple graph classes.

**Theorem 4.** $\mathcal{FOC}_2 \subseteq R^2\text{-}GNN$ on any simple graph class, and $\mathcal{FOC}_2 \subsetneq R^2\text{-}GNN$ on some simple graph class.

The key idea of the construction is that we will first transform the $\mathcal{FOC}_2$ formula into a new form which we call *relation-specified $\mathcal{FOC}_2$* (an equivalent form to $\mathcal{FOC}_2$, see more details in our Appendix), and then we are able to construct an equivalent R$^2$-GNN inductively over the parser tree of the transformed formula.

Having Theorem 4, one may wonder about the inclusion relationship of R$^2$-GNN and $\mathcal{FOC}_2$ in the backward direction. In Proposition 3, we showed that for arbitrary universal graph classes, this inclusion relationship fails. However, given a bounded graph class, we can show that for each R$^2$-GNN Boolean node classifier, one can write an equivalent $\mathcal{FOC}_2$ classifier. An intuition about why this is the case is that all graphs in a bounded graph class will have at most $n$ constants, for some known $n \in \mathbb{N}$, so for each R$^2$-GNN classifier, we can construct an equivalent $\mathcal{FOC}_2$ classifier with a finite number of sub-formulas to recover the features obtained at different layers of R$^2$-GNN.

**Theorem 5.** $R^2\text{-}GNN \subseteq \mathcal{FOC}_2$ on any bounded graph class, and $R^2\text{-}GNN \subsetneq \mathcal{FOC}_2$ on some bounded graph class.

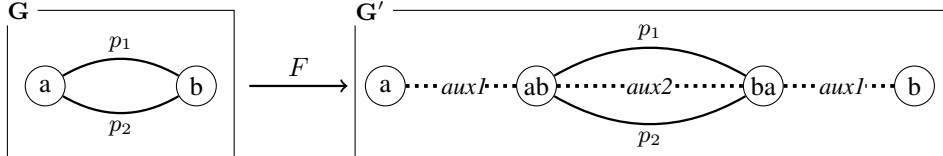

Figure 2: Graph Transformation.

Combining Theorem 4 and Theorem 5, we have the following corollary.

**Corollary 5.1.** $R^2$-*GNN* $= \mathcal{FOC}_2$ *on any* bounded simple *graph class.*

At Last, one may be curious about the complexity of logical classifier in Theorem 5. Here we can give a rather loose bound as follows:

**Theorem 6.** *For any bounded graph class $\mathcal{G}_b$. Suppose any $G \in \mathcal{G}_b$ has no more than $N$ nodes, and $\mathcal{G}_b$ has unary predicate set $P_1$ and relation (binary predicate) set $P_2$. Let $m_1 := |P_1|, m_2 := |P_2|$, then for any node classifier $c$, suppose $c$ can be represented as an $R^2$-GNN with depth (layer number) $L$, then by Theorem 5 there is a $\mathcal{FOC}_2$ classifier $\varphi$ equivalent to $c$ over $\mathcal{G}_b$, and the following hold:*

- *The quantifier depth of $\varphi$ is no more than $L$.*
- *The size of $\varphi$ (quantified by the number of nodes of $\varphi$'s parse tree) is no more than $2^{2f(L)}$, where $f(L) := 2^{2^{2(N+1)f(L-1)}}, f(0) = O(2^{2^{2(m_1+m_2)}})$.*

The key idea of Theorem 6 is the following: First, by Lemma 27 in our appendix, the combination of **ALL** $\mathcal{FOC}_2$ logical classifiers with quantifier depth no more than $L$ can already distinguish accepting and rejecting instances of $c$. Then by Proposition 26 (This is a key point of this bound; please refer to our appendix), We know the number of intrinsically different bounded-depth $\mathcal{FOC}_2$ classifiers is finite, so we only need to get an upper bound on this number. Finally, we can get the desired bound by iteratively using the fact that a boolean combination of a set of formulas can be always written as DNF (disjunctive normal form). The tower of power of two comes from $L$ rounds of DNF enumerations. Although the bound seems scary, it is a rather loose bound. We give a detailed proof of Theorem 6 in the appendix along with the proof of Theorem 5.

## 5 $R^2$-GNN capture $\mathcal{FOC}_2$ over transformed multi-relational graphs

As we pointed out in the previous section, one of the reasons why $R^2$-GNN cannot capture $\mathcal{FOC}_2$ classifiers over arbitrary universal graph classes is that in multi-relational graphs, they cannot distinguish whether information about having a neighbour connected via a particular relation comes from the same neighbour node or different neighbour nodes. Towards solving this problem, we propose a *graph transformation $F$* (see Definition 7), which enables $R^2$-GNN to capture all $\mathcal{FOC}_2$ classifiers on multi-relational graphs. Similar transformation operations have also been used and proved to be an effective way to encode multi-relational graphs in previous studies, e.g., MGNNs Tena Cucala et al. [2021], Indigo Liu et al. [2021] and Time-then-Graph Gao and Ribeiro [2022].

**Definition 7.** *Given a multigraph $G = (V, \mathcal{E}, P_1, P_2)$, the transformation $F$ will map $G$ to another graph $F(G) = (V', \mathcal{E}', P_1', P_2')$ with changes described as follows:*

- *for any two nodes $a, b \in V$, if there exists at least one relation $p \in P_2$ between $a$ and $b$, we add two new nodes $ab$ and $ba$ to $V'$.*
- *we add a new unary predicate {primal} and two new binary predicates {aux1,aux2}. Hence, $F(P_1) := P_1' = P_1 \cup \{\text{primal}\}$, and $F(P_2) := P_2' = P_2 \cup \{\text{aux1, aux2}\}$. For each node $v' \in V'$, primal$(v') = 1$ iff $v'$ is also in $V$; otherwise, primal$(v') = 0$;*
- *for each triplet of the form $(a, p_2, b)$ in $\mathcal{E}$, we add to $\mathcal{E}'$ four new triples: $(ab, \text{aux1}, a)$, $(ba, \text{aux1}, b)$ and $(ab, \text{aux2}, ba)$ as well as $(ab, p_2, ba)$.*

An example is in Figure 2. We can see that after applying the *graph transformation*, we need to execute two more hops to propagate information from node $a$ to node $b$. However, now we are able to distinguish whether the information about different relations comes from the same node or different nodes. This transformation can be implemented and stored in linear time/space complexity $O(|V| + |\mathcal{E}|)$, which is very efficient.

**Definition 8.** *Given a classifier $\mathcal{C}$ and a transformation function $F$, we define $\mathcal{C} \circ F$ to be a new classifier, an extension of $\mathcal{C}$ with an additional transformation operation on the input graph.*

With *graph transformation $F$*, we get a more powerful class of classifiers than $R^2$-GNN. We analyze the logical expressiveness of $R^2$-GNN $\circ F$ in multi-relational graphs, which means first transform a graph $G$ to $F(G)$ and then run an $R^2$-GNN on $F(G)$. We will see in the following that this transformation $F$ boosts the logical expressiveness of $R^2$-GNN prominently.

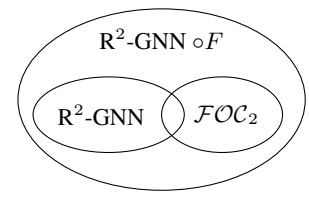

Figure 3: Relations of $R^2$-GNN, $\mathcal{FOC}_2$ and $R^2$-GNN $\circ F$.

**Theorem 9.** *$R^2$-GNN $\subseteq R^2$-GNN $\circ F$ on any* universal *graph class.*

**Theorem 10.** *$\mathcal{FOC}_2 \subseteq R^2$-GNN $\circ F$ on any* universal *graph class.*

Theorem 9 demonstrates that $R^2$-GNN with *graph transformation $F$* have more expressiveness than $R^2$-GNN; and Theorem 10 shows the connection between $\mathcal{FOC}_2$ and $R^2$-GNN equipped with *graph transformation $F$*. We depict their relations in Figure 3. Theorem 9 is a natural result since no information is lost in the process of transformation, while Theorem 10 is an extension on Theorem 4, whose formal proofs can be found in the Appendix. As for the backward direction, we have the result shown in Theorem 11.

**Theorem 11.** *$R^2$-GNN $\circ F \subseteq \mathcal{FOC}_2$ on any* bounded *graph class.*

The proof of the theorem is relatively straightforward based on previous results: by Theorem 5, it follows that $R^2$-GNN $\circ F \subseteq \mathcal{FOC}_2 \circ F$ on any bounded graph class. Then, it suffices to prove $\mathcal{FOC}_2 \circ F \subseteq \mathcal{FOC}_2$, which we do by using induction over the quantifier depth.

By combining Theorem 10 and Theorem 11, we obtain Corollary 11.1, stating that $\mathcal{FOC}_2$ and $R^2$-GNN $\circ F$ have the same expressiveness with respect to bounded graph classes. Corollary 11.1 does not hold for arbitrary universal graph classes, but our finding is nevertheless exciting because, in many real-world applications there are upper bounds over input graph size.

**Corollary 11.1.** *$R^2$-GNN $\circ F = \mathcal{FOC}_2$ on any bounded graph class.*

To show the strict separation as in Figure 3, we can combine Proposition 3 and theorems 4 and 9 and Theorem 10 to directly get the following:

**Corollary 11.2.** *$R^2$-GNN $\subsetneq R^2$-GNN $\circ F$ on some universal graph class, and $\mathcal{FOC}_2 \subsetneq R^2$-GNN $\circ F$ on some simple graph class.*

One may think after transformation $F$, the logic $\mathcal{FOC}_2 \circ F$ with new predicateds becomes stronger as well. However by a similar proof as for Theorem 10 and Lemma 28, we can actually show $\mathcal{FOC}_2 \circ F \subseteq \mathcal{FOC}_2$ always holds, so $F$ won't bring added power for $\mathcal{FOC}_2$. However, it indeed make $R^2$-GNN strictly more expressive.

## 6 Temporal Graphs

As stated in Gao and Ribeiro [2022], a temporal knowledge graph, composed of multiple snapshots, can consistently undergo transformation into an equivalent static representation as a multi-relational graph. Consequently, this signifies that our theoretical results initially devised for multi-relational graphs can be extended to apply to temporal graphs, albeit through a certain manner of transfer.

Following previous work Jin et al. [2019], Pareja et al. [2020], Park et al. [2022], Gao and Ribeiro [2022], we define a temporal knowledge graph as a set of graph "snapshots" distributed over a sequence of **finite** and **discrete** time points $\{1, 2, \ldots, T\}$. Formally, a temporal knowledge graph is a set $G = \{G_1, \cdots, G_T\}$ for some $T \in \mathbb{N}$, where each $G_t$ is a static multi-relational graph. All these $G_t$ share the same node set and predicate set.

In a temporal knowledge graph, a relation or unary fact between two nodes might hold or disappear across the given timestamps. For example, a node $a$ may be connected to a node $b$ via a relation $p$ in the first snapshot, but not in the second; in this case, we have $(a, p, b)$ in $G_1$ not in $G_2$. To keep track of which relations hold at which snapshots, we propose *temporal predicates*, an operation which we define in Definition 12.

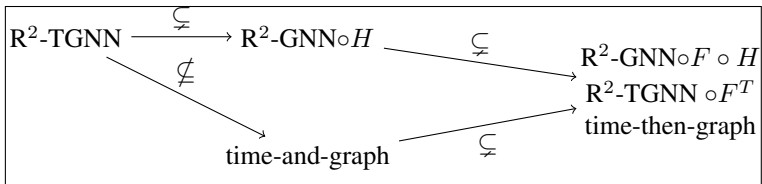

Figure 4: Hierarchic expressiveness.

**Definition 12.** *Given a temporal graph* $G = \{G_1, \cdots, G_T\}$, *where each* $G_t$ *is of the form* $(V_t, \mathcal{E}_t, P_1, P_2)$, *temporal predicates are obtained from* $G$ *by replacing, for each* $t \in \{1, \ldots, T\}$ *and each* $p \in P_2$, *each triple* $(v_a, p, v_b) \in \mathcal{E}_t$ *with* $(v_a, p^t, v_b)$, *where* $p^t$ *is a fresh predicate, unique for* $p$ *and* $t$. *Similarly, each unary fact* $(v_a, q) \in \mathcal{E}_t, q \in P_1$ *should be replaced by* $(v_a, q^t)$.

Note that temporalising introduces $T \times |P|$ new predicates in total. By *temporalizing predicates*, we assign a superscript to each predicate and use it to distinguish relations over different timestamps.

**Definition 13.** *Given a temporal knowledge graph* $G = \{G_1, \ldots, G_T\}$, *the collapse function* $H$ *maps* $G$ *to the static graph* $H(G)$ *obtained by taking the union of graphs over all timestamps in the temporalization of* $G$.

As we have proved in Section 5, for multi-relational graphs, $R^2$-GNN with *graph transformation* is more powerful than the pure $R^2$-GNN. Here, we transfer these theoretical findings in multi-relational graphs to the setting of temporal knowledge graphs. To be more specific, after *temporalizing predicates*, we apply a *graph transformation* to each graph snapshot.

**Definition 14.** *We define* $F^T$ *to be the temporal graph transformation that takes any temporal knowledge graph as input, applies* graph transformation *to each snapshot and outputs. Specially, non-primal nodes,* aux1 *and* aux2 *edges added in any snapshot should be added into all snapshots.*

**$R^2$-TGNN**    Gao and Ribeiro [2022] casts node representation in temporal graphs into two frameworks: *time-and-graph* and *time-then-graph*. Due to space constraints, we refer interested readers to Gao and Ribeiro [2022] for more details about the two frameworks. Here, we define a more general GNN-based framework abbreviated as $R^2$-TGNN, where each $R^2$-TGNN is a sequence $\{\mathcal{A}_t\}_{t=1}^T$, where each $\mathcal{A}_t$ is an $R^2$-GNN model. Given a temporal knowledge graph $G = \{G_1, \ldots, G_T\}$, where $G_t = (V_t, \mathcal{E}_t, P_1, P_2)$ for each $t \in \{1, \ldots, T\}$. The updating rule is as follows:

$$\mathbf{x}_v^t = \mathcal{A}_t\left(G_t, v, \mathbf{y}^t\right) \quad \text{where} \quad \mathbf{y}_v^t = [I_{G_t}(v) : \mathbf{x}_v^{t-1}], \forall v \in V(G_t) \tag{4}$$

where $I_{G_t}(v)$ is the one-hot initial feature vector of node $v$ at timestamp $t$, and $\mathcal{A}_t(G_t, v, \mathbf{y}^t)$ calculates the new feature vector of $v$ by running the $R^2$-GNN model $\mathcal{A}_t$ on $G_t$, but using $\mathbf{y}^t$ as the initial feature vectors. As shown in Theorem 15, $R^2$-TGNN composed with $F^T$ have the same expressiveness as *time-then-graph*[4], while being more powerful than *time-and-graph*.

**Theorem 15.** *time-and-graph* $\subsetneq R^2$-*TGNN* $\circ F^T = $ *time-then-graph*.

We also establish the validity of Theorem 16, which asserts that $R^2$-TGNN with *graph transformation* maintains the same expressive power, whether it is applied directly to the temporal graph or to the equivalent collapsed static multi-relational graph

**Theorem 16.** $R^2$-*TGNN* $\circ F^T = R^2$-*GNN* $\circ F \circ H$

We also prove a strict inclusion that $R^2$-TGNN $\subsetneq R^2$-TGNN$\circ H$. Finally we get the following hierarchy of these frameworks as in Figure 6. the proof of Theorem 17 is in the appendix.

**Theorem 17.** *The following hold:*

- *$R^2$-GNN $\subsetneq R^2$-GNN $\circ H \subsetneq R^2$-TGNN $\circ F \circ H = R^2$-TGNN $\circ F^T = $ time-then-graph.*
- *time-and-graph $\subsetneq R^2$-TGNN $\circ F^T$.*
- *$R^2$-TGNN $\nsubseteq$ time-and-graph.*

[4]Since temporalized predicates and timestamps make the definitions of bounded/simple/universal graph class vague, we no longer distinguish them in temporal settings. In theorem statements of this section, $=$, $\subseteq$ always hold for any temporal graph class, and $\subsetneq$, $\nsubseteq$ hold for some temporal graph class

| $\mathcal{FOC}_2$ classifier | $\varphi_\mathbf{1}$ | | | $\varphi_\mathbf{2}$ | | | $\varphi_\mathbf{3}$ | | | $\varphi_\mathbf{4}$ | | |
|---|---|---|---|---|---|---|---|---|---|---|---|---|
| Aggregation | sum | max | mean | sum | max | mean | sum | max | mean | sum | max | mean |
| Temporal Graphs Setting | | | | | | | | | | | | |
| R-TGNN | 100 | 60.7 | 65.4 | 61.0 | 51.3 | 52.4 | 93.7 | 82.3 | 84.4 | 83.5 | 60.0 | 61.3 |
| $R^2$-TGNN | 100 | 63.5 | 66.8 | 93.1 | 57.7 | 60.2 | 94.5 | 83.3 | 85.9 | 85.0 | 62.3 | 66.2 |
| $R^2$-TGNN $\circ F^T$ | **100** | 67.2 | 68.1 | **99.0** | 57.6 | 62.2 | **100** | 88.8 | 89.2 | **98.1** | 73.4 | 77.5 |
| Aggregated Static Graphs Setting | | | | | | | | | | | | |
| R-GNN $\circ H$ | 100 | 61.2 | 69.9 | 62.3 | 51.3 | 55.5 | 94.7 | 80.5 | 83.2 | 80.2 | 60.1 | 60.4 |
| $R^2$-GNN $\circ H$ | 100 | 62.7 | 66.8 | 92.4 | 56.3 | 58.5 | 95.5 | 84.2 | 85.2 | 81.0 | 58.3 | 64.5 |
| $R^2$-GNN $\circ F \circ H$ | **100** | 70.2 | 70.8 | **98.8** | 60.6 | 60.2 | **100** | 85.6 | 86.5 7 | **95.5** | 70.3 | 79.7 |

Table 1: Test set node classification accuracies (%) on synthetic temporal multi-relational graphs datasets and their aggregated static multi-relational graphs datasets. The best results are highlighted for two different settings.

# 7 Experiment

We empirically verify our theoretical findings for multi-relational graphs by evaluating and comparing the testing performance of $R^2$-GNN with *graph transformation* and less powerful GNNs (R-GNN and $R^2$-GNN). We did two groups of experiments on synthetic datasets and real-world datasets, respectively. Details for datasets generation and statistical information as well as hyper-parameters can be found in the Appendix.

## 7.1 Synthetic Datasets

We first define three simple $\mathcal{FOC}_2$ classifiers

$$\varphi_\mathbf{1} := \exists^{\geq 2} y (p_1^1(x,y) \wedge Red^1(y)) \wedge \exists^{\geq 1} y(p_1^2(x,y) \wedge Blue^2(y))$$

$$\varphi_\mathbf{2} := \exists^{[10,20]} y(\neg p_1^2(x,y) \wedge \varphi_1(y)) \qquad \varphi_\mathbf{3} := \exists^{\geq 2} y(p_1^1(x,y) \wedge p_1^2(x,y))$$

Besides, we define another complicate $\mathcal{FOC}_2$ classifier denoted as $\varphi_4$ shown as follows:

$$\varphi_\mathbf{4} := \bigvee_{3 \leq t \leq 10} (\exists^{\geq 2} y(Black^t(y) \wedge Red^{t-1}(y) \wedge Blue^{t-2}(y) \wedge p_1^t(x,y) \wedge p_2^{t-1}(x,y) \wedge p_3^{t-2}(x,y) \wedge \varphi^t(y))$$

$$\text{where} \quad \varphi^t(y) := \exists^{\geq 2} x(p_1^t(x,y) \wedge Red^t(x)) \wedge \exists^{\geq 1} x(p_2^{t-1}(x,y) \wedge Blue^{t-2}(x))$$

For each of them, we generate an independent dataset containing 7k multi-relational graphs of size up to 50-1000 nodes for training and 500 multi-relational graphs of size similar to the train set. We tried different configurations for the aggregation functions and evaluated the node classification performances of three temporal GNN methods (R-TGNNs, $R^2$-TGNNs and $R^2$-TGNNs $\circ F^T$) on these datasets.

We verify our hypothesis empirically according to models' actual performances of fitting these three classifiers. Theoretically, $\varphi_1$ should be captured by all three models because the classification result of a node is decided by the information of its neighbor nodes, which can be accomplished by the general neighborhood based aggregation mechanism. $\varphi_2$ should not be captured by R-TGNN because the use of $\neg p_1^2(x,y)$ as a guard means that the classification result of a node depends on the global information including those isolated nodes, which needs a global readout. For $\varphi_3$ and $\varphi_4$, they should only be captured by $R^2$-TGNNs $\circ F^T$. An intuitive explanation for this argument is that if we *temporalise predicates* and then collapse the temporal graph into its equivalent static multi-relational graph using $H$, we will encounter the same issue as in the Figure 1. Thus we can't distinguish expected nodes without *graph transformation*.

Results for temporal GNN methods and static GNN methods on four synthetic datasets can be found in Table 1. We can see that $R^2$-GNN with *graph transformation* achieves the best performance. Our theoretical findings show that it is a more expressive model, and the experiments indeed suggest that the model can exploit this theoretical expressiveness advantage to produce better results. Besides, we can also see that $R^2$-TGNN $\circ F \circ H$ and $R^2$-TGNN $\circ F^T$ achieve almost the same performance, which is in line with Theorem 16.

Table 2: Results on temporal graphs.

| Models | Category | Source | Brain-10 | | |
|---|---|---|---|---|---|
| | | | sum | max | mean |
| GCRN-M2 | time-and-graph | Seo et al. [2018] | 77.0 | 61.2 | 73.1 |
| DCRNN | time-and-graph | Li et al. [2018] | 84.0 | 70.1 | 66.5 |
| TGAT | time-then-graph | Xu et al. [2020] | 80.0 | 72.3 | 79.0 |
| TGN | time-then-graph | Rossi et al. [2020a] | 91.2 | **88.5** | 89.2 |
| GRU-GCN | time-then-graph | Gao and Ribeiro [2022] | 91.6 | 88.2 | 87.1 |
| R-TGNN | – | – | 85.0 | 82.3 | 82.8 |
| $R^2$-TGNN | – | – | **94.8** | 82.3 | 91.0 |
| $R^2$-TGNN $\circ F^T$ | – | – | 94.0 | 83.5 | **92.5** |

## 7.2 Real-world Datasets

For real-world static multi-relational graphs benchmarks, we used AIFB and MUTAG from Ristoski and Paulheim [2016]. Since open source datasets for the node classification on temporal knowledge graphs are rare, we only tried one dataset Brain-10 Gao and Ribeiro [2022] for temporal settings.[5]

Table 3: Results on two static multi-relational graphs.

| Models | AIFB | | | MUTAG | | |
|---|---|---|---|---|---|---|
| | sum | max | mean | sum | max | mean |
| R-GNN | **91.7** | 73.8 | 82.5 | **76.5** | 63.3 | 73.2 |
| $R^2$-GNN | **91.7** | 73.8 | 82.5 | **85.3** | 62.1 | 79.5 |
| $R^2$-GNN $\circ F$ | **97.2** | 75.0 | 89.2 | **88.2** | 65.5 | 82.1 |
| R-GCN | **95.8** | 77.9 | 86.3 | **73.2** | 65.7 | 72.1 |

For static multi-relational graphs, we compare the performances of our methods with RGCN Schlichtkrull et al. [2018]. Note that RGCN assigns each node an index and the initial embedding of each node is initialised based on the node index, so the initialisation functional is not permutation-equivariant Chen et al. [2019a] and RGCN cannot be used to perform an isomorphism test. However, from Table 3, we can see that $R^2$-GNN with *graph transformation* still achieves the highest accuracy while being able to be used for the graph isomorphism test. Besides, $R^2$-GNN $\circ F$ also performs better compared with both R-GNN and $R^2$-GNN. This again suggests that the extra expressive power gained by adding a *graph transformation* step to $R^2$-GNN can be exploited by the model to obtain better results.

For temporal graphs, Gao and Ribeiro [2022] have classified existing temporal models into two categories, *time-and-graph* and *time-then-graph*, and shown that *time-then-graph* models have better performance. We choose five models mentioned in Gao and Ribeiro [2022] as our baseline and include the best accuracy of the dataset Brain-10 reported in Gao and Ribeiro [2022]. As we expected, $R^2$-TGNNand $R^2$-TGNN $\circ F^T$ achieve better performance than that of the baseline models and R-TGNN accoring to Table 2. However, we observed that although in theory, $R^2$-TGNN $\circ F^T$ has stronger expressive power than $R^2$-TGNN, we did not see an improvement when using $R^2$-TGNN $\circ F^T$ (0.8% accuracy drop). To some extent, it may show that some commonly used benchmarks are inadequate for testing advanced GNN variants. Similar phenomena have also been observed in previous works Chen et al. [2019b], Barceló et al. [2020].

## 8 Conclusion

We analyze expressivity of $R^2$-GNNs with and without *graph transformation* in multi-relational graphs under different situations. Furthermore, we extend our theoretical findings to the temporal graph setting. Our experimental results confirm our theoretical insights, particularly demonstrating the state-of-the-art performance achieved by our *graph transformation* technique.

---

[5]The other three temporal dataset mentioned in Gao and Ribeiro [2022] are not released.

# 9 Acknowledgements

The authors extend their gratitude to Bernardo Cuenca Grau and David Tena Cucala for their valuable insights, stimulating discussions, and support.

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

# Appendix

## Table of Contents

# A Preliminaries for Proofs

In this section, we give some preliminaries which will be used to prove the theorems, propositions and lemmas shown in our main body. In what follows, we fix a unary predicate set $P_1$ and a binary predicate set $P_2$.

**Definition 18.** *For an $R^2$-GNN, we say it is a 0/1-GNN if the recursive formula used to compute vectors $\mathbf{x}_v^{(i)}$ for each node $v$ in a multi-relational graph $G = \{V, \mathcal{E}, P_1, P_2\}$ on each layer $i$ is in the following form*

$$\mathbf{x}_v^{(i)} = f\left(C^{(i)}\left(\mathbf{x}_v^{(i-1)} + \sum_{r \in P_2}\sum_{u \in V} A_r^{(i)}\mathbf{x}_u^{(i-1)} + R^{(i)}\left(\sum_{u \in V}\mathbf{x}_u^{(i-1)}\right) + b^{(i)}\right)\right) \tag{5}$$

*where $C^{(i)}, A_j^{(i)}, R^{(i)}$ are all integer matrices of size $d_i \times d_{i-1}$, $b^{(i)}$ is bias column vector with size $d_i \times 1$, where $d_{i-1}$ and $d_i$ are input/output dimensions, and $f$ is defined as $max(0, min(x, 1))$.*

*Furthermore, we restrict the final output dimension be $d_L = 1$. Since all matrices have integer elements, initial vectors are integer vectors by initialisation function $I(\cdot)$ (Section 2.2), and $max(0, min(x, 1))$ will map all integers to $0/1$, it's easy to see that the output of this kind of model is always $0/1$, which can be directly used as the classification result. We call such model 0/1-GNN. A model instance can be represented by $\{C^{(i)}, (A_j^{(i)})_{j=1}^K, R^{(i)}, b^{(i)}\}_{i=1}^L$, where $K = |P_2|$*

**Lemma 19.** *Regard 0/1-GNN as node classifier, then the set of node classifiers represented by 0/1-GNN is closed under $\wedge, \vee, \neg$.*

*Proof.* Given two 0/1-GNN $\mathcal{A}_1, \mathcal{A}_2$, it suffices to show that we can construct $\neg\mathcal{A}_1$ and $\mathcal{A}_1 \wedge \mathcal{A}_2$ in 0/1-GNN framework. That's because construction of $\mathcal{A}_1 \vee \mathcal{A}_2$ can be reduced to constructions of $\wedge, \neg$ by De Morgan's law, e.g., $a \vee b = \neg(\neg a \wedge \neg b)$.

1. Construct $\neg\mathcal{A}_1$. Append a new layer to $\mathcal{A}_1$ with dimension $d_{L+1} = 1$. For matrices and bias $C^{(L+1)}, (A_j^{(L+1)})_{j=1}^K, R^{(L+1)}, b^{(L+1)}$ in layer $L+1$, set $C_{1,1}^{L+1} = -1$ and $b_1^{L+1} = 1$ and other parameters 0. Then it follows $\mathbf{x}_v^{(L+1)} = max(0, min(-\mathbf{x}_v^{(L)} + 1, 1))$. Since $\mathbf{x}_v^{(L)}$ is the 0/1 classification result outputted by $\mathcal{A}_1$. It's easy to see that the above equation is exactly $\mathbf{x}_v^{(L+1)} = \neg\mathbf{x}_v^{(L)}$

2. Construct $\mathcal{A}_1 \wedge \mathcal{A}_2$. Without loss of generality, we can assume two models have same layer number $L$ and same feature dimension $d_l$ in each layer $l \in \{1......L\}$. Then, we can construct a new 0/1-GNN $\mathcal{A}$. $\mathcal{A}$ has $L+1$ layers. For each of the first $L$ layers, say $l$-th layer, it has feature dimension $2d_l$. Let $\{C_1^{(l)}, (A_{j,1}^{(l)})_{j=1}^K, R_1^{(l)}, b_1^{(l)}\}, \{C_2^{(l)}, (A_{j,2}^{(l)})_{j=1}^K, R_2^{(l)}, b_2^{(l)}\}$ be parameters in layer $l$ of $\mathcal{A}_1, \mathcal{A}_2$ respectively. Parameters for layer $l$ of $\mathcal{A}$ are defined below

$$\mathbf{C}^{(l)} := \begin{bmatrix} \mathbf{C}_1^{(l)} & \\ & \mathbf{C}_2^{(l)} \end{bmatrix} \mathbf{A}_j^{(l)} := \begin{bmatrix} \mathbf{A}_{j,1}^{(l)} & \\ & \mathbf{A}_{j,2}^{(l)} \end{bmatrix} \mathbf{R}^{(l)} := \begin{bmatrix} \mathbf{R}_1^{(l)} & \\ & \mathbf{R}_2^{(l)} \end{bmatrix} \mathbf{b}^{(l)} := \begin{bmatrix} \mathbf{b}_1^{(l)} \\ \mathbf{b}_2^{(l)} \end{bmatrix} \tag{6}$$

Initialization function of $\mathcal{A}$ is concatenation of initial feature of $\mathcal{A}_1, \mathcal{A}_2$. Then it's easy to see that the feature $\mathbf{x}_v^L$ after running first $L$ layers of $\mathcal{A}$ is a two dimension vector, and the two dimensions contains two values representing the classification results outputted by $\mathcal{A}_1, \mathcal{A}_2$ respectively.

For the last layer $L+1$, it has only one output dimension. We just set $\mathbf{C}_{1,1}^{L+1} = \mathbf{C}_{1,2}^{L+1} = 1, \mathbf{b}_1^{L+1} = -1$ and all other parameters 0. Then it's equivalent to $\mathbf{x}_v^{(L+1)} = max(0, min(\mathbf{x}_{v,1}^{(L)} + \mathbf{x}_{v,2}^{(L)} - 1, 1))$ where $\mathbf{x}_{v,1}^{(L)}, \mathbf{x}_{v,2}^{(L)}$ are output of $\mathcal{A}_1, \mathcal{A}_2$ respectively. It's easy to see that the above equation is equivalent to $\mathbf{x}_v^{(L+1)} = \mathbf{x}_{v,1}^{(L)} \wedge \mathbf{x}_{v,2}^{(L)}$ so the $\mathcal{A}$ constructed in this way is exactly $\mathcal{A}_1 \wedge \mathcal{A}_2$ □

**Definition 20.** *A $\mathcal{FOC}_2$ formula is defined inductively according to the following grammar:*

$$A(x), r(x,y), \varphi_1 \wedge \varphi_2, \varphi_1 \vee \varphi_2, \neg\varphi_1, \exists^{\geq n}y(\varphi_1(x,y)) \text{ where } A \in P_1 \text{ and } r \in P_2 \tag{7}$$

**Definition 21.** *For any subset $S \subseteq P_2$, let $\varphi_S(x,y)$ denote the $\mathcal{FOC}_2$ formula $(\bigwedge_{r \in S} r(x,y)) \wedge (\bigwedge_{r \in P_2 \setminus S} \neg r(x,y))$. Note that $\varphi_S(x,y)$ means there is a relation $r$ between $x$ and $y$ if and only if $r \in S$, so $\varphi_S(x,y)$ can be seen as a formula to restrict specific relation distribution between two nodes. $\mathcal{RSFOC}_2$ is inductively defined according to the following grammar:*

$$A(x), \varphi_1 \wedge \varphi_2, \ \varphi_1 \vee \varphi_2, \ \neg\varphi_1, \exists^{\geq n} y \Big( \varphi_S(x,y) \wedge \varphi_1(y) \Big) \text{ where } A \in P_1 \text{ and } S \subseteq P_2 \quad (8)$$

Next, we prove that $\mathcal{FOC}_2$ and $\mathcal{RSFOC}_2$ have the same expressiveness, namely, each $\mathcal{FOC}_2$ node classifier can be rewritten in the form $\mathcal{RSFOC}_2$.

**Lemma 22.** $\mathcal{FOC}_2 = \mathcal{RSFOC}_2$.

*Proof.* Comparing the definitions of $\mathcal{RSFOC}_2$ and $\mathcal{FOC}_2$, it is obvious that $\mathcal{RSFOC}_2 \subseteq \mathcal{FOC}_2$ trivially holds, so we only need to prove the other direction, namely, $\mathcal{FOC}_2 \subseteq \mathcal{RSFOC}_2$. In particular, a Boolean logical classifier only contains one free variable, we only need to prove that for any one-free-variable $\mathcal{FOC}_2$ formula $\varphi(x)$, we can construct an equivalent $\mathcal{RSFOC}_2$ formula $\psi(x)$.

We prove Lemma 22 by induction over $k$, where $k$ is the quantifier depth of $\varphi(x)$.

In the base case where $k = 0$, $\varphi(x)$ is just the result of applying conjunction, disjunction or negation to a bunch of unary predicates $A(x)$, where $A \in P_1$. Given that the grammar of generating $\varphi(x)$ is the same in $\mathcal{RSFOC}_2$ and $\mathcal{FOC}_2$ when $k = 0$, so the lemma holds for $k = 0$.

For the indutive step, we assume that Lemma 22 holds for all $\mathcal{RSFOC}_2$ formula with quantifier depth no more than $m$, we next need to consider the case when $k = m + 1$.

We can decompose $\varphi(x)$ to be boolean combination of a bunch of $\mathcal{FOC}_2$ formulas $\varphi_1(x), \ldots, \varphi_N(x)$, each of which is in the form $\varphi_i(x) := A(x)$ where $A \in P_1$ or $\varphi_i(x) := \exists^{\geq n} y(\varphi'(x,y))$. See the following example for reference.

**Example 23.** *Assume $\varphi(x) := \big(A_1(x) \wedge \exists y(r_1(x,y))\big) \vee \big(\exists y \big(A_2(y) \wedge r_2(x,y)\big) \wedge \exists y(r_3(x,y))\big)$. It can be decomposed into boolean combination of four subformulas shown as follows:*

- $\varphi_1(x) = A_1(x)$
- $\varphi_2(x) = \exists y(r_1(x,y))$
- $\varphi_3(x) = \exists y\big(A_2(y) \wedge r_2(x,y)\big)$
- $\varphi_4(x) = \exists y(r_3(x,y))$

We can see that grammars of $\mathcal{FOC}_2$ and $\mathcal{RSFOC}_2$ have a common part: $A(x), \varphi_1 \wedge \varphi_2, \varphi_1 \vee \varphi_2, \neg\varphi_1$, so we can only focus on those subformulas $\varphi_i(x)$ in the form of $\exists^{\geq n} y \varphi'(x,y)$. In other words, if we can rewrite these $\mathcal{FOC}_2$ subformulas into another form satisfying the grammar of $\mathcal{RSFOC}_2$, we can naturally construct the desired $\mathcal{RSFOC}_2$ formula $\psi(x)$ equivalent to $\mathcal{FOC}_2$ formula $\varphi(x)$.

Without loss of generality, in what follows, we consider the construction for $\varphi(x) = \exists^{\geq n} y(\varphi'(x,y))$. Note that $\varphi(x)$ has quantifier depth no more than $m + 1$, and $\varphi'(x,y)$ has quantifier depth no more than $m$.

We can decompose $\varphi'(x,y)$ into three sets of subformulas $\{\varphi_i^x(x)\}_{i=1}^{N_x}, \{\varphi_i^y(y)\}_{i=1}^{N_y}, \{r_i(x,y)\}_{i=1}^{|P_2|}$, where $N_x$ and $N_y$ are two natural numbers, $\varphi_i^x, \varphi_i^y$ are its maximal subformulas whose free variable is assigned to $x$ and $y$, respectively. $\varphi'(x)$ is the combination of these sets of subformulas using $\wedge, \vee, \neg$.

**Example 24.** *Assume that we have a $\mathcal{FOC}_2$ formula in the form of $\varphi'(x,y) = \Big(r_1(x,y) \wedge \exists x(r_2(x,y))\Big) \vee \Big(\exists y\big(\exists x(r_3(x,y)) \vee \exists y(r_1(x,y))\big) \wedge \exists y\big(A_2(y) \wedge r_2(x,y)\big)\Big)$*

*It can be decomposed into the following subformulas:*

- $\varphi_1^x(x) := \exists y\big(\exists x(r_3(x,y)) \vee \exists y(r_1(x,y))\big);$
- $\varphi_2^x(x) := \exists y\big(A_2(y) \wedge r_2(x,y)\big);$
- $\varphi_1^y(y) := \exists x(r_2(x,y));$
- $r_1(x,y)$

Assume that $N := \{1, \ldots, N_x\}$, we construct a $\mathcal{RSFOC}_2$ formula $\varphi_T^x(x) := (\bigwedge_{i \in T} \varphi_i^x(x)) \wedge (\bigwedge_{i \in N \setminus T} \neg\varphi_i^x(x))$, where $T \subseteq N$. It is called the *x-specification* formula, which means $\varphi_T^x(x)$ is *true* iff the following condition holds: for all $i \in T$, $\varphi_i^x(x)$ is *true* and for all $i \in N \setminus T$, $\varphi_i^x(x)$ is *false*.

By decomposing $\varphi'(x,y)$ into three subformula sets, we know Boolean value of $\varphi'(x,y)$ can be decided by Boolean values of these formulas $\{\varphi_i^x(x)\}_{i=1}^{N_x}, \{\varphi_i^y(y)\}_{i=1}^{N_y}, \{r_i(x,y)\}_{i=1}^{|P_2|}$. Now for any two specific subsets $S \subseteq P_2, T \subset N$, we assume $\varphi_S(x,y)$ and $\varphi_T^x(x)$ are all *true* (Recall the definition of $\varphi_S(x,y)$ in Definition 21). Then Boolean values for formulas in $\{\varphi_i^x(x)\}_{i=1}^{N_x}, \{r_i(x,y)\}_{i=1}^{|P_2|}$ are determined and Boolean value of $\varphi'(x,y)$ depends only on Boolean values of $\{\varphi_i^y(y)\}_{i=1}^{N_y}$. Therefore, we can write a new $\mathcal{FOC}_2$ formula $\varphi_{S,T}^y(y)$ which is a boolean combination of $\{\varphi_i^y(y)\}_{i=1}^{N_y}$. This formula should satisfy the following condition: For any graph $G$ and two nodes $a,b$ on it, the following holds,

$$\varphi_S(a,b) \wedge \varphi_T^x(a) \Rightarrow \left( \varphi'(a,b) \Leftrightarrow \varphi_{S,T}^y(b) \right) \tag{9}$$

By our inductive assumption, $\varphi'(x,y)$ has a quantifier depth which is no more than $m$, so $\{\varphi_i^x(y)\}_{i=1}^{N_x}, \{\varphi_i^y(y)\}_{i=1}^{N_y}$ also have quantifier depths no more than $m$. Therefore, each of them has $\mathcal{RSFOC}_2$ correspondence. Furthermore, since $\wedge, \vee, \neg$ are allowed operation in $\mathcal{RSFOC}_2$, $\varphi_T^x(x)$ and $\varphi_{S,T}^y(y)$ can also be rewritten as $\mathcal{RSFOC}_2$ formulas.

Given that $\varphi_S(x,y)$ and $\varphi_T^x(y)$ specify the boolean values for all $\{\varphi_i^x(y)\}_{i=1}^{N_x}, \{\varphi_i^r(x,y)\}_{i=1}^{|P_2|}$ formulas, so we can enumerate all possibilities over $S \subseteq P_2$ and $T \subseteq N$. Obviously for any graph $G$ and a node pair $(a,b)$, there exists an unique $(S,T)$ pair such that $\varphi_S(a,b) \wedge \varphi_T^x(a)$ holds.

Hence, combining Equation (9), $\varphi'(x,y)$ is true only when there exists a $(S,T)$ pair such that $\varphi_S(x,y) \wedge \varphi_T^x(x) \wedge \varphi_{S,T}^y(y)$ is *true*. Formally, we can rewrite $\varphi'(x,y)$ as following form:

$$\varphi'(x,y) \equiv \bigvee_{S \subseteq P_2, T \subseteq N} \left( \varphi_S(x,y) \wedge \varphi_T^x(x) \wedge \varphi_{S,T}^y(y) \right) \tag{10}$$

In order to simplify the formula above, let $\phi_T(x)$ denote the following formula:

$$\phi_T(x,y) := \bigvee_{S \subseteq P_2} \left( \varphi_S(x,y) \wedge \varphi_{S,T}^y(y) \right) \tag{11}$$

Then we can simplify Equation (10) to the following form:

$$\varphi'(x,y) \equiv \bigvee_{T \subseteq N} \left( \varphi_T^x(x) \wedge \phi_T(x,y) \right) \tag{12}$$

Recall that $\varphi(x) = \exists^{\geq n} y (\varphi'(x,y))$, so it can be rewritten as:

$$\varphi(x) \equiv \exists^{\geq n} y \left( \bigvee_{T \subseteq N} \left( \varphi_T^x(x) \wedge \phi_T(x,y) \right) \right) \tag{13}$$

Since for any graph $G$ and its node $a$, there exists exactly one $T$ such that $\varphi_T^x(a)$ is *true*. Therefore, Equation (13) can be rewritten as the following formula:

$$\varphi(x) \equiv \bigvee_{T \subseteq N} \left( \varphi_T^x(x) \wedge \exists^{\geq n} y (\phi_T(x,y)) \right) \tag{14}$$

Let $\widehat{\varphi}_T(x) := \exists^{\geq n} y (\phi_T(x,y))$. Since $\wedge, \vee$ are both allowed in $\mathcal{RSFOC}_2$. If we want to rewrite $\varphi(x)$ in the $\mathcal{RSFOC}_2$ form, it suffices to rewrite $\widehat{\varphi}_T(x)$ as a $\mathcal{RSFOC}_2$ formula, which is shown as follows,

$$\widehat{\varphi}_T(x) := \exists^{\geq n} y (\phi_T(x,y)) = \exists^{\geq n} y \left( \bigvee_{S \subseteq P_2} \left( \varphi_S(x,y) \wedge \varphi_{S,T}^y(y) \right) \right) \tag{15}$$

Similar to the previous argument, since for any graph $G$ and of of its node pairs $(a,b)$, the *relation-specification* formula $\varphi_S(x,y)$ restricts exactly which types of relations exists between $(a,b)$, there is exactly one subset $S \subseteq P_2$ such that $\varphi_S(a,b)$ holds.

Therefore, for all $S \subseteq P_2$, we can define $n_S$ as the number of nodes $y$ such that $\varphi_S(x,y) \wedge \varphi_{S,T}^y(y)$ holds. Since for two different subsets $S_1, S_2 \subseteq P_2$ and a fixed $y$, $\varphi_{S_1}(x,y)$ and $\varphi_{S_2}(x,y)$ can't hold simultaneously, the number of nodes $y$ that satisfies $\varphi_S(x,y) \wedge \varphi_{S,T}^y(y)$ is exactly the sum $\sum_{S \subseteq P_2} n_S$. Therefore, in order to express Equation (15), which means there exists at least $n$ nodes $y$ such that $\bigvee_{S \subseteq P_2} (\varphi_S(x,y) \wedge \varphi_{S,T}^y(y))$ holds, it suffices to enumerate all possible values for $\{n_S | S \subseteq P_2\}$ that satisfies $(\sum_{S \subseteq P_2} n_S) = n, n_S \in \mathbb{N}$. Formally, we can rewrite $\widehat{\varphi}_T(x)$ as follows:

$$\widehat{\varphi}_T(x) \equiv \bigvee_{(\sum_{S \subseteq P_2} n_S) = n} \left( \bigwedge_{S \subseteq P_2} \exists^{\geq n_S} y(\varphi_S(x,y) \wedge \varphi_{S,T}^y(y)) \right) \tag{16}$$

Note that $\exists^{\geq n_S} y(\varphi_S(x,y) \wedge \varphi_{S,T}^y(y))$ satisfies the grammar of $\mathcal{RSFOC}_2$, so $\widehat{\varphi}_T(x)$ can be rewritten as $\mathcal{RSFOC}_2$. Then, since $\varphi_T^x(x)$ can also be rewritten as $\mathcal{RSFOC}_2$ by induction, combining Equation (14) and Equation (15), $\varphi(x)$ is in $\mathcal{RSFOC}_2$. We finish the proof. $\square$

# B   Proof of Proposition 3

**Proposition 3.** $\mathcal{FOC}_2 \not\subseteq R^2\text{-GNN}$ and $R^2\text{-GNN} \not\subseteq \mathcal{FOC}_2$ on some universal graph class $\mathcal{G}_u$.

*Proof.* First, we prove $\mathcal{FOC}_2 \not\subseteq R^2\text{-GNN}$.

Consider the two graphs $G_1, G_2$ in Figure 1. $(G_1, a), (G_2, a)$ can be distinguished by the $\mathcal{FOC}_2$ formula $\varphi(x) := \exists^{\geq 1} y(p_1(x,y) \wedge p_2(x,y))$. However, we will prove that any $R^2\text{-GNN}$ can't distinguish any node in $G_1$ from any node in $G_2$.

Let's prove it by induction over the layer number $L$ of $R^2\text{-GNN}$. That's to say, we want to show that for any $L \geq 0$, $R^2\text{-GNN}$ with no more than $L$ layers can't distinguish any node of $G_1$ from that of $G_2$.

For the base case where $L = 0$, since each node feature vector is initialized by the unary predicate information, so the result trivially holds.

Assume any $R^2\text{-GNN}$ with no more than $L = m$ layers can't distinguish nodes of $G_1$ from nodes of $G_2$. Then we want to prove the result for $L = m + 1$.

For any $R^2\text{-GNN}$ model $\mathcal{A}$ with $m+1$ layers, let $\mathcal{A}'$ denote its first $m$ layers, we know outputs of $\mathcal{A}'$ on any node from $G_1$ or $G_2$ are the same, suppose the common output feature is $\mathbf{x}^{(m)}$.

Recall the updating rule of $R^2\text{-GNN}$ in Equation (2). We know the output of $\mathcal{A}$ on any node $v$ in $G_1$ or $G_2$ is defined as follows,

$$\mathbf{x}_v^{(m+1)} = C^{(m+1)} \left( \mathbf{x}_v^{(m)}, \left( A_1^{(m+1)}(\{\!\{\mathbf{x}_{u_1(v)}^{(m)}\}\!\}) \right), A_2^{(m+1)}(\{\!\{\mathbf{x}_{u_2(v)}^{(m)}\}\!\}) \right), R^{(m+1)}(\{\!\{\mathbf{x}_a^{(m)}, \mathbf{x}_b^{(m)}, \mathbf{x}_c^{(m)}, \mathbf{x}_d^{(m)}\}\!\}) \right)$$
$$\tag{17}$$

Here $C^{(m+1)}, A_1^{(m+1)}, A_2^{(m+1)}, R^{(m+1)}$ are parameters in the layer $m+1$ of $\mathcal{A}$, $u_1(v), u_2(v)$ is the only $r_1, r_2$-type neighbor of $v$, and $a, b, c, d$ are nodes from the corresponding graph $G_1$ or $G_2$. From Figure 1 we can see they are well defined.

By induction, since any node pairs from $G_1$ and $G_2$ can't be distinguished by $\mathcal{A}'$, we have $\mathbf{x}_v^{(m)}, \mathbf{x}_{u_1(v)}^{(m)}, \mathbf{x}_{u_2(v)}^{(m)}, \mathbf{x}_a^{(m)}, \mathbf{x}_b^{(m)}, \mathbf{x}_c^{(m)}, \mathbf{x}_d^{(m)}$ are all the same feature $\mathbf{x}^{(m)}$. Therefore, Equation (17) have the same expression for all nodes $v$ from $G_1$ and $G_2$, which implies any $\mathcal{A}$ with $m+1$ layers can't distinguish nodes from $G_1$ and $G_2$.

Next, we then prove $R^2\text{-GNNs} \not\subseteq \mathcal{FOC}_2$.

Assume we want to construct a classifier $c$ which classifies a node into true iff *the node has a larger number of $r_1$-type neighbors than that of $r_2$-type neighbors*.

First, we prove that we can construct an 0/1-GNN $\mathcal{A}$ to capture $c$. It only has one layer with parameters $C^{(1)}, A_1^{(1)}, A_2^{(1)}, R^{(1)}$, and feature dimension $d_0 = d_1 = 1$. We assume that each node has the same initial feature vector, i.e., $\mathbf{1}$. We set $A_{1,(1,1)}^{(1)} = 1, A_{2,(1,1)}^{(1)} = -1$, where $A_{1,(1,1)}^{(1)}$ denotes the only element in $A_1^{(1)}$ placed in the first row and first column (similar for $A_{2,(1,1)}^{(1)}$) and all other

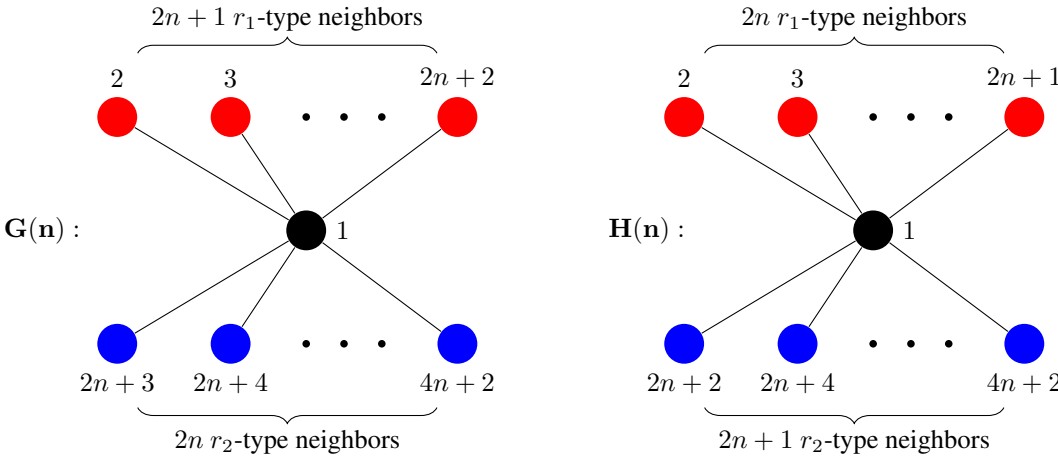

Figure 5: $G(n)$ and $H(n)$.

parameters $0$. It's easy to see that $\mathcal{A}$ is equivalent to our desired classifier $c$ on any graph since we have $\mathbf{x}_v^{(1)} = max(0, min(1, \sum_{u \in \mathcal{N}_{G,1}(v)} 1 - \sum_{u \in \mathcal{N}_{G,2}(v)} 1))$.

Next, we show $\mathcal{FOC}_2$ can't capture $c$ on $\mathcal{G}_s$. In order to show that, for any natural number $n$, we can construct two single-edge graphs $G(n), H(n)$ as follows:

$$V(G(n)) = V(H(n)) = \{1, 2 ...... 4n + 2\}$$
$$E(G(n)) = \{r_1(1,i)|\forall i \in [2, 2n+2]\} \cup \{r_2(1,i)|i \in [2n+3, 4n+2]\}$$
$$E(H(n)) = \{r_1(1,i)|\forall i \in [2, 2n+1]\} \cup \{r_2(1,i)|i \in [2n+2, 4n+2]\}$$

We prove the result by contradiction. Assume there is a $\mathcal{FOC}_2$ classifier $\varphi$ that captures the classifier $c$, then it has to classify $(G(n), 1)$ as *true* and $(H(n), 1)$ as *false* for all natural number $n$. However, in the following we will show that it's impossible, which proves the non-existence of such $\varphi$.

Suppose threshold numbers used on counting quantifiers of $\varphi$ don't exceed $m$, then we only need to prove that $\varphi$ can't distinguish $(G(m), 1), (H(m), 1)$, which contradicts our assumption.

For simplicity, we use $G, H$ to denote $G(m), H(m)$. In order to prove the above argument. First, we define a *node-classification* function $CLS(\cdot)$ as follows. It has $G$ or $H$ as subscript and a node of $G$ or $H$ as input.

1. $CLS_G(1) = CLS_H(1) = 1$. It means the function returns $1$ when the input is the *center* of $G$ or $H$.

2. $CLS_G(v_1) = CLS_H(v_2) = 2, \forall v_1 \in [2, 2m+2], \forall v_2 \in [2, 2m+1]$, which means the function returns $2$ when the input is a $r_1$-neighbor of *center*.

3. $CLS_G(v_1) = CLS_H(v_2) = 3, \forall v_1 \in [2m+3, 4m+2], \forall v_2 \in [2m+2, 4m+2]$, which means the function returns $3$ when the input is a $r_2$-neighbor of *center*.

**Claim 1**: Given any $u_1, v_1 \in V(G), u_2, v_2 \in V(H)$, if $(CLS_G(u_1), CLS_G(v_1)) = (CLS_H(u_2), CLS_H(v_2))$, then any $\mathcal{FOC}_2$ formula with threshold numbers no larger than $m$ can't distinguish $(u_1, v_1)$ and $(u_2, v_2)$.

This claim is enough for our result. We will prove that for any constant $d$ and any $\mathcal{FOC}_2$ formula $\phi$ with threshold numbers no larger than $m$ and quantifier depth $d$, $\phi$ can't distinguish $(u_1, v_1)$ and $(u_2, v_2)$ given that $(CLS_G(u_1), CLS_G(v_1)) = (CLS_H(u_2), CLS_H(v_2))$

The result trivially holds for the base case where $d = 0$. Now let's assume the result holds for $d \le k$, we can now prove the inductive case when $d = k + 1$.

Since $\wedge, \vee, \neg, r(x,y)$ trivially follows, we can only consider the case when $\phi(x,y)$ is in the form $\exists^{\ge N} y \phi'(x,y), N \le m$ or $\exists^{\ge N} x \phi'(x,y), N \le m$, where $\phi'(x,y)$ is a $\mathcal{FOC}_2$ formula with threshold

numbers no more than $m$ and quantifier depth no more than $k$. Since these two forms are symmetrical, without loss of generality, we only consider the case $\exists^{\geq N} y \phi'(x,y), N \leq m$.

Let $N_1$ denote the number of nodes $v_1' \in V(G)$ such that $(G, u_1, v_1') \models \phi'$ and $N_2$ denote the number of nodes $v_2' \in V(H)$ such that $(H, u_2, v_2') \models \phi'$. Let's compare values of $N_1$ and $N_2$. First, By induction, since we have $CLS_G(u_1) = CLS_H(u_2)$ from precondition, so for any $v_1' \in V(G), v_2' \in V(H)$, which satisfies $CLS_G(v_1') = CLS_H(v_2')$, $\phi'(x,y)$ can't distinguish $(u_1, v_1')$ and $(u_2, v_2')$. Second, isomorphism tells us $\phi'$ can't distinguish node pairs from the same graph if they share the same $CLS$ values. Combining these two facts, there has to be a subset $S \subseteq \{1,2,3\}$, such that $N_1 = \sum_{a \in S} N_G(a)$ and $N_2 = \sum_{a \in S} N_H(a)$, where $N_G(a)$ denotes the number of nodes $u$ on $G$ such that $CLS_G(u) = a$, ($N_H(a)$ is defined similarly).

It's easy to see that $N_G(1) = N_H(1) = 1$, and $N_G(a), N_H(a) > m$ for $a \in \{2,3\}$. Therefore, at least one of $N_1 = N_2$ and $m < min\{N_1, N_2\}$ holds. In neither case $\exists^{\geq N} y \phi'(x,y), N \leq m$ can distuigush $(u_1, v_1)$ and $(u_2, v_2)$. $\qquad \square$

Note that in the above proof our graph class $\{G(n), H(n) | n \in \mathbb{N}\}$ is actually a *simple* graph class, so we can actually get the following stronger argument.

**Corollary 24.1.** $R^2\text{-}GNN \nsubseteq \mathcal{FOC}_2$ on some simple graph class.

## C  Proof of Theorem 4

**Theorem 4.** $\mathcal{FOC}_2 \subseteq R^2\text{-}GNN$ on any simple graph class, and $\mathcal{FOC}_2 \subsetneq R^2\text{-}GNN$ on some simple graph class.

*Proof.* We just need to show $\mathcal{FOC}_2 \subseteq R^2$-GNN on any simple graph class, and the second part can be just concluded from Corollary 24.1. By Lemma 22, $\mathcal{FOC}_2 = \mathcal{RSFOC}_2$, so it suffices to show $\mathcal{RSFOC}_2 \subseteq 0/1$-GNN. By Lemma 19, $0/1$-GNN is closed under $\wedge, \vee, \neg$, so we can only focus on formulas in $\mathcal{RSFOC}_2$ of form $\varphi(x) = \exists^{\geq n} y(\varphi_S(x,y) \wedge \varphi'(y)), S \subseteq P_2$. If we can construct an equivalent $0/1$-GNN $\mathcal{A}$ for all formulas of above form, then we can capture all formulas in $\mathcal{RSFOC}_2$ since other generating rules $\wedge, \vee, \neg$ is closed under $0/1$-GNN. In particular, for the setting of *single-edge* graph class, $\varphi$ is meaningful only when $|S| \leq 1$. That's because $|S| > 2$ implies that $\varphi$ is just the trivial $\bot$ in any *single-edge* graph class $\mathcal{G}_s$.

Do induction over quantifier depth $k$ of $\varphi(x)$. In the base case where $k = 0$, the result trivially holds since in this situation, the only possible formulas that needs to consider are unary predicates $A(x)$, where $A \in P_1$, which can be captured by the initial one-hot feature. Next, assume our result holds for all formulas with quantifier depth $k$ no more than $m$, it suffices to prove the result when quantifier depth of $\varphi(x) = \exists^{\geq n} y(\varphi_S(x,y) \wedge \varphi'(y))$ is $m + 1$. It follows that quantifier depth of $\varphi'(y)$ is no more than $m$.

By induction, there is a $0/1$-GNN model $\mathcal{A}'$ such that $\mathcal{A}' = \varphi'$ on single-edge graph class. To construct $\mathcal{A}$, we only need to append another layer on $\mathcal{A}'$. This layer $L + 1$ has dimension 1, whose parameters $C^{(L+1)}, (A_j^{(L+1)})_{j=1}^K, R^{(L+1)}, b^{(L+1)}$ are set as follows:

1. When $|S| = 1$: Suppose $S = \{j\}$, set $A_{j,(1,1)}^{L+1} = 1, b^{L+1} = 1 - n$, where $A_{j,(1,1)}^{L+1}$ denotes the element on the first row and first column of matrix $A_j^{(L+1)}$. Other parameters in this layer are 0. This construction represents $\mathbf{x}_v^{(L+1)} = max(0, min((\sum_{u \in \mathcal{N}_{G,j}(v)} \mathbf{x}_u^{(L)}) - (n-1), 1))$. Since $\mathbf{x}_u^{(L)}$ is classification result outputted by $\mathcal{A}'$ which is equivalent to $\varphi'$, $\sum_{u \in \mathcal{N}_{G,j}(v)} \mathbf{x}_u^{(L)}$ counts the number of $j$-type neighbor $u$ of $v$ that satisfies $\varphi'(u)$. Therefore $\mathbf{x}_v^{(L+1)} = 1$ if and only if there exists at least $n$ $j$-type neighbors satisfying the condition $\varphi'$, which is exactly what $\varphi(x)$ means.

2. When $|S| = 0$: Let $K = |P_2|$, for all $j \in [K]$, set $A_{j,(1,1)}^{L+1} = -1, R_{1,1}^{(L+1)} = 1, b^{L+1} = 1 - n$ and all other parameters 0. This construction represents $\mathbf{x}_v^{(L+1)} = max(0, min((\sum_{u \in V(G)} \mathbf{x}_u^{(L)}) - (\sum_{j=1}^K \sum_{u \in \mathcal{N}_{G,j}(v)} \mathbf{x}_u^{(L)}) - (n-1), 1))$. Since we only

consider single-edge graph, $(\sum_{u\in V(G)} \mathbf{x}_u^{(L)}) - (\sum_{j=1}^{K}\sum_{u\in\mathcal{N}_{G,j}(v)}\mathbf{x}_u^{(L)})$ exactly counts the number of nodes $u$ that satisfies $\varphi'(y)$ and doesn't have any relation with $v$. It's easy to see that $\mathbf{x}_v^{(L+1)} = 1$ iff there exists at least $n$ such nodes $u$, which is exactly what $\varphi(x)$ means.

Hence, we finish the proof for Theorem 4 – for each $\mathcal{FOC}_2$ formula over the single-edge graph class, we can construct an $\mathrm{R}^2$-GNN to capture it.

$\square$

## D  Proof of Theorem 5 and Theorem 6

**Theorem 5.** $R^2$-GNN $\subseteq \mathcal{FOC}_2$ on any bounded graph class, and $R^2$-GNN $\subsetneq \mathcal{FOC}_2$ on some bounded graph class.

**Theorem 6.** For any bounded graph class $\mathcal{G}_b$. Suppose any $G \in \mathcal{G}_b$ has no more than $N$ nodes, and $\mathcal{G}_b$ has unary predicate set $P_1$ and relation (binary predicate) set $P_2$. Let $m_1 := |P_1|, m_2 := |P_2|$, then for any node classifier $c$, suppose $c$ can be represented as an $R^2$-GNN with depth (layer number) $L$, then by Theorem 5 there is a $\mathcal{FOC}_2$ classifier $\varphi$ equivalent to $c$ over $\mathcal{G}_b$. Moreover, the followings hold:

1. The quantifier depth of $\varphi$ is no more than $L$.

2. The size of $\varphi$ (quantified by the number of nodes of $\varphi$'s parse tree) is no more than $2^{2f(L)}$, where $f(L) := 2^{2^{2(N+1)\bar{f}(L-1)}}, f(0) = O(2^{2^{2(m_1+m_2)}})$.

For Theorem 5, we just need to show $\mathrm{R}^2$-GNN $\subseteq \mathcal{FOC}_2$ on any bounded graph class. The second part can then be shown by the fact that the graph class $\{G_1, G_2\}$ in Figure 1 is a bounded graph class but $\mathcal{FOC}_2 \not\subseteq \mathrm{R}^2$-GNN still holds. In the following proof, we also show how to get the complexity upper bound claimed in Theorem 6. If we want to prove $\mathrm{R}^2$-GNN $\subseteq \mathcal{FOC}_2$, it suffices to show that for any $\mathrm{R}^2$-GNN $\mathcal{A}$, there exists an equivalent $\mathcal{FOC}_2$ formula $\varphi$ on any bounded graph class $\mathcal{G}_b$. It implies that for two graphs $G_1, G_2$ and their nodes $a, b$, if they are classified differently by $\mathcal{A}$, there exists some $\mathcal{FOC}_2$ formula $\varphi$ that can distinguish them. Conversly, if $a, b$ can't be distinguished by any $\mathcal{FOC}_2$ formula, then they can't be distinguished by any $\mathrm{R}^2$-GNN as well.

**Definition 25.** For a set of classifiers $\Psi = \{\psi_1 \ldots \psi_m\}$, a $\Psi$-truth-table $T$ is a 0/1 string of length $m$. $T$ can be seen as a classifier, which classifies a node $v$ to be true if and only if for any $1 \le i \le m$, the classification result of $\psi_i$ on $v$ equals to $T_i$, where $T_i$ denotes the $i$-th bit of string $T$. We define $\mathcal{T}(\Psi) := \{0,1\}^m$ as the set of all $\Psi$-truth-tables. We have that for any graph $G$ and its node $v$, $v$ satisfies exactly one truth-table $T$.

**Proposition 26.** Let $\mathcal{FOC}_2(n)$ denote the set of formulas of $\mathcal{FOC}_2$ with quantifier depth no more than $n$. For any bounded graph class $\mathcal{G}_b$ and $n$, only finitely many intrinsically different node classifiers on $\mathcal{G}_b$ can be represented by $\mathcal{FOC}_2(n)$. Furthermore, define $N, m_1, m_2$ as in Theorem 6, the number of intrinsically different $\mathcal{FOC}_2(n)$ node classifiers on $\mathcal{G}_b$ and their parse tree sizes are all upper bounded by $f(n)$ as defined in Theorem 6.

*Proof.* Suppose all graphs in $\mathcal{G}_b$ have no more than $N$ constants, then for any natural number $m > N$, formulas of form $\exists^{\ge m} y(\varphi(x,y))$ are always false. Therefore, it's sufficient only to consider $\mathcal{FOC}_2$ logical classifiers with threshold numbers no more than $N$ on $\mathcal{G}_b$.

There are only $m_1 + m_2$ predicates, and each boolean combination of unary predicates using $\wedge, \vee, \neg$ can be rewritten in the form of Disjunctive Normal Form (DNF) (Davey and Priestley [2002]). So there are only at most $f(0) = 2^{2^{2(m_1+m_2)}}$ intrinsically different formulas in $\mathcal{FOC}_2$ with quantifier depth 0. Note that $2(m_1 + m_2)$ is the number of terms, $2^{2(m_1+m_2)}$ is the number of different truth-table conjunctions on these terms, and $2^{2^{2(m_1+m_2)}}$ is the number of different DNFs on these conjunctions. Each DNF has parse tree of size at most $1 + 2^{2(m_1+m_2)}(1 + 2m_1 + 2m_2) \le 1000 \cdot 2^{2^{2(m_1+m_2)}}$. Therefore, define $f(0) = 1000 \cdot 2^{2^{2(m_1+m_2)}} = O(2^{2^{2(m_1+m_2)}})$, we know the number of different $\mathcal{FOC}_2$ formulas with quantifier depth 0 and parse tree size of these formulas can both be upper bounded by $f(0)$.

By induction, suppose there are only $f(k)$ intrinsically different $\mathcal{FOC}_2(k)$ formulas on $\mathcal{G}_b$. and each meaningful $\mathcal{FOC}_2(k+1)$ formula is generated by the following grammar

$$\varphi_1 \wedge \varphi_2, \varphi_1 \vee \varphi_2, \neg\varphi_2, \exists^{\geq m}y(\varphi'(x,y)), m \leq N \tag{18}$$

where $\varphi_1, \varphi_2$ are $\mathcal{FOC}_2(k+1)$ formulas and $\varphi'$ is $\mathcal{FOC}_2(k)$ formulas.

Given that only the rule $\exists^{\geq m}y(\varphi'(x,y))$ can increase the quantifier depth from $k$ to $k + 1$, $m \leq N$, and there are only $f(k)$ intrinsically different $\varphi'(x,y) \in \mathcal{FOC}_2(k)$ on $\mathcal{G}_b$ by induction. Therefore, there are only $(2N + 2)f(k)$ intrinsically different $\mathcal{FOC}_2(k + 1)$ formulas of form $\exists^{\geq m}y(\varphi'(x,y)), \exists^{\geq m}x(\varphi'(x,y))$ or in $\mathcal{FOC}_2(k)$ on $\mathcal{G}_b$. Moreover, their boolean combination using $\wedge, \vee, \neg$ can be always rewritten in the DNF form, So there are also finitely many intrinsically different $\mathcal{FOC}_2(k + 1)$ logical classifiers on $\mathcal{G}_b$. Similarly, we can bound the number of different DNF by $f(k + 1) = 2^{2^{2(N+1)f(k)}}$, where $2(N + 1)f(k)$ is the number of "building blocks" which are sub-formulas with smaller quantifier depth or outermost symbol $\exists$, $2^{2(N+1)f(k)}$ is the number of different conjunctions on these building blocks, and $f(k + 1) = 2^{2^{2(N+1)f(k)}}$ is the number of different DNFs on these conjunctions. Parse tree size of each of these DNFs is at most $1 + 2^{2(N+1)f(k)}(1 + 2(N + 1)f(k)(1 + f(k))) \leq 2^{2^{2(N+1)f(k)}} = f(k + 1)$. The LHS is from the inductive assumption that each $\mathcal{FOC}_2(k)$ formula has a equivalent representation within $f(k)$ parse tree size. The inequality is because we know $f(k) \geq 1000$. Thus, we can upper bound the number of intrinsically different $\mathcal{FOC}_2(k + 1)$ formulas on $\mathcal{G}_b$ and their parse tree size both by $f(k + 1)$. $\quad\square$

**Lemma 27.** *For any two pairs $(G_1, v_1)$ and $(G_2, v_2)$, where $G_1$ and $G_2$ are two* bounded *graphs from $\mathcal{G}_b$ and $v_1$ and $v_2$ are two nodes in $G_1$ and $G_2$, respectively. If all logical classifiers in $\mathcal{FOC}_2(L)$ can't distinguish $v_1, v_2$, then any R$^2$-GNN with layer no more than $L$ can't distinguish them as well.*

*Proof.* By one-hot feature initialization function of R$^2$-GNN, $\mathcal{FOC}_2(0)$ can distinguish all different one-hot intial features, so the lemma trivially holds for the base case ($L = 0$).

For the inductive step, we suppose Lemma 27 holds for all $L \leq k$, then we can assume $v_1, v_2$ can't be distinguished by $\mathcal{FOC}_2(k + 1)$. Let $N = k + 1$

$G_1$ and $G_2$ are *bounded* graphs from $\mathcal{G}_b$, so $\mathcal{FOC}_2(N)$ has finitely many intrinsically different classifiers according to Proposition 26. Let $\mathcal{TT}_N(v)$ denote the $\mathcal{FOC}_2(N)$-truth-table satisfied by $v$. According to Definition 25, we know that for any $T \in \mathcal{T}(\mathcal{FOC}_2(N))$, there exists a $\mathcal{FOC}_2(N)$ classifier $\varphi_T$ such that for any node $v$ on $G_i$, where $i \in 1, 2$, $\mathcal{TT}_N(v) = T \Leftrightarrow (G_i, v) \models \varphi_T$.

Assume there is an R$^2$-GNN $\mathcal{A}$ that distinguish $v_1, v_2$ with layer $L = k + 1$. Let $\widehat{\mathcal{A}}$ denote its first $k$ layers. By update rule of R$^2$-GNN illustrated in Equation 2, output of $\mathcal{A}$ on node $v$ of graph $G$, $\mathbf{x}_v^{(k+1)}$ only dependent on the following three things:

- output of $\widehat{\mathcal{A}}$ on $v$, $\mathbf{x}_v^{(k)}$

- multiset of outputs of $\widehat{\mathcal{A}}$ on $r$-type neighbors of $v$ for each $r \in P_2$, $\{\mathbf{x}_u^{(k)} | u \in \mathcal{N}_{G,r}(v)\}$

- multiset of outputs of $\widehat{\mathcal{A}}$ on all nodes in the graph, $\{\mathbf{x}_u^{(k)} | u \in \mathcal{N}_{G,r}(v)\}$

By induction, since $v_1, v_2$ can't be distinguished by $\mathcal{FOC}_2(k)$, they has same feature outputted by $\widehat{\mathcal{A}}$. Then there are two remaining possibilities.

- $\{\!\{\mathcal{TT}_k(u) | u \in \mathcal{N}_{G_1,r}(v_1)\}\!\} \neq \{\!\{\mathcal{TT}_k(u) | u \in \mathcal{N}_{G_2,r}(v_2)\}\!\}$ for some binary predicate $r$. Therefore, there exists a $\mathcal{FOC}_2(k)$-truth-table $T$, such that $v_1, v_2$ have differently many $r$-type neighbors that satisfies $\varphi_T$. Without loss of generality, suppose $v_1, v_2$ have $n_1, n_2 (n_1 < n_2)$ such neighbors respectively. we can write a $\mathcal{FOC}_2(k + 1)$ formula $\exists^{\geq n_2}y(r(x,y) \wedge \varphi_T(y))$ that distinguishes $v_1$ and $v_2$, which contradicts the precondition that they can't be distinguished by $\mathcal{FOC}_2(k + 1)$ classifiers.

- $\{\!\{\mathcal{TT}_k(u) | u \in V(G_1)\}\!\} \neq \{\!\{\mathcal{TT}_k(u) | u \in V(G_2)\}\!\}$. Therefore, there exists a $\mathcal{FOC}_2(k)$-truth-table $T$, such that $G_1, G_2$ have differently many nodes that satisfies $\varphi_T$. Without loss of generality, suppose $G_1, G_2$ have $n_1, n_2 (n_1 < n_2)$ such nodes respectively. we can write

a $\mathcal{FOC}_2(k+1)$ formula $\exists^{\geq n_2} y \varphi_T(y)$ that distinguishes $v_1$ and $v_2$, which contradicts the precondition that they can't be distinguished by $\mathcal{FOC}_2(k+1)$ classifiers.

Since all possibilities contradicts the precondition that $v_1, v_2$ can't be distinguished by $\mathcal{FOC}_2(k+1)$, such an $\mathcal{A}$ that distinguishes $v_1, v_2$ doesn't exist. □

We can now gather all of these to prove Theorem 5 and Theorem 6.

*Proof.* For any $R^2$-GNN $\mathcal{A}$, suppose it has $L$ layers. For any graph $G \in \mathcal{G}_b$ and its node $v$, let $\mathcal{TT}_L(v)$ denote the $\mathcal{FOC}_2(L)$-truth-table satisfied by $v$. For any $T \in \mathcal{T}(\mathcal{FOC}_2(L))$, since $\mathcal{G}_b$ is a bounded graph class, using Proposition 26, there exists a $\mathcal{FOC}_2(L)$ classifier $\varphi_T$ such that for any node $v$ in graph $G \in \mathcal{G}_b$, $\mathcal{TT}_L(v) = T \Leftrightarrow (G, v) \models \varphi_T$. Moreover, by Proposition 26, since $T$ is a truth table on at most $f(L)$ formulas, $\varphi_T$ can be written as a conjunction over $f(L)$ literals, which means $\varphi_T$ has parse tree size at most $1 + f(L)^2$ since by Proposition 26, every formula in $\mathcal{FOC}_2(L)$ is equivalent to some $\mathcal{FOC}_2$ formula with parsee tree size at most $f(L)$.

By Lemma 27, If two nodes $v_1, v_2$ have same $\mathcal{FOC}_2(L)$-truth-table ($\mathcal{TT}_L(v_1) = \mathcal{TT}_L(v_2)$), they can't be distinguished by $\mathcal{A}$. Let $S$ denote the subset of $\mathcal{T}(\mathcal{FOC}_2(L))$ that satisfies $\mathcal{A}$. By Proposition 26 and Definition 25, $\Phi := \{\varphi_T | T \in S\}$ is a finite set with $|\Phi| \leq 2^{f(L)}$, then disjunction of formulas in $\Phi$, $(\bigvee_{T \in S} \varphi_T)$ is a $\mathcal{FOC}_2$ classifier that equals to $\mathcal{A}$ under bounded graph class $\mathcal{G}_b$. Furthermore, by the above upper bound of parse tree size of any $\varphi_T$, $(\bigvee_{T \in S} \varphi_T)$ has parse tree size no more than $1 + 2^{f(L)}(1 + f(L)^2) \leq 2^{2f(L)}$, where the inequality is from $f(L) \geq 1000$. We complete the proof. □

# E    Proof of Theorem 9

**Theorem 9.** $R^2$-*GNN* $\subseteq R^2$-*GNN* $\circ F$ *on any* universal *graph class* $\mathcal{G}_u$.

*Proof.* Assume that we have a predicate set $P = P_1 \cup P_2$, $K = |P_2|$ and let $P' = P \cup \{primal, aux1, aux2\}$ denote the predicate set after transformation $F$. For any $R^2$-GNN $\mathcal{A}$ under $P$, we want to construct another $R^2$-GNN $\mathcal{A}'$ under $P'$, such that for any graph $G$ under $P$ and its node $v$, $v$ has the same feature outputted by $\mathcal{A}(G, v)$ and $\mathcal{A}'(F(G), v)$. Let $L$ denote the layer number of $\mathcal{A}$.

We prove this theorem by induction over the number of layers $L$. In the base ($L = 0$), our result trivially holds since the one-hot initialization over $P'$ contains all unary predicate information in $P$. Now suppose the result holds for $L \leq k$, so it suffices to prove it when $L = k + 1$.

For the transformed graph $F(G)$, *primal*(v) is *true* if and only if $v$ is the node in the original graph $G$. Without loss of generality, if we use one-hot feature initialization on $P'$, we can always keep an additional dimension in the node feature vector $\mathbf{x}_v$ to show whether *primal*(v) is *true*, its value is always $0/1$, in the proof below when we use $\mathbf{x}$ to denote the feature vectors, we omit this special dimension for simplicity. But keep in mind that this dimension always keeps so we can distinguish original nodes and added nodes.

Recall that an $R^2$-GNN is defined by $\{C^{(i)}, (A_j^{(i)})_{i=1}^K, R^{(i)}\}_{i=1}^L$. By induction, let $\widehat{\mathcal{A}}$ denote the first $k$ layers of $\mathcal{A}$, and let $\widehat{\mathcal{A}'}$ denote the $R^2$-GNN equivalent with $\widehat{\mathcal{A}}$ on $F$ transformation such that $\widehat{\mathcal{A}} = \widehat{\mathcal{A}'} \circ F$. We will append three layers to $\widehat{\mathcal{A}'}$ to construct $\mathcal{A}'$ that is equivalent to $\mathcal{A}$. Without loss of generality, we can assume all layers in $\mathcal{A}$ have same dimension length $d$. Suppose $L'$ is the layer number of $\widehat{\mathcal{A}'}$, so we will append layer $L' + 1, L' + 2, L' + 3$. for all $l \in \{L' + 1, L' + 2, L' + 3\}$, let $\{C^{a,(l)}, C^{p,(l)}, (A_j^{*,(l)})_{j=1}^K, A_{aux1}^{*,(l)}, A_{aux2}^{*,(l)}, R^{*,(l)}\}$ denote the parameters in $l$-th layer of $\mathcal{A}$. Here, $A_{aux1}^{*,(l)}, A_{aux2}^{*,(l)}$ denotes the aggregation function corresponding to two new predicates *aux1, aux2*, added in transformation $F$, and $C^{p,(l)}, C^{a,(l)}$ are different combination function that used for primal nodes and non-primal nodes. Note that with the help of the special dimension mentioned above, we can distinguish primal nodes and non-primal nodes. Therefore, It's safe to use different combination functions for these two kinds of nodes. Note that here since we add two predicates *aux1, aux2*, the input for combination function should be in the form $C^p(\mathbf{x}_0, (\mathbf{x}_j)_{j=1}^K, \mathbf{x}_{aux1}, \mathbf{x}_{aux2}, \mathbf{x}_g)$ where $\mathbf{x}_0$ is the feature vector of the former layer, and $\mathbf{x}_j, 1 \leq j \leq K$ denote the output of aggregation function

$A_j^{*,(l)}$, $\mathbf{x}_{aux1}$,$\mathbf{x}_{aux2}$ denote the output of aggregation function $A_{aux1}^{*,(l)}$,$A_{aux2}^{*,(l)}$, and $\mathbf{x}_g$ denotes the feature outputted by global readout function $R^{*,(l)}$. For aggregation function and global readout function, their inputs are denoted by $\mathbf{X}$, meaning a multiset of feature vector. Note that all aggregation functions and readout functions won't change the feature dimension, only combination functions $C^{p,(l)}$,$C^{a,(l)}$ will transform $d_{l-1}$ dimension features to $d_l$ dimension features.

1). layer $L'+1$: input dimension is $d$, output dimension is $d' = Kd$. For feature vector $\mathbf{x}$ with length $d'$, let $\mathbf{x}^{(i)}, i \in \{1, \ldots, K\}$ denote its $i$-th slice in dimension $[(i-1)d+1, id]$. Let $[\mathbf{x}_1, \ldots, \mathbf{x}_m]$ denote concatenation of $\mathbf{x}_1, \ldots, \mathbf{x}_m$, and let $[\mathbf{x}]^n$ denote concatenation of $n$ copies of $\mathbf{x}$, $\mathbf{0}^n$ denote zero vectors of length $n$. parameters for this layer are defined below:

$$C^{p,(L'+1)}(\mathbf{x}_0,(\mathbf{x}_j)_{j=1}^K,\mathbf{x}_{aux1},\mathbf{x}_{aux2},\mathbf{x}_g) = [\mathbf{x}_0,\mathbf{0}^{d'-d}] \tag{19}$$

$$C^{a,(L'+1)}(\mathbf{x}_0,(\mathbf{x}_j)_{j=1}^K,\mathbf{x}_{aux1},\mathbf{x}_{aux2},\mathbf{x}_g) = [\mathbf{x}_{aux1}]^K \tag{20}$$

$$A_{aux1}^{*,(L'+1)}(\mathbf{X}) = \sum_{\mathbf{x}\in\mathbf{X}} \mathbf{x} \tag{21}$$

Other parameters in this layer are set to functions that always output zero-vector.

We can see here that the layer $L'+1$ do the following thing:

For all primal nodes $a$ and its non-primal neighbor $e_{ab}$, pass concatenation of $K$ copies of $\mathbf{x}_a$ to $\mathbf{x}_{e_{ab}}$, and remains the feature of primal nodes unchanged.

2). layer $L'+2$, also has dimension $d' = Kd$, has following parameters.

$$C^{p,(L'+2)}(\mathbf{x}_0,(\mathbf{x}_j)_{j=1}^K,\mathbf{x}_{aux1},\mathbf{x}_{aux2},\mathbf{x}_g) = \mathbf{x}_0 \tag{22}$$

$$C^{a,(L'+2)}(\mathbf{x}_0,(\mathbf{x}_j)_{j=1}^K,\mathbf{x}_{aux1},\mathbf{x}_{aux2},\mathbf{x}_g) = \sum_{j=1}^K \mathbf{x}_j \tag{23}$$

$$\forall j \in [1,K], A_j^{*,(L'+2)}(\mathbf{X}) = [\mathbf{0}^{(j-1)d}, \sum_{\mathbf{x}\in\mathbf{X}} \mathbf{x}^{(j)},\mathbf{0}^{(K-j)d}] \tag{24}$$

All other parameters in this layer are set to function that always outputs zero vectors. This layer do the following thing:

For all primal nodes, keep the feature unchanged, for all added node pair $e_{ab}$,$e_{ba}$. Switch their feature, but for all $r_i \in P_2$, if there is no $r_i$ relation between $a$,$b$, the $i$-th slice of $\mathbf{x}_{e_{ab}}$ and $\mathbf{x}_{e_{ba}}$ will be set to $\mathbf{0}$.

3). layer $L'+3$, has dimension $d$, and following parameters.

$$C^{p,(L'+3)}(\mathbf{x}_0,(\mathbf{x}_j)_{j=1}^K,\mathbf{x}_{aux1},\mathbf{x}_{aux2},\mathbf{x}_g) = C^{(L)}(\mathbf{x}_0^{(1)},(\mathbf{x}_{aux1}^{(j)})_{j=1}^K,\mathbf{x}_g^{(1)}) \tag{25}$$

$$R^{*,(L'+3)}(\mathbf{X}) = [R^{(L)}(\{\!\{\mathbf{x}_v^{(1)}|\mathbf{x}_v \in \mathbf{X},\textbf{primal}(v)\}\!\}),\mathbf{0}^{d'-d}] \tag{26}$$

$$A_{aux1}^{*,(L'+3)}(\mathbf{X}) = [A_1^{(L)}(\{\!\{\mathbf{x}^{(1)}|\mathbf{x}\in\mathbf{X}\}\!\})......A_K^{(L)}(\{\!\{\mathbf{x}^{(K)}|\mathbf{x}\in\mathbf{X}\}\!\})] \tag{27}$$

Note that $C^{(L)}$,$A_j^{(L)}$,$R^{(L)}$ are all parameters in the last layer of $\mathcal{A}$ mentioned previously. All other parameters in this layer are set to functions that always output zero vectors. We can see that this layer simulates the work of last layer of $\mathcal{A}$ as follows:

- For all $1 \leq j \leq K$, use the $j$-th slice of feature vector $\mathbf{x}^{(j)}$ to simulate $A_j^{(L)}$ and store results of aggregation function $A_j^{(L)}$ on this slice.

- Global readout trivially emulates what $R^{(L)}$ does, but only reads features for primal nodes. It can be done since we always have a special dimension in feature to say whether it's a primal node.

- We just simulate what $C^{(L)}$ does on primal nodes. For $1 \leq j \leq K$ The type $r_j$ aggregation result (output of $A_j^{(L)}$) used for input of $C^{(L)}$ is exactly $j$-th slice of return value of $A_{aux1}^{*,(L'+3)}$.

By construction above, $\mathcal{A}'$ is a desired model that have the same output as $\mathcal{A}$.

$\square$

# F  Proof of Theorem 10

**Theorem 10.** $\mathcal{FOC}_2 \subseteq R^2\text{-}GNN \circ F$ *on any* universal *graph class* $\mathcal{G}_u$.

*Proof.* For any $\mathcal{FOC}_2$ classifier $\varphi$ under predicate set $P$, we want to construct a 0/1-GNN $\mathcal{A}$ on $P' = P \cup \{primal, aux1, aux2\}$ equivalent to $\varphi$ with *graph transformation F*.

Recall that $\mathcal{FOC}_2 = \mathcal{RSFOC}_2$ shown in Lemma 22 and 0/1-GNNs $\subseteq$ $R^2$-GNNs, it suffices to prove that 0/1-GNN$\circ F$ capture $\mathcal{RSFOC}_2$. By Lemma 19, since $\wedge, \vee, \neg$ are closed under 0/1-GNN it suffices to show that when $\varphi$ is in the form $\exists^{\geq n}\big(\varphi_S(x,y) \wedge \varphi'(y)\big)$,$S \subseteq P_2$, we can capture it.

We prove by induction over quantifier depth $m$ of $\varphi$. Since 0-depth formulas are only about unary predicate that can be extracted from one-hot initial feature, our theorem trivially holds for $m = 0$. Now, we assume it also holds for $m \leq k$, it suffices to prove the case when $m = k + 1$. Then there are two possibilities:

1. When $S \neq \emptyset$:

Consider the following logical classifier under $P'$:

$$\widehat{\varphi}_S(x) := \Big(\bigwedge_{r \in S} \exists x r(x,y)\Big) \wedge \Big(\bigwedge_{r \notin S} \neg \exists x r(x,y)\Big) \tag{28}$$

$\widehat{\varphi}_S(x)$ restricts that for any $r \in P'$, $x$ has $r$-type neighbor if and only if $r \in S$. Review the definition of transformation $F$, we know that for any added node $e_{ab}$, $(F(G), e_{ab}) \models \widehat{\varphi}_S$ if and only if $(G,a,b) \models \varphi_S(a,b)$, where $\varphi_S(x,y)$ is the *relation-specification* formula defined in Definition 21 That is to say for any $r_i, 1 \leq i \leq K$, there is relation $r_i$ between $a,b$ if and only if $i \in S$.

Now consider the following formula:

$$\widehat{\varphi} := \exists^{\geq n} y \Big( aux1(x,y) \wedge \widehat{\varphi}_S(y) \wedge \Big( \exists x \big( aux2(x,y) \wedge (\exists y(aux1(x,y) \wedge \varphi'(y))) \big) \Big) \Big) \tag{29}$$

For any graph $G$ and its node $v$, it's easy to see that $(G,v) \models \varphi \Leftrightarrow (F(G),v) \models \widehat{\varphi}$. Therefore we only need to capture $\widehat{\varphi}$ by 0/1-GNN on every primal node of transformed graphs. By induction, since quantifier depth of $\varphi'(y)$ is no more than $k$, we know $\varphi'(y)$ is in 0/1-GNN. $\widehat{\varphi}$ is generated from $\varphi'(y)$ using rules $\wedge$ and $\exists y\big(r(x,y) \wedge \varphi'(y)\big)$. By Lemma 19, $\wedge$ is closed under 0/1-GNN. For $\exists y\big(r(x,y) \wedge \varphi'(y)\big)$, we find that the construction needed is the same as construction for single-element $S$ on single-edge graph class $\mathcal{G}_s$ used in Theorem 4. Therefore, since we can manage these two rules, we can also finish the construction for $\widehat{\varphi}$, which is equivalent to $\varphi$ on primal nodes of transformed graph.

2. When $S = \emptyset$

First, consider the following two logical classifiers:

$$\bar{\varphi}(x) := \Big( primal(x) \wedge \varphi'(x) \Big) \tag{30}$$

$\bar{\varphi}$ says a node is primal, and satisfies $\varphi'(x)$. Since $\varphi'(x)$ has quantifier depth no more than $k$, and $\wedge$ is closed under 0/1-GNN. There is a 0/1-GNN $\mathcal{A}_1$ equivalent to $\bar{\varphi}$ on transformed graph. Then, consider the following formula.

$$\tilde{\varphi}(x) := \exists y \big( aux2(x,y) \wedge (\exists x, aux11(x,y) \wedge \varphi'(x)) \big) \tag{31}$$

$\tilde{\varphi}(x)$ evaluates on added nodes $e_{ab}$ on transformed graph, $e_{ab}$ satisfies it iff $b$ satisfies $\varphi'$

Now for a graph $G$ and its node $v$, define $n_1$ as the number of nodes on $F(G)$ that satisfies $\bar{\varphi}$, and define $n_2$ as the number of *aux1*-type neighbors of $v$ on $F(G)$ that satisfies $\tilde{\varphi}$. Since $\varphi(x) = \exists^{\geq n} y(\varphi_\emptyset(x,y) \wedge \varphi'(y))$ It's easy to see that $(G,v) \models \varphi$ if and only if $n_1 - n_2 \geq n$.

Formally speaking, for a node set $S$, let $|S|$ denote number of nodes in $S$, we define the following classifier $c$ such that for any graph $G$ and its node $a$, $c(F(G),a) = 1 \Leftrightarrow (G,a) \models \varphi$

$$c(F(G),a) = 1 \Leftrightarrow |\{v|v \in V(F(G)), (F(G),v) \models \bar{\varphi}\}| - |\{v|v \in \mathcal{N}_{F(G),auxl1}(v), (F(G),v) \models \tilde{\varphi}\}| \geq n \tag{32}$$

So how to construct a model $\mathcal{A}$ to capture classifier $c$? First, by induction $\bar{\varphi}, \tilde{\varphi}$ are all formulas with quantifier depth no more than $k$ so by previous argument there are 0/1-GNN models $\bar{\mathcal{A}}, \tilde{\mathcal{A}}$ that capture them respectively. Then we can use feature concatenation technic introduced in Equation (6) to construct a model $\widehat{\mathcal{A}}$ based on $\bar{\mathcal{A}}, \tilde{\mathcal{A}}$, such that $\widehat{\mathcal{A}}$ has two-dimensional output, whose first and second dimensions have the same output as $\bar{\mathcal{A}}, \tilde{\mathcal{A}}$ respectively.

Then, suppose $\widehat{\mathcal{A}}$ has $L$ layers, The only thing we need to do is to append a new layer $L + 1$ to $\widehat{\mathcal{A}}$, it has output dimension 1. parameters of it are $\{C^{(L+1)}, (A_j^{(L+1)})_{j=1}^K, A_{aux1}^{(L+1)}, A_{aux2}^{(L+1)}, R^{(L+1)}\}$ as defined in Equation (5). The parameter settings are as follows:

$\mathbf{R}_{1,1}^{(L+1)} = 1, \mathbf{A}_{aux1,(1,2)}^{(L+1)} = -1, \mathbf{b}_1^{(L+1)} = 1 - n$. Other parameters are set to 0, where $\mathbf{A}_{aux1,(1,2)}^{(L+1)}$ denotes the value in the first row and second column of $\mathbf{A}_{aux1}^{(L+1)}$.

In this construction, we have

$\mathbf{x}_v^{(L+1)} = max(0, min(1, \sum_{u \in V(F(G))} \mathbf{x}_{u,1}^{(L)} - \sum_{u \in \mathcal{N}_{F(G),aux1}(v)} \mathbf{x}_{u,2}^{(L)} - (n-1)))$, which has exactly the same output as classifier $c$ defined above in Equation (32). Therefore, $\mathcal{A}$ is a desired model. $\square$

## G   Proof of Theorem 11

**Theorem 11.** $R^2$-GNN $\circ F \subseteq \mathcal{FOC}_2$ on any bounded graph class $\mathcal{G}_b$.

Before we go into theorem itself, we first introduce Lemma 28 that will be used in following proof.

**Lemma 28.** *Let $\varphi(x,y)$ denote a $\mathcal{FOC}_2$ formula with two free variables, for any natural number $n$, the following sentence can be captured by $\mathcal{FOC}_2$:*

*__There exists no less than__ $n$ __ordered node pairs__ $(a,b)$ __such that__ $(G,a,b) \models \varphi$.*

*Let $c$ denote the graph classifier such that $c(G) = 1$ iff $G$ satisfies the sentence above.*

*Proof.* The basic intuition is to define $m_i, 1 \leq i < n$ as the number of nodes $a$, such that there are **exactly** $i$ nodes $b$ that $\varphi(a,b)$ is *true*. Specially, we define $m_n$ as the number of nodes $a$, such that there are **at least** $n$ nodes $b$ that $\varphi(a,b)$ is *true*. Since $\sum_{i=1}^n im_i$ exactly counts the number of valid ordered pairs when $m_n = 0$, and it guarantees the existence of at least $n$ valid ordered pairs when $m_n > 0$. It's not hard to see that for any graph $G$, $c(G) = 1 \Leftrightarrow \sum_{i=1}^n im_i \geq n$. Futhermore, fix a valid sequence $(m_1......m_n)$ such that $\sum_{i=1}^n im_i \geq n$, there has to be another sequence $(k_1......k_n)$ such that $n \leq \sum_{i=1}^n ik_i \leq 2n$ and $k_i \leq m_i$ for all $1 \leq i \leq n$. Therefore, We can enumerate all possibilities of valid $(k_1......k_n)$, and for each valid $(k_1......k_n)$ sequence, we judge whether there are **at least** $k_i$ such nodes $a$ for every $1 \leq i \leq n$.

Formally, $\varphi_i(x) := \exists^{[i]} y \varphi(x,y)$ can judge whether a node $a$ has exactly $i$ partners $b$ such that $\varphi(a,b) = 1$, where $\exists^{[i]} y \varphi(x,y)$ denotes "there are exactly $i$ such nodes $y$" which is the abbreviation of formula $(\exists^{\geq i} y \varphi(x,y)) \wedge (\neg \exists^{\geq i+1} y \varphi(x,y))$. The $\mathcal{FOC}_2$ formula equivalent to our desired sentence $c$ is as follows:

$$\bigvee_{\sum_{i=1}^n n \leq ik_i \leq 2n} \left( \bigwedge_{i=1}^{n-1} \exists^{\geq k_i} x \left( \exists^{[i]} y \varphi(x,y) \right) \right) \wedge \left( \exists^{\geq k_n} x \left( \exists^{\geq n} y \varphi(x,y) \right) \right) \tag{33}$$

This $\mathcal{FOC}_2$ formula is equivalent to our desired classifier $c$. $\square$

With the Lemma 28, we now start to prove Theorem 11.

*Proof.* By Theorem 5, it follows that $R^2$-GNNs $\circ F \subseteq \mathcal{FOC}_2 \circ F$. Therefore it suffices to show $\mathcal{FOC}_2 \circ F \subseteq \mathcal{FOC}_2$.

By Lemma 22, it suffices to show $\mathcal{RSFOC}_2 \circ F \subseteq \mathcal{FOC}_2$. Since $\wedge, \vee, \neg$ are common rules. We only need to show for any $\mathcal{RSFOC}_2$ formula of form $\varphi(x) := \exists^{\geq n} y(\varphi_S(x,y) \wedge \varphi'(y))$ under transformed predicate set $P' = P \cup \{aux1, aux2, primal\}$, there exists an $\mathcal{FOC}_2$ formula $\varphi^1$ such that for any graph $G$ under $P$ and its node $v$, $(G,v) \models \varphi^1 \Leftrightarrow (F(G),v) \models \varphi$.

In order to show this, we consider a stronger result:

For any such formula $\varphi$, including the existence of valid $\varphi^1$, we claim there also exists an $\mathcal{FOC}_2$ formula $\varphi^2$ with two free variables such that the following holds: for any graph $G$ under $P$ and its added node $e_{ab}$ on $F(G)$, $(G,a,b) \models \varphi^2 \Leftrightarrow (F(G),e_{ab}) \models \varphi$. Call $\varphi^1, \varphi^2$ as first/second discriminant of $\varphi$.

Now we need to prove the existence of $\varphi^1$ and $\varphi^2$.

We prove by induction over quantifier depth $m$ of $\varphi$, Since we only add a single unary predicate *primal* in $P'$, any $\varphi(x)$ with quantifier depth 0 can be rewritten as $(primal(x) \wedge \varphi^1(x)) \vee (\neg primal(x) \wedge \varphi^2(x))$, where $\varphi^1(x), \varphi^2(x)$ are two formulas that only contain predicates in $P$. Therefore, $\varphi^1$ can be naturally seen as the first discriminant of $\varphi$. Moreover, since $\varphi^2(x)$ always evaluates on non-primal nodes, it is equivalent to $\bot$ or $\top$ under $\neg primal(x)$ constraint. Therefore, the corresponding $\bot$ or $\top$ can be seen as the second discriminant, so our theorem trivially holds for $m = 0$. Now assume it holds for $m \leq k$, we can assume quantifier depth of $\varphi = \exists^{\geq n} y(\varphi_S(x,y) \wedge \varphi'(y))$ is $m = k+1$.

Consider the construction rules of transformation $F$, for any two primal nodes in $F(G)$, there is no relation between them, for a primal node $a$ and an added node $e_{ab}$, there is exactly a single relation of type *aux1* between them. For a pair of added nodes $e_{ab}, e_{ba}$, there are a bunch of relations from the original graph $G$ and an additional *aux2* relation between them. Therefore, it suffices to only consider three possible kinds of $S \subseteq P_2 \cup \{aux1, aux2\}$ according to three cases mentiond above. Then, we will construct first/second determinants for each of these three cases. Since $\varphi'(y)$ has quantifier depth no more than $k$, by induction let $\widehat{\varphi}^1, \widehat{\varphi}^2$ be first/second discriminants of $\varphi'$ by induction.

1. $S = \{\textbf{aux1}\}$:

for primal node $a$, $\varphi(a)$ means the following: there exists at least $n$ nodes $b$, such that there is some relation between $a,b$ on $G$ and the added node $e_{ab}$ on $F(G)$ satisfies $\varphi'$. Therefore, the first determinant of $\varphi$ can be defined as following:

$$\varphi^1(x) := \exists^{\geq n} y, \left( \bigvee_{r \in P_2} r(x,y) \right) \wedge \widehat{\varphi}^2(x,y) \tag{34}$$

for added nodes $e_{ab}$ on $F(G)$, $\varphi(e_{ab})$ means $a$ satisfies $\varphi'$, so the second determinant of $\varphi$ is the following:

$$n = 1 : \varphi^2(x,y) := \widehat{\varphi}^1(x), \ n > 1 : \varphi^2(x,y) := \bot \tag{35}$$

2. $S = \{\textbf{aux2}\} \cup T, T \subseteq P_2, T \neq \emptyset$

primal nodes don't have *aux2* neighbors, so first determinant is trivially *false*.

$$\varphi^1(x) := \bot \tag{36}$$

For added node $e_{ab}$, $e_{ab}$ satisfies $\varphi$ iff there are exactly relations between $a,b$ of types in $T$, and $e_{ba}$ satisfies $\varphi'$. Therefore the second determinant is as follows, where $\varphi_T(x,y)$ is the *relation-specification* formula under $P$ introduced in Definition 21

$$n = 1 : \varphi^2(x,y) := \varphi_T(x,y) \wedge \widehat{\varphi}^2(y,x), n > 1 : \varphi^2(x,y) := \bot \tag{37}$$

3. $S = \emptyset$

For a subset $S \subseteq P_2 \cup \{aux1, aux2\}$, let $\varphi_S(x,y)$ denote the *relation-specification* formula under $P_2 \cup \{aux1, aux2\}$ defined in Definition 21.

Since we consider on bounded graph class $\mathcal{G}_b$, node number is bounded by a natural number $N$. For any node $a$ on $F(G)$, let $m$ denote the number of nodes $b$ on $F(G)$ such that $\varphi'(b) = 1$, let $m_0$ denote the number of nodes $b$ on $F(G)$ such that $\varphi'(b) = 1$ and there is a single relation *aux1*, between $(a,b)$ on $F(G)$, (That is equivalent to $\varphi_{\{aux1\}}(a,b) = 1$). For any $T \subseteq P_2$, let $m_T$ denote the number of nodes $b$ on $F(G)$ such that $\varphi'(b) = 1$ and $a,b$ has exactly relations of types in $T \cup \{aux2\}$ on $F(G)$, (That is equivalent to $\varphi_{T \cup \{aux2\}}(a,b) = 1$).

Note that the number of nodes $b$ on $F(G)$ such that $a,b$ don't have any relation, (That is equivalent to $\varphi_\emptyset(a,b) = 1$) and $\varphi'(b) = 1$ equals to $m - m_0 - \sum_{T \subseteq P_2} m_T$. Therefore, for any transformed graph $F(G)$ and its node $v$, $(F(G),v) \models \varphi \Leftrightarrow m - m_0 - \sum_{T \subseteq P_2} m_T \geq n$. Since $|V(G)| \leq N$ for all $G$ in bounded graph class $\mathcal{G}_b$, transformed graph $F(G)$ has node number no more than $N^2$. Therefore, we can enumerate all possibilities of $m,m_0,m_T \leq N^2, T \subset P_2$ such that the above inequality holds, and for each possibility, we judge whehter there exists exactly such number of nodes for each corresponding parameter. Formally speaking, $\varphi$ can be rewritten as the following form:

$$\tilde{\varphi}_{m,m_0}(x) := \left(\exists^{[m]} y \varphi'(y)\right) \wedge \left(\exists^{[m_0]} y (\varphi_{\{aux1\}}(x,y) \wedge \varphi'(y))\right) \tag{38}$$

$$\varphi(x) \equiv \bigvee_{m - m_0 - \sum_{T \subseteq P_2} m_T \geq n, 0 \leq m, m_0, m_T \leq N^2} \left( \tilde{\varphi}_{m,m_0}(x) \wedge \left( \bigwedge_{T \subseteq P_2} \exists^{[m_T]} y, (\varphi_{T \cup \{aux2\}}(x,y) \wedge \varphi'(y)) \right) \right) \tag{39}$$

where $\exists^{[m]} y$ denotes there are exactly $m$ nodes $y$.

Since first/second determinant can be constructed trivially under combination of $\wedge, \vee, \neg$, and we've shown how to construct determinants for formulas of form $\exists^{\geq n} y (\varphi_S(x,y) \wedge \varphi'(y))$ when $S = \{aux1\}$ and $S = \{aux2\} \cup T, T \subseteq P_2$ in the previous two cases. Therefore, in Equation (38) and Equation (39), the only left part is the formula of form $\exists^{[m]} y \varphi'(y)$. The only remaining work is to show how to construct first/second determinants for formula in form $\varphi(x) := \exists^{\geq n} y \varphi'(y)$.

Let $m_1$ denote the number of primal nodes $y$ that satisfies $\varphi'(y)$ and let $m_2$ denote the number of non-primal nodes $y$ that satisfies $\varphi'(y)$. It's not hard to see that for any node $v$ on $F(G)$, $(F(G),v) \models \varphi \Leftrightarrow m_1 + m_2 \geq n$. Therefore, $\varphi(x) = \exists^{\geq n} y \varphi'(y)$ that evaluates on $F(G)$ is equivalent to the following sentence that evaluates on $G$: *"There exists two natural numbers $m_1, m_2$ such that the following conditions hold: 1. $m_1 + m_2 = n$. 2. There are at least $m_1$ nodes $b$ on $G$ that satisfies $\widehat{\varphi}^1$, (equivalent to $(F(G),b) \models \varphi'$). 3. There are at least $m_2$ ordered node pairs $a,b$ on $G$ such that $a,b$ has some relation and $(G,a,b) \models \widehat{\varphi}^2$, (equivalent to $(F(G),e_{ab}) \models \varphi'$)."*

Formally speaking, rewrite the sentence above as formula under $P$, we get the following construction for first/second determinants of $\varphi$.

$$\varphi^1(x) = \varphi^2(x,y) = \bigvee_{m_1 + m_2 = n} \left( (\exists^{\geq m_1} y, \widehat{\varphi}^1(y)) \wedge \overline{\varphi}_{m_2} \right) \tag{40}$$

where $\overline{\varphi}_{m_2}$ is the $\mathcal{FOC}_2$ formula that expresses *"There exists at least $m_2$ ordered node pairs $(a,b)$ such that $(G,a,b) \models \widehat{\varphi}^2(x,y) \wedge (\bigvee_{r \in P_2} r(x,y))$".* We've shown the existence of $\overline{\varphi}_{m_2}$ in Lemma 28 $\qquad\square$

## H    Proof of Theorem 15

**Theorem 15.** *time-and-graph $\subsetneq R^2$-TGNN $\circ F^T$ = time-then-graph.*

For a graph $G$ with $n$ nodes, let $\mathbb{H}^V \in \mathbb{R}^{n \times d_v}$ denote node feature matrix, and $\mathbb{H}^E \in \mathbb{R}^{n \times n \times d_e}$ denote edge feature matrix, where $\mathbb{H}^E_{ij}$ denote the edge feature vector from $i$ to $j$.

First we need to define the GNN used in their frameworks. Note that for the comparison fairness, we add the the global readout to the node feature update as we do in $R^2$-GNNs. It recursively calculates the feature vector $\mathbb{H}^{V,(l)}_i$ of the node i at each layer $1 \leq l \leq L$ as follows:

$$\mathbb{H}^{V,(l)}_i = u^{(l)} \left( g^{(l)}(\{\!\!\{(\mathbb{H}^{V,(l-1)}_i, \mathbb{H}^{V,(l-1)}_j, \mathbb{H}^E_{ij}) \mid j \in \mathcal{N}(i)\}\!\!\}), r^{(l)}(\{\!\!\{\mathbb{H}^{V,(l-1)}_j | j \in V\}\!\!\}) \right) \tag{41}$$

where $\mathcal{N}(i)$ denotes the set of all nodes that adjacent to $i$, and $u^{(l)}, g^{(l)}, r^{(l)}$ are learnable functions. Note that here the GNN framework is a little different from the general definition defined in Equation (2). However, this framework is hard to fully implement and many previous works implementing *time-and-graph* or *time-then-graph* Gao and Ribeiro [2022] (Li et al. [2019], Seo et al. [2016], Chen et al. [2018], Manessi et al. [2020], Sankar et al. [2018], Rossi et al. [2020b]) don't reach the expressiveness of Equation (41). This definition is more for the theoretical analysis. In contrast, our

definition for GNN in Equation (1) and Equation (2) is more practical since it is fully captured by a bunch of commonly used models such as Schlichtkrull et al. [2018]. For notation simplicity, for a GNN $\mathcal{A}$, let $\mathbb{H}^{V,(L)} = \mathcal{A}(\mathbb{H}^V, \mathbb{H}^E)$ denote the node feature outputted by $\mathcal{A}$ using $\mathbb{H}^V, \mathbb{H}^E$ as initial features.

**Proposition 29.** *(Gao and Ribeiro [2022]):time-and-graph $\subsetneq$ time-then-grahp*

The above proposition is from **Theorem 1** of Gao and Ribeiro [2022]. Therefore, in order to complete the proof of Theorem 15, we only need to prove $R^2$-TGNN $\circ F^T = $ *time-then-graph*.

Let $G = \{G_1, \ldots, G_T\}$ denote a temporal knowledge graph, and $\mathbb{A}^t \in \mathbb{R}^{n \times |P_1|}, \mathbb{E}^t \in \mathbb{R}^{n \times n \times |P_2|}, 1 \leq t \leq T$ denonte one-hot encoding feature of unary facts and binary facts on timestamp $t$, where $P_1, P_2$ are unary and binary predicate sets.

The updating rule of a *time-then-graph* model can be generalized as follows:

$$\forall i \in V, \ \mathbb{H}_i^V = \textbf{RNN}([\mathbb{A}_i^1 \ldots \ldots \mathbb{A}_i^T]) \tag{42}$$

$$\forall i, j \in V, \ \mathbb{H}_{i,j}^E = \textbf{RNN}([\mathbb{E}_{i,j}^1 \ldots \ldots \mathbb{E}_{i,j}^T]) \tag{43}$$

$$\mathbf{X} := \mathcal{A}(\mathbb{H}^V, \mathbb{H}^E) \tag{44}$$

where $\mathcal{A}$ is a GNN defined above, **RNN** is an arbitrary Recurrent Neural Network. $\mathbf{X} \in \mathbb{R}^{n \times d}$ is the final node feature output of *time-then-graph*.

First we need to prove *time-then-graph* $\subseteq R^2$-TGNN$\circ F^T$. That is, for any *time-then-graph* model, we want to construct an equivalent $R^2$-TGNN $\mathcal{A}'$ to capture it on transformed graph. We can use nodes added after transformation to store the edge feature $\mathbb{H}^E$, and use primal nodes to store the node feature $\mathbb{H}^V$. By simulating **RNN** through choosing specific functions in $R^2$-TGNN, we can easily construct a $R^2$-TGNN $\mathcal{A}'$ such that for any node $i$, and any node pair $i,j$ with at least one edge in history, $\mathbf{x}_i = \mathbb{H}_i^V$ and $\mathbf{x}_{e_{ij}} = \mathbb{H}_{i,j}^E$ hold, where $\mathbf{x}_i$ and $\mathbf{x}_{e_{ij}}$ are features of corresponding primal node $i$ and added node $e_{ij}$ outputted by $\mathcal{A}'$.

Note that $\mathcal{A}'$ is a $R^2$-TGNN, it can be represented as $\mathcal{A}'_1 \ldots \ldots \mathcal{A}'_T$, where each $\mathcal{A}'_t, 1 \leq t \leq T$ is a $R^2$-GNN. $\mathcal{A}'$ has simulated work of **RNN**, so the remaining work is to simulate $\mathcal{A}(\mathbb{H}^V, \mathbb{H}^E)$. We do the simulation over induction on layer number $L$ of $\mathcal{A}$.

When $L = 0$, output of $\mathcal{A}$ is exactly $\mathbb{H}^V$, which has been simulated by $\mathcal{A}'$ above.

Suppose $L = k + 1$, let $\tilde{\mathcal{A}}$ denote $R^2$-GNN extracted from $\mathcal{A}$ but without the last layer $k + 1$. By induction, we can construct a $R^2$-TGNN $\tilde{\mathcal{A}}'$ that simulates $\tilde{\mathcal{A}}(\mathbb{H}^V, \mathbb{H}^E)$. Then we need to append three layers to $\tilde{\mathcal{A}}'$ to simulate the last layer of $\mathcal{A}$.

Let $u^{(L)}, g^{(L)}, r^{(L)}$ denote parameters of the last layer of $\mathcal{A}$. Using notations in Equation (2), let $\{C^{(l)}, (A_j^{(l)})_{j=1}^{|P_2|}, A_{aux1}^{(l)}, A_{aux2}^{(l)}, R^{(l)}\}_{l=1}^3$ denote parameters of the three layers appended to $\tilde{\mathcal{A}}'_T$. They are defined as follows:

First, we can choose specific function in the first two added layers, such that the following holds:

**1.** For any added node $e_{ij}$, feature outputted by the new model is $\mathbf{x}_{e_{ij}}^{(2)} = [\mathbb{H}_{ij}^E, \mathbf{x}_i', \mathbf{x}_j']$, where $\mathbf{x}^{(2)}$ denotes the feature outputted by the second added layer, and $\mathbf{x}_i', \mathbf{x}_j'$ are node features of $i,j$ outputted by $\tilde{\mathcal{A}}'$. For a feature $\mathbf{x}$ of added node of this form, we define $\mathbf{x}_0, \mathbf{x}_1, \mathbf{x}_2$ as corresponding feature slices where $\mathbb{H}_{ij}^E, \mathbf{x}_i', \mathbf{x}_j'$ have been stored.

**2.** For any primal node, its feature $\mathbf{x}$ only stores $\mathbf{x}_i'$ in $\mathbf{x}_1$, and $\mathbf{x}_0, \mathbf{x}_2$ are all slices of dummy bits.

Let $\mathbf{X}$ be a multiset of features that represents function input. For the last added layer, we can choose specific functions as follows:

$$R^{(3)}(\mathbf{X}) := r^{(L)}(\{\!\{\mathbf{x}_1 | \mathbf{x} \in \mathbf{X}, \textbf{primal}(\mathbf{x})\}\!\}) \tag{45}$$

$$A_{aux1}^{(3)}(\mathbf{X}) := g^{(L)}(\{\!\{(\mathbf{x}_1, \mathbf{x}_2, \mathbf{x}_0) | \mathbf{x} \in \mathbf{X}\}\!\}) \tag{46}$$

$$C^{(3)}(\mathbf{x}_{aux1}, \mathbf{x}_g) := u^{(L)}(\mathbf{x}_{aux1}, \mathbf{x}_g) \tag{47}$$

where $\mathbf{x}_{aux1},\mathbf{x}_g$ are outputs of $R^{(3)}$ and $A^{(3)}_{aux1}$, and all useless inputs of $C^{(3)}$ are omitted. Comparing this construction with Equation (41). It's east to see that after the last layer appended, we can construct an equivalent $R^2$-TGNN $\mathcal{A}'$ that captures $\mathcal{A}$ on transformed graph. By inductive argument, we prove *time-then-graph* $\subseteq$ R $^2$-TGNN $\circ F^T$.

Then we need to show $R^2$-TGNN $\circ F^T \subseteq$ *time-then-graph*.

In Theorem 16, we will prove $R^2$-TGNN $\circ F^T = R^2$-GNN $\circ F \circ H$. Its proof doesn't dependent on Theorem 15, so let's assume it's true for now. Then, instead of proving $R^2$-TGNN $\circ F^T$, it's sufficient to show $R^2$-GNN $\circ F \circ H \subseteq$ *time-then-graph*.

Let $P_1^T, P_2^T$ denote the set of temporalized unary and binary predicate sets defined in Definition 12. Based on *most expressive ability* of Recurrent Neural Networks shown in Siegelmann and Sontag [1992], we can get a *most expressive representation* for unary and binary fact sequences through **RNN**. A *most expressive* RNN representation function is always injective, thus there exists a decoder function translating most-expressive representations back to raw sequences. Therefore, we are able to find an appropriate **RNN** such that its output features $\mathbb{H}^V,\mathbb{H}^E$ in Equation (42), Equation (43) contain all information needed to reconstruct all temporalized unary and binary facts related to the corresponding nodes.

For any $R^2$-GNN $\mathcal{A}$ on transformed collpsed temporal knowledge graph, we want to construct an equivalent *time-then-graph* model $\{\mathbf{RNN},\mathcal{A}'\}$ to capture $\mathcal{A}$. In order to show the existence of the *time-then-graph* model, we will do an inductive construction over layer number $L$ of $\mathcal{A}$. Here in order to build inductive argument, we will consider a following stronger result and aim to prove it: In additional to the existence of $\mathcal{A}'$, we claim there also exists a function $f_{\mathcal{A}}$ with the following property: For any two nodes $a,b$ with at least one edge, $f_{\mathcal{A}}(\mathbf{x}'_a,\mathbf{x}'_b,\mathbb{H}^E_{ab}) = \mathbf{x}_{e_{ab}}$, where $\mathbf{x}'_a,\mathbf{x}'_b,\mathbb{H}^E_{ab}$ are features of $a$, $b$ and edge information between $a,b$ outputted by $\mathcal{A}'$, and $\mathbf{x}_{e_{ab}}$ is the feature of added node $e_{ab}$ outputted by $\mathcal{A} \circ F \circ H$. It suffices to show that there exists such function $f_{\mathcal{A}}$ as well as a *time-then-graph* model $\{\mathbf{RNN},\mathcal{A}'\}$ such that the following conditions hold:

For any graph $G$ and its node $a,b \in V(G)$,

1. $\mathbb{H}^{V,(l)}_a = [\mathbf{x}_a,Enc(\{\!\!\{\mathbf{x}_{e_{aj}}|j \in \mathcal{N}(a)\}\!\!\})]$.

2.If there is at least one edge between $a,b$ in history, $f_{\mathcal{A}}(\mathbb{H}^{V,(l)}_a,\mathbb{H}^{V,(l)}_b,\mathbb{H}^E_{ab}) = \mathbf{x}_{e_{ab}}$. Otherwise, $f_{\mathcal{A}}(\mathbb{H}^{V,(l)}_a,\mathbb{H}^{V,(l)}_b,\mathbb{H}^E_{ab}) = \mathbf{0}$

where $\mathbb{H}^{V,(l)}_a,\mathbb{H}^{V,(l)}_b$ are node features outputted by $\mathcal{A}'$, while $\mathbf{x}_a,\mathbf{x}_{e_{ab}}$ are node features outputted by $\mathcal{A}$ on transformed collpased graph. $Enc(\mathbf{X})$ is some injective encoding that stores all information of multiset $\mathbf{X}$. For a node feature $\mathbb{H}^{V,(l)}_a$ of above form, let $\mathbb{H}^{V,(l)}_{a,0} := \mathbf{x}_a, \mathbb{H}^{V,(l)}_{a,1} = Enc(\{\!\!\{\mathbf{x}_{e_{aj}}|j \in \mathcal{N}(a)\}\!\!\})$ denote two slices that store independent information in different positions.

For the base case $L = 0$. the node feature only depends on temporalized unary facts related to the corresponding node. Since by **RNN** we can use *most expressiveness representation* to capture all unary facts. A specific **RNN** already captures $\mathcal{A}$ when $L = 0$. Moreover, there is no added node $e_{ab}$ that relates to any unary fact, so a constant function already satisfies the condition of $f_{\mathcal{A}}$ when $L = 0$. Therefore, our result holds for $L = 0$

Assume $L = k+1$, let $\widehat{\mathcal{A}}$ denote the model generated by the first $k$ layers of $\mathcal{A}$. By induction, there is *time-then-graph* model $\widehat{\mathcal{A}}'$ and function $f_{\widehat{\mathcal{A}}'}$ that captures output of $\widehat{\mathcal{A}}'$ on transformed collapsed graph. We can append a layer to $\widehat{\mathcal{A}}'$ to build $\mathcal{A}'$ that simulates $\mathcal{A}$. Let $\{C^{(L)},(A^{(L)}_j)^{T|P_2|}_{j=1},A^{(L)}_{aux1},A^{(L)}_{aux2},R^{(L)}\}$ denote the building blocks of layer $L$ of $\mathcal{A}$, and let $u^*,g^*,r^*$ denote functions used in the layer that will be appended to $\widehat{\mathcal{A}}'$. They are defined below:

$$g^*(\{\!\!\{(\mathbb{H}^{V,(l-1)}_i,\mathbb{H}^{V,(l-1)}_j,\mathbb{H}^E_{ij}|j \in \mathcal{N}(i))\}\!\!\}) := A^{(L)}_{aux1}(\{\!\!\{f_{\widehat{\mathcal{A}}'}(\mathbb{H}^{V,(l-1)}_i,\mathbb{H}^{V,(l-1)}_j,\mathbb{H}^E_{ij})|j \in \mathcal{N}(i)\}\!\!\}) \quad (48)$$

$$r^*(\{\!\!\{\mathbb{H}^{V,(l-1)}_j|j \in V(G)\}\!\!\}) = R^{(L)}\left(\{\!\!\{\mathbb{H}^{V,(l-1)}_{j,0}|j \in V(G)\}\!\!\} \cup (\bigcup_{j \in V(G)} Dec(\mathbb{H}^{V,(l-1)}_{j,1}))\right) \quad (49)$$

$$u^*(\mathbf{x}_g,\mathbf{x}_r) = C^{(L)}(\mathbf{x}_g,\mathbf{x}_r) \quad (50)$$

where $\mathbf{x}_g,\mathbf{x}_r$ are outputs of $g^*$ and $r^*$. $Dec(\mathbf{X})$ is a decoder function that do inverse mapping of $Enc(\mathbf{X})$ mentioned above, so $Dec(\mathbb{H}^{V,(l-1)}_{j,1})$ is actually $\{\!\!\{\mathbf{x}_{e_{aj}}|j \in \mathcal{N}(a)\}\!\!\}$. Note that primal nodes

in transformed graph only has type *aux1*- neighbors, so two inputs $\mathbf{x}_g,\mathbf{x}_r$, one for *aux1* aggregation output and one for global readout are already enough for computing the value. Comparing the three rules above with Equation (2), we can see that our new model $\mathcal{A}'$ perfectly captures $\mathcal{A}$.

We've captured $\mathcal{A}$, and the remaining work is to construct $f_{\mathcal{A}}$ defined above to complete inductive assumption. We can just choose a function that simulates message passing between pairs of added nodes $e_{ab}$ and $e_{ba}$ as well as message passing between $e_{ab}$ and $a$, and that function satisfies the condition for $f_{\mathcal{A}}$. Formally speaking, $f_{\mathcal{A}}$ can be defined below:

$$f_{\mathcal{A}}(\mathbb{H}_i^{V,(l)},\mathbb{H}_j^{V,(l)},\mathbb{H}_{ij}^E) := \mathbf{Sim}_{\mathcal{A}_L}(\mathbb{H}_i^{V,(l-1)}, \mathbb{H}_g^{(l-1)},g_{ij},g_{ji}, \mathbb{H}_{ij}^E) \tag{51}$$

$$g_{ij} := f_{\widehat{\mathcal{A}}'}(\mathbb{H}_i^{V,(l-1)},\mathbb{H}_j^{V,(l-1)},\mathbb{H}_{ij}^E), \mathbb{H}_g^{(l-1)} := \{\!\!\{\mathbb{H}_i^{V,(l-1)}|i \in V(G)\}\!\!\} \tag{52}$$

Let's explain this equation, $\mathbf{Sim}_{\mathcal{A}_L}(a,g,s,b,e)$ is a local simulation function which simulates single-iteration message passing in the following scenario:

Suppose there is a graph $H$ with three constants $V(H) = \{a,e_{ab},e_{ba}\}$. There is an *aux1* edge between $a$ and $e_{ab}$, an *aux2* edge between $e_{ab}$ and $e_{ba}$, and additional edges of different types between $e_{ab}$ and $e_{ba}$. The description of additional edges can be founded in $e$. Initial node features of $a,e_{ab},e_{ba}$ are set to $a,s,b$ respectively. and the global readout output is $g$. Finally, run $L$-th layer of $\mathcal{A}$ on $H$, and $\mathbf{Sim}_{\mathcal{A}_L}$ is node feature of $e_{ab}$ outputted by $\mathcal{A}_L$.

Note that if we use appropriate injective encoding or just use concatenation technic, $\mathbb{H}_g^{(l-1)}, \mathbb{H}_i^{V,(l-1)},\mathbb{H}_j^{V,(l-1)}$ can be accessed from $\mathbb{H}_i^{V,(l)},\mathbb{H}_i^{V,(l)}$. Therefore the above definition for $f_{\mathcal{A}}$ is well-defined. Moreover, in the above explanation we can see that $f_{\mathcal{A}}(\mathbb{H}_i^{V,(l-1)},\mathbb{H}_j^{V,(l-1)},\mathbb{H}_{ij}^E)$ is exactly node feature of $e_{ij}$ outputted by $\mathcal{A}$ on the transformed collapsed graph, so our proof finishes.

# I  Proof of Theorem 16

**Theorem 16.** $R^2\text{-}TGNN \circ F^T = R^2\text{-}TGNN \circ F \circ H$.

First, we recall the definition for R$^2$-TGNN as in Equation (53):

$$\mathbf{x}_v^t = \mathcal{A}_t\left(G_t,v,\mathbf{y}^t\right) \quad \text{where} \quad \mathbf{y}_v^t = [I_{G_t}(v) : \mathbf{x}_v^{t-1}], \forall v \in V(G_t) \tag{53}$$

We say a R$^2$-TGNN is *homogeneous* if $\mathcal{A}_1,\ldots,\mathcal{A}_T$ share the same parameters. In particular, we first prove Lemma 30, namely, *homogeneous* R$^2$-TGNN and R$^2$-TGNN (where paramters in $\mathcal{A}_1,\ldots,\mathcal{A}_T$ may differ) have the same expressiveness.

**Lemma 30.** *homogenous* $R^2\text{-}TGNN = R^2\text{-}TGNN$

*Proof.* The forward direction *homogeneous* R$^2$-TGNN$\subseteq$ R$^2$-TGNN trivially holds. It suffices to prove the backward direction.

Let $\mathcal{A} : \{\mathcal{A}_t\}_{t=1}^T$ denote a R$^2$-TGNN. Without loss of generality, we can assume all models in each timestamps have the same layer number $L$. Then for each $1 \leq t \leq T$, we can assume all $\mathcal{A}_t$ can be represented by $\{C_t^{(l)},(A_{t,j}^{(l)})_{j=1}^{|P_2|},R_t^{(l)}\}_{l=1}^L$. Futhormore, without loss of generality, we can assume all output dimensions for $A_{t,j}^{(l)},R_t^{(l)}$ and $C_t^{(l)}$ are $d$. As for input dimension, all of these functions also have input dimension $d$ for $2 \leq l \leq L$. Specially, by updating rules of R$^2$-TGNN Equation (53), in the initialization stage of each timestamp we have to concat a feature with length $|P_1|$ to output of the former timestamp, so the input dimension for $A_{t,j}^{(1)},R_t^{(1)},C_t^{(1)}$ is $d + |P_1|$.

We can construct an equivalent *homogeneous* R$^2$-TGNN with $L$ layers represented by $\{C^{*,(l)},(A_j^{*,(l)})_{j=1}^{|P_2|},R^{*,(l)}\}_{l=1}^L$. For $2 \leq l \leq L$, $C^{*,(l)} A_j^{*,(l)},R^{*,(l)}$ use output and input feature dimension $d' = Td$. Similar to the discussion about feature dimension above, since we need to concat the unary predicates information before each timestamp, for layer $l = 1$, $C^{*,(1)},A_j^{*,(1)},R^{*,(1)}$ have input dimension $d' + |P_1|$ and output dimension $d'$. For dimension alignment, $\mathbf{x}_v^0$ used in Equation (53) is defined as zero-vector with length $d'$.

Next let's define some symbols for notation simplicity. For a feature vector $\mathbf{x}$, let $\mathbf{x}[i,j]$ denotes the slice of $\mathbf{x}$ in dimension $[i,j]$. By the discussion above, in the following construction process we will only need feature $\mathbf{x}$ with dimension $d'$ or $d' + |P_1|$. When $\mathbf{x}$ has dimension $d'$, $\mathbf{x}^{(i)}$ denotes $\mathbf{x}[(i-1)d+1,id]$, otherwise it denotes $\mathbf{x}[|P_1| + (i-1)d + 1, |P_1| + id]$. Let $[\mathbf{x}_1......\mathbf{x}_T]$ or $[\mathbf{x}_t]_{t=1}^T$ denotes the concatenation of a sequence of feature $\mathbf{x}_1......\mathbf{x}_T$, and $[\mathbf{x}]^n$ denote concatenation of $n$ copies of $\mathbf{x}$, $\mathbf{0}^n$ denotes zero vectors of length $n$. Furthermore. Let $\mathbf{X}$ denotes a multiset of $\mathbf{x}$. Follows the updating rules defined in Equation (2), for all $1 \leq j \leq |P_2|, 1 \leq l \leq L, A_j^{*,(l)}, R^{*,(l)}$ should get input of form $\mathbf{X}$, and the combination function $C^{*,(l)}$ should get input of form $(\mathbf{x}_0, (\mathbf{x}_j)_{j=1}^{|P_2|}, \mathbf{x}_g)$, where $\mathbf{x}_0$ is from the node itself, $(\mathbf{x}_j)_{j=1}^{|P_2|}$ are from aggregation functions $(A_j^{*,(l)})_{j=1}^{|P_2|}$ and $\mathbf{x}_g$ is from the global readout $R^{*,(l)}$. The dimension of $\mathbf{x}$ or $\mathbf{X}$ should match the input dimension of corresponding function. For all $1 \leq l \leq L$, parameters in layer $l$ for the new model are defined below

$$l = 1 : C^{*,(l)}(\mathbf{x}_0, (\mathbf{x}_j)_{j=1}^{|P_2|}, \mathbf{x}_g) := [C_t^{(l)}([\mathbf{x}_0[1,|P_1|], \mathbf{x}_0^{(t-1)}], (\mathbf{x}_j^{(t)})_{j=1}^{|P_2|}, \mathbf{x}_g^{(t)})]_{t=1}^T \tag{54}$$

$$2 \leq l \leq L : C^{*,(l)}(\mathbf{x}_0, (\mathbf{x}_j)_{j=1}^{|P_2|}, \mathbf{x}_g) := [C_t^{(l)}(\mathbf{x}_0^{(t)}, (\mathbf{x}_j^{(t)})_{j=1}^{|P_2|}, \mathbf{x}_g^{(t)})]_{t=1}^T \tag{55}$$

$$\forall j \in [K], l = 1 : A_j^{*,(l)}(\mathbf{X}) = [A_{t,j}^{(l)}(\{\!\{[\mathbf{x}[1,|P_1|], \mathbf{x}^{(t-1)}] | \mathbf{x} \in \mathbf{X}\}\!\})]_{t=1}^T \tag{56}$$

$$l = 1 : R^{*,(l)}(\mathbf{X}) = [R_t^{(l)}(\{\!\{[\mathbf{x}[1,|P_1|], \mathbf{x}^{(t-1)}] | \mathbf{x} \in \mathbf{X}\}\!\})]_{t=1}^T \tag{57}$$

$$\forall j \in [K], 2 \leq l \leq L : A_j^{*,(l)}(\mathbf{X}) = [A_{t,j}^{(l)}(\{\!\{\mathbf{x}^{(t)} | \mathbf{x} \in \mathbf{X}\}\!\})]_{t=1}^T \tag{58}$$

$$2 \leq l \leq L : R^{*,(l)}(\mathbf{X}) = [R_t^{(l)}(\{\!\{\mathbf{x}^{(t)} | \mathbf{x} \in \mathbf{X}\}\!\})]_{t=1}^T \tag{59}$$

The core trick is to use $T$ disjoint slices $\mathbf{x}^{(1)}......\mathbf{x}^{(T)}$ to simulate $T$ different models $\mathcal{A}_1......\mathcal{A}_T$ at the same time, Since these slices are isolated from each other, a proper construction above can be found. The only speciality is that in layer $l = 1$, we have to incorporate the unary predicate information $\mathbf{x}[1,|P_1|]$ into each slice. By the construction above, we can see that for any node $v$, $\mathbf{x}_v^{(T)}$ is exactly the its feature outputted by $\mathcal{A}$. Therefore, we finally construct an *homogeneous* $R^2$-TGNN equivalent with $\mathcal{A}$. $\qquad\square$

Now, we start to prove Theorem 16.

**Theorem 16.** *$R^2$-TGNNs $\circ F^T = R^2$-GNNs $\circ F \circ H$ on any universal graph class $\mathcal{G}_u$.*

*Proof.* Since $R^2$-TGNN $\circ F^T$ only uses a part of predicates of $P' = F(H(P))$ in each timestamp, the forward direction $R^2$-TGNN $\circ F^T \subseteq R^2$-GNN $\circ F \circ H$ trivially holds.

For any $R^2$-GNN $\mathcal{A}$ under $P'$, we want to construct an $R^2$-TGNN $\mathcal{A}'$ under $F^T(P)$ such that for any temporal knowledge graph $G$, $\mathcal{A}'$ outputs the same feature vectors as $\mathcal{A}$ on $F^T(G)$. We can assume $\mathcal{A}$ is represented as $(C^{(l)}, (A_j^{(l)})_{j=1}^K, A_{aux1}^{(l)}, A_{aux2}^{(l)}, R^{(l)})_{l=1}^L$, where $K = T|P_2|$.

First, by setting feature dimension to be $d' = T|P| + 3$. We can construct an $R^2$-TGNN $\mathcal{A}'$ whose output feature stores all facts in $F(H(G))$ for any graph $G$. Formally speaking, $\mathcal{A}'$ should satisfy the following condition:

For any primal node $a$, its feature outputted by $\mathcal{A}' \circ F^T$ should store all unary facts of form $A_i(a), A_i \in T|P_1|$ or $primal(a)$ on $F(H(G))$. For any non-primal node $e_{ab}$, its feature outputted by $\mathcal{A}' \circ F^T$ should store all binary facts of form $r_i(a,b), r_i \in T|P_2|$ or $r_{aux1}(a,b), r_{aux2}(a,b)$ where $b$ is another node on $F(H(G))$.

The $\mathcal{A}'$ is easy to construct since we have enough dimension size to store different predicates independently, and these facts are completely encoded into the initial features of corresponding timestamp. Let $(\mathcal{A}_1'......\mathcal{A}_T')$ denote $\mathcal{A}'$.

Next, in order to simulate $\mathcal{A}$, we need to append some layers to $\mathcal{A}_T'$. Let $L$ denote the layer number of $\mathcal{A}$, we need to append $L$ layers represented as $(C^{*,(l)}, (A_j^{*,(l)})_{j=1}^{|P_2|}, A_{aux1}^{*,(l)}, A_{aux2}^{*,(l)}, R^{*,(l)})_{l=1}^L$

Since we have enough information encoded in features, we can start to simulate $\mathcal{A}$. Since neighbor distribution of primal nodes don't change between $F^T(G)_T$ and $F(H(G))$, it's easy to simulate all messages passed to primal nodes as destinations by $A_{aux1}^{*,(l)}$. For messages passed to non-primal node

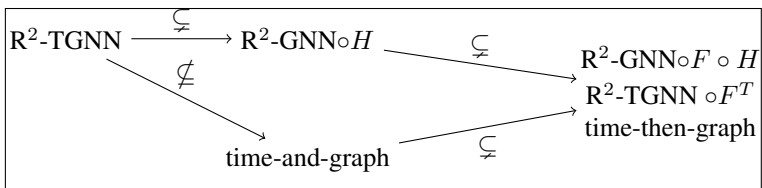

Figure 6: Hierarchic expressiveness.

$e_{ab}$ as destination, it can be divided into messages from $a$ and messages from $e_{ba}$. The first class of messages is easy to simulate since the $aux1$ edge between $e_{ab}$ and $a$ is the same on $F^T(G)_T$ and $F(H(G))$.

For the second class of messages, since edges of type $r_i, 1 \leq i \leq T|P_2|$ may be lost in $F^T(G)_T$, we have to simulate these messages only by the unchanged edge of type **aux2**. It can be realized by following construction:

$$1 \leq l \leq L, A_{aux2}^{*,(l)}(\mathbf{X}) = [[A_j'^{,(l)}(\mathbf{X})]]_{j=1}^{K}, A_{aux2}^{(l)}(\mathbf{X})] \tag{60}$$

where $K = T|P_2|$, $A_j'^{,(l)}(\mathbf{X}) := A_j^{(l)}(\mathbf{X})$ if and only if $e_{ba}$ has neighbor $r_j$ on $F(H(G))$ , otherwise $A_j'^{(l)}(\mathbf{X}) := \mathbf{0}$. Note that $\mathbf{X}$ is exactly the feature of $e_{ba}$, and we can access the information about its $r_j$ neighbors from feature since $\mathcal{A}'$ has stored information about these facts.

In conclusion, we've simulated all messages between neighbors. Furthermore, since node sets on $F^T(G)_T$ and $F(H(G))$ are the same, global readout $R^{(l)}$ is also easy to simulate by $R^{*,(l)}$. Finally, using the original combination function $C^{(l)}$, we can construct an R²-TGNN on $F^T$ equivalent to $\mathcal{A}$ on $F(H(G))$ for any temporal knowledge graph $G$.

$\square$

## J  Proof of Theorem 17

Based on Theorem 15, Theorem 16 and Corollary 11.2, in order to prove Theorem 17, it suffices to show the following theorems.

**Theorem 31.** *If time range $T > 1$ R²-TGNN $\subsetneq$ R²-GNN $\circ H$.*

**Theorem 32.** *If time range $T > 1$ R²-TGNN $\not\subseteq$ time-and-graph.*

*Proof.* Since a formal proof Theorem 32 relates to too many details in definition of time-and-graph (Please refer to Gao and Ribeiro [2022]) which is not the focus here. We will just a brief proof sketch of Theorem 32: That's because time-and-graph can not capture a chain of information that is continuously scattered in time intervals. Specifically, $\varphi(x) := \exists^{\geq 1} y, \left( r_1^2(x,y) \wedge (\exists^{\geq 1} x, r_1^1(y,x)) \right)$ can't be captured by time-and-graph but $\varphi(x)$ is in R²-TGNN.

We mainly give a detaild proof of Theorem 31: Since in each timestamp $t$, R²-TGNN only uses a part of predicates in temporalized predicate set $P' = H(P)$, R²-TGNN $\subseteq$ R²-GNN $\circ H$ trivially holds. To show R²-TGNN is strictly weaker than R²-GNN $\circ H$. Consider the following classifier:

Let time range $T = 2$, and let $r$ be a binary predicate in $P_2$. Note that there are two different predicates $r^1, r^2$ in $P' = H(P)$. Consider the following temporal graph $G$ with 5 nodes $\{1,2,3,4,5\}$. its two snapshots $G_1, G_2$ are as follows:

$G_1 = \{r(1,2), r(4,5)\}$

$G_2 = \{r(2,3)\}$.

It follows that after transformation $H$, the static version of $G$ is:

$H(G) = \{r_1(1,2), r_1(4,5), r_2(2,3)\}$.

Consider the logical classifier $\exists y \left( r_1(x,y) \wedge (\exists x r_2(x,y)) \right)$ under $P'$. It can be captured by some R²-GNN under $P'$. Therefore, R²-GNN $\circ H$ can distinguish nodes 1,4.

| datasets | $\varphi_1$ | $\varphi_2$ | $\varphi_3$ | $\varphi_4$ |
|---|---|---|---|---|
| Avg # Nodes | 477 | 477 | 477 | 477 |
| Time_range | 2 | 2 | 2 | 10 |
| # Unary predicate | 2 | 2 | 2 | 3 |
| # Binary predicate(non-temporalized) | 1 | 1 | 1 | 3 |
| Avg # Degree (in single timestamp) | 3 | 3 | 3 | 5 |
| Avg # positive percentage | 50.7 | 52 | 25.3 | 73.3 |

Table 4: statistical information for synthetic datasets.

| datasets | AIFB | MUTAG | Brain-10 |
|---|---|---|---|
| # Nodes | 8285 | 23644 | 5000 |
| Time_Range | \ | \ | 12 |
| # Relation types | 45 | 23 | 20 |
| # Edges | 29043 | 74227 | 1761414 |
| # Classes | 4 | 2 | 10 |
| # Train Nodes | 140 | 272 | 4500 |
| # Test Nodes | 36 | 68 | 500 |

Table 5: statistical information for Real datasets.

| hyper-parameter | range |
|---|---|
| learning rate | 0.01 |
| combination | mean/max/add |
| aggregation/readout | mean/max/add |
| layer | 1,2,3 |
| hidden dimension | 10,64,100 |

Table 6: Hyper-parameters.

However, any $R^2$-TGNN based on updating rules in Equation (53) can't distinguish these two nodes, so $R^2$-TGNN is strictly weaker than $R^2$-GNN $\circ H$. $\qquad\square$

Based on Theorem 31, we can consider logical classifier $\varphi_{\mathbf{3}} \coloneqq \exists^{\geq 2} y(p_1^1(x,y) \wedge p_1^2(x,y))$. Note that this classifier is just renaming version of Figure 1. Therefore $\varphi_3$ can't be captured by $R^2$-GNN $\circ H$, not to say weaker framework $R^2$-GNN by Theorem 31.

.

# K  Experiment Supplementary

## K.1  Synthetic dataset generation

For each synthetic datasets, we generate 7000 graphs as tranining set and 500 graphs as test set. Each graph has $50 - 1000$ nodes. In graph generation, we fix the expected edge density $\delta$. In order to generate a graph with $n$ nodes, we pick $\delta n$ pairs of distinct nodes uniformly randomly. For each selected node pair $a,b$, each timestamp $t$ and each binary relation type $r$, we add $r^t(a,b)$ and $r^t(a,b)$ into the graph with independent probability $\frac{1}{2}$.

## K.2  Statistical Information for Datasets

We list the information for synthetic dataset in Table 4 and real-world dataset in Table 5. Note that synthetic datasets contains many graphs, but real-world datasets only contains a single graph. Therefore, for real-world dataset, we have two disjoint node set as train split and test split for training and testing respectively. In training, the model can see the subgraph induced by train split and unlabelled nodes, in testing, the model can see the whole graph but only evaluate the performance on test split.

| $\mathcal{FOC}_2$ classifier | $\varphi_1$ | $\varphi_2$ | $\varphi_3$ | $\varphi_4$ |
|---|---|---|---|---|
| R-GAT $\circ H$ | 100 | 61.4 | 88.6 | 82.0 |
| R$^2$-GAT $\circ H$ | 100 | 93.5 | 95.0 | 82.2 |
| R$^2$-GAT $\circ F \circ H$ | **100** | **98.2** | **100** | **95.8** |

Table 7: Extra results on synthetic datasets

| | AIFB | MUTAG | DGS | AM |
|---|---|---|---|---|
| # of nodes | 8285 | 23644 | 333845 | 1666764 |
| # of edges | 29043 | 74227 | 916199 | 5988321 |
| R-GCN | 95.8 | 73.2 | 83.1 | 89.3 |
| R-GAT | 96.9 | 74.4 | 86.9 | 90.0 |
| R-GNN | 91.7 | 76.5 | 81.2 | 89.5 |
| R$^2$-GNN | 91.7 | 85.3 | 85.5 | 89.9 |
| R$^2$-GNN $\circ F$ | **97.2** | **88.2** | **88.0** | **91.4** |

Table 8: Extra results for static real-world datasets.

| Models | GRU-GCN$\circ F^T$ | TGN $\circ F^T$ | R-TGNN | R-TGNN $\circ F^T$ | R$^2$-TGNN | R$^2$-TGNN$\circ F^T$ |
|---|---|---|---|---|---|---|
| Brain-10 | 95.0 | 94.2 | 85.0 | 90.9 | 94.8 | 94.0 |

Table 9: Extra results for temporal real-world dataset Brain-10.

### K.3 Hyper-parameters

For all experiments, we did grid search according to Table 6.

### K.4 More Results

Apart from those presented in main part, we have some extra experimental results here:

1. Extra results on synthetic datasets but using different base model architecture, where R-GAT refers to Busbridge et al. [2019] and R$^2$-GAT refers to its extension with global readout. Please Refer to Table 7. These results show the generality of our results on different base models within the framework.

2. Extra results for static real-world datasets. Add a base model R-GATBusbridge et al. [2019] and two larger real-world datasets DGS and AM from Schlichtkrull et al. [2018]. Please refer to Table 8. From the results for two bigger datasets DGM and AM, we can see our framework outperforms the other baselines, which confirms the scalability of our method and theoretical results. These results show our method is effective both on small and large graphs.

3. Extra results for temporal real-world dataset Brain-10. Please refer to Table 9. These results implies that our method is effective on different base models in temporal settings. Moreover, we can see separate improvements from global readout and graph transformation respectively. As we said in the main part, the drop in the last column may be due to the intrinsic drawbacks of current real-world datasets. Many real-world datasets can not be perfectly modeled as first-order-logic classifier. This non-logical property may lead to less convincing experimental results. As Barceló et al. [2020] commented, these commonly used benchmarks are inadequate for testing advanced GNN variants.

