# OpenReview forum: "Calibrate and Boost Logical Expressiveness of GNN Over Multi-Relational and Temporal Graphs"
_NeurIPS.cc/2023/Conference — NeurIPS 2023 poster_

### Official Review · Reviewer_pnMq · 2023-07-05

**Soundness:** 4 excellent
**Presentation:** 3 good
**Contribution:** 2 fair
**Rating:** 6
**Confidence:** 4

**Summary:**

The paper deals with classifying multi-relational graphs, and specifically, the R-GNN architecture, which deals with more than one type of edge by producing a different neighbor set for each of them, aggregating the messages of each neighbor, and then merging them together in a final update layer. The first results in the paper are that these architectures are limited, as Figure 1 shows, these architectures cannot even count the number of total neighbours of a node. To fix this, the proposal is to maintain the architecture, but apply a graph transformation task, some sort of graph reification. Intersectingly, R-GNNs and R^2-GNNs recover the expressive power of similar architectures in the single-relation scenario, when applied not under the original graphs but in the transformed graphs. The second part of the paper looks to implement these ideas specifically in Temporal Graphs, wherein one has a different relation for each timestamp measured in the temporal part. As multi-relational graphs, this is indeed a nice application of the framework. Results and experiments show that the proposed architecture competes with several other proposals for classification of temporal graphs.

**Strengths:**

* Sound, robust theoretical study relating multi-label GNNs with logic. The proofs seem correct and use a variety of techniques.
 *   Claims are backed up with examples, so that it is easy to quickly grasp the ideas of the paper.
 *   The claims in the paper are backed up with experiments: It really does seem that the power of R-GNNs (and similar architectures) benefits from applying the graph transformation.

**Weaknesses:**

*   The results are tailored specifically to a single architecture (R-GNNs). One could think of several other ways to incorporate edge information, I know for instance GATs (Velickovic et al. 2018), but the paper does not discuss any other approach, and we don't know if the weakness shown in the results of section 4 are due to the specific architecture, or if there is a bigger problem underlying multi-relation graphs that must be tackled with the transformation.

*    The proposed solution involves a linear transformation in graphs, but probably demands much more layers in a GNN, as the length of every path is now multiplied by two. This involves adding extra costs to the learning process, and probably some difficulties in using the node embedding of the graph.

*    Synthetics experiments do not compare against other GNN architectures capable of dealing with different edges, appart from R-GNN. Real life experiments do compare against other temporal variants, but it is difficult to see if the added power is due to the global readout, the transformation, or the mix of everything in the architecture proposed.

**Questions:**

*  Please justify why a study on R-GNN is important, and whether your insights could be valuable when dealing with any other architecture incorporating edges.

  *  I'd tend to think that maybe every multi-labelled GNN would benefit from the graph transformation approach, and like wise for the global readout. What would happen if I run GRU-GCN or TGN over a temporal graph that has already been transformed? Or if instead of using R^2-GNN I use any other GNN with the graph transformation?

**Limitations:**

*    Please discuss potential inconveniences regarding using the transformation in graphs. You probably need more layers, with the added cost on training. Also, even if the transformation is linear, it is not free for huge graphs, so your approach is probably better tailored to datasets containing several small graphs.

---

> ### Author Rebuttal · Authors · 2023-08-05
>
> We sincerely thank Reviewer pnMq for the recognition of our theoretical results on both static and temporal settings as well as our proposed novel transformation. Below, we would like to give detailed responses to each of your comments.
>
> ___
> > **Q1**: Please justify why a study on R-GNN is important, and whether your insights could be valuable for other architectures such as GAT.
>
> **A1**: It is worth noting that R-GNN is not a specific model architecture; it is a framework that contains a bunch of different GNN architectures. In the paper, we just said it's generalized from R-GCN [1], but our real goal is to define a generalized framework as an abstraction of most Message-Passing GNNs (MPGNN). We feel sorry for not pointing this out explicitly and perfectly in the paper. In the definitions (line 112 and 118), the functions can be set as any functions you like, such as matrix multiplications or QKV-attentions. Most commonly used GNN such as R-GCN[1] and R-GAT[2] are captured (upper-bounded) within our R-GNN frameworks. Many other related works, such as [3], [4] and [5] also use intrinsically the same frameworks as our R-GNN if you read the definitions, so R-GNN is a kind of abstraction framework for MPGNN that has been approved widely and studied a lot in the community. Therefore, we think analyzing these frameworks leads to common results for many existing GNNs.
>
> In particular, GAT [6] is a kind of single-relation GNN which is not our topic, and its multi-relational variation R-GAT [2] can be incorporated in our R-GNN framework by setting the combination function as QKV-attention module.
>
> ___
> > **Q2**: I think every multi-labelled GNN would benefit from graph transformation, and like wise for the global readout. What would happen if I run GRU-GCN or TGN over a transformed temporal graph? Or using any other GNN with the graph transformation?
>
> **A2**: Intuitively, every (temporal) Message-Passing GNN can be augmented by graph transformation and global readout, if they are within our framework. We've added experiments that test performance of the most recent model GRU-GCN/TGN with graph transformation and R-GAT [2] plus global readout and graph transformation. It can be seen that graph transformation indeed improves performance in most cases. Please check the tables below.
>
> | accuracy| $\varphi_1$ | $\varphi_2$ | $\varphi_3$ | $\varphi_4$ |
> | ------------------------------ | ----------- | ----------- | ----------- | ----------- |
> | R-GAT $\circ H$           | 100         | 61.4        | 88.6        | 82.0        |
> | R-GAT+readout $\circ H$        | 100         | 93.5        | 95.0        | 82.2        |
> | R-GAT+readout $\circ F\circ H$ | **100**         | **98.2**        | **100**         | **95.8**        |
>
> | Models  | GRU-GCN | TGN  | GRU-GCN$\circ F^T$ | TGN$\circ F^T$ |
> | -------- | ------- | ---- | ------------------ | -------------- |
> | Brain-10 | 91.6| 91.2 | 95.0| 94.2|
>
> ___
> > **Q3**: Please discuss potential inconveniences regarding using the transformation in graphs. You probably need more layers, with the added cost on training. Your approach is probably better tailored to datasets containing several small graphs
>
> **A3**: You are right. The complexity of a model depends on its depth (layer number) and width (feature dimension). The graph transformation introduces 1 new unary predicate, so one needs to add one feature dimension, which makes the model slightly bigger but acceptable. On the other hand, if we observe the process of graph transformation as shown in Figure 2 of the paper, we will find that one-hop message passing is stretched to three-hop ones, which may require 3 times the depth to do the same task.
>
> However, in real-world datasets, the scalability of graph transformation is not as weak as imagined. We've added two real-world datasets DGS and AM (suggested by *Reviewer Hbfa*), whose graphs are much bigger in the following table. These results show our method is effective both on small and large graphs.
> | Models | AIFB | MUTAG | DGS  | AM   |
> | ------------------ | ---- | ----- | ---- | ---- |
> |# of nodes|8285|23644|333845|1666764|
> |# of edges|29043|74227|916199|5988321|
> | R-GNN| 91.7 | 76.5  | 81.2 | 89.5 |
> | R$^2$-GNN| 91.7 | 85.3  | 85.5 | 89.9 |
> | R$^2$-GNN$\circ F$ |**97.2**|**88.2**|**88.0**| **91.4** |
>
> ___
> >**Q4**: Synthetics experiments do not compare against other GNN. It is difficult to see if the added power is due to the global readout, the transformation, or the mix of everything in the architecture proposed.
>
> **A4**: We've added R-GAT [2] synthetic experiments in the first table of Answer 1. In fact, as we've mentioned in Answer 1, R-GAT is also an architecture within R-GNN. From the experiments, we can see expressiveness improvements by using graph transformation/global readout on different kinds of equivariant MPGNN.
>
> We've added some experiments shown in the table below to help you better see which components of our method bring added power. These experiments show separate improvements from global readout and graph transformation. As we said in the paper, the drop in the last column may be due to the intrinsic drawbacks of current real-world datasets. Many real-world datasets can not be perfectly modeled as first-order-logic classifier.
>  This non-logical property may lead to less convincing experimental results. As [3]  commented,  these commonly used benchmarks are inadequate for testing advanced GNN variants.
>
> |Models|R-TGNN|R-TGNN$\circ F^T$|R$^2$-TGNN|R$^2$-TGNN$\circ F^T$|
> |--------|------|-----------------|----------|---------------------|
> |Brain-10|85.0|90.9|94.8|94.0|
>
> [1]Modeling relational data with graph convolutional networks ESWC2018
>
> [2]Relational graph attention networks arXiv
>
> [3]The logical expressiveness of graph neural networks ICLR2020
>
> [4]A Theory of Link Prediction via Relational Weisfeiler-Leman arXiv
>
> [5]Logical Expressiveness of Graph Neural Network for Knowledge Graph Reasoning arXiv
>
> [6]Graph Attention Networks ICLR2018

---

> > ### Comment · Reviewer_pnMq · 2023-08-18
> >
> > Thanks for the very detailed answers.
> >
> > I am happy about the additional experiments, and it does indeed show how it can be used in real life. I also acknowledge the fact about more general models, which I can now follow correctly.
> >
> > I'm raising my score as I think the proposal has a much more important contribution now. But I think it is important to separate the idea of the graph transformation with the global readout in the presentation, both are improvements to R-GNNs, that the former helps is less expected (for me) than the latter.

---

> > > ### Author Response · Authors · 2023-08-20
> > >
> > > Thanks for your comment. We are happy to see your confusion resolved and satisfaction on our new experiments. The reason why we  don’t focus on separate graph transformation is that our main motivation is the following: From theorems in Section 3, we’ve seen R-GNN+global readout (R$^2$-GNN) brings some improvement but is still logically weak, so how to theoretically improve its logical expressiveness (to capture $FOC_2$)? Our solution is to use graph transformation and we derive some beautiful theory behind R$^2$-GNN+transformation in Section 4. Graph transformation is certainly orthogonal to model choice. However, if we just use transformation separately, the theoretical logical expressiveness will not be so satisfactory (can’t capture $FOC_2$). We can indeed give theory behind R-GNN+transformation, but it will be just a weaker version of the current Section 4 and all ideas will be similar. Therefore, for brevity we directly analyse the most powerful R$^2$-GNN+transformation and reach our final goal—boost the logical expressiveness to capture $FOC_2$. Of course, it is interesting to observe empirical improvement brought by separate graph transformation. Experiments in Answer 4 show separate graph transformation empirically helps less than global readout. We should add some related discussions in experiment part.

---

### Official Review · Reviewer_KFPi · 2023-07-06

**Soundness:** 3 good
**Presentation:** 2 fair
**Contribution:** 3 good
**Rating:** 7
**Confidence:** 2

**Summary:**

This paper justifies the logical expressiveness of R^2-GNN from the perspective of "universal" graph classes, including multigraphs and infinite graphs. Specifically, the R^2-GNN seems to be identical to the previous ACR-GNN in [1]. The engagement of graph classes makes theoretical statements in finer settings than previous ones in [1] which only involve one graph. The thorough inspection of different graph classes captures the expressiveness power of R^2-GNN compared to logical formulas in FOC_2. Moreover, this paper adapts a commonly used graph transformation $F$ to enhance the expressiveness of R^2-GNN to be stronger than FOC_2 on universal graph classes.

The theoretical results collapse into existing ones in [1] when the graph classes contain finite graphs, which are naturally bounded. The theoretical results can be also extended to temporal graphs under static representations, which are studied in [2]. To achieve this, the authors investigate the expressiveness hierarchy between the collapse function $H$, two static representations, GNN and TGNN (also defined by the author), and graph transformation $F$.

Empirical results support the expressiveness results partially related to $F$, $H$, and TGNN.

[1] Barceló, P., Kostylev, E. V., Monet, M., Pérez, J., Reutter, J., & Silva, J. P. (2020, April). The logical expressiveness of graph neural networks. In 8th International Conference on Learning Representations (ICLR 2020).
[2] Gao, J., & Ribeiro, B. (2021). On the equivalence between temporal and static graph representations for observational predictions. arXiv preprint arXiv:2103.07016.

**Strengths:**

This paper makes a large amount of work to (1) justify the logical expressiveness of R^2-GNN compared to FOC_2 in universal, bounded, and simple graph classes; (2) empirical technics to improve the expressiveness of GNN.

The theoretical results are original. Though some results are still missing to achieve a complete understanding of this problem. The presented ones are significant and inspiring.

**Weaknesses:**

The presentation of this paper should be improved given the dense representation of theorems and proofs.

Firstly, the content is not self-contained. The definition of static representation of temporal graphs is not included even in the appendix.

Most importantly, the key definition to justify logical expressiveness is missing. In Lines 130-131, the author states that the R^2-GNN is the set of all R^2-GNN-based boolean node classifiers. However, it is far from a satisfactory definition, which might be risky for accepting this paper. Please check the question part for my questions about the definition.

AnotherW ambiguous part of this paper is that the graph transformation $F$ for multi-graphs is isolated to FOC_2 classifiers. In fact, $F$ should be orthogonal to classifiers.

**Questions:**

1. What is the definition of one boolean classifier set A as the subset of another boolean classifier set B for a given graph class GC? There are two possible definitions

1. For any a in A, there exists a b(a) in B so that a and b(a) achieve the same results on all G in GC.
2. For any G in GC and any a in A, there exists a b(G, a) in B so that a and b(G, a) achieve the same result on G.

2. $F$ enhanced the power of R^2-GNN classifiers but it never relates the R^2-GNN. Can we state that $F$ also enhances the expressiveness of FOC_2 classifiers? What will Figure 3 be if we include FOC_2 \odot F (or with proper modification of FOC_2 with new predicates introduced)?

3.  For the temporal graph, the author draws two lines of hierarchies of expressiveness. What are the relationships between R^2-TGNN and time-and-graph and R^2-GNN\odot H and time-and-graph?

**Limitations:**

Yes

---

> ### Author Rebuttal · Authors · 2023-08-05
>
> We sincerely thank Reviewer KFPi for acknowledging the novelty and theoretical significance of our findings within multi-relational and temporal context. In particular, we are enthused by your thought-provoking comments, which motivate us to have deeper contemplation. Below, we would like to give detailed responses to each of your comments.
>
> ___
> > **Q1**: What is the definition of one boolean classifier set A as the subset of another boolean classifier set B for a given graph class GC? There are two possible definitions. The definition for R^2-GNN classifiers is not satisfactory
>
> **A1**:  The correct definition is: *For any a in A, there exists a b(a) in B so that a and b(a) achieve the same results on all G in GC.*
>
> A R$^2$-GNN classifier is defined as any classifier that can be represented as a R$^2$-GNN model (outputs node features) equipped with a binary classification function $CLS(x): \mathbb{R}^D\rightarrow 0/1$. (where $x$ means that an input of $CLS()$ is a node feature outputted by the model, and $D$ represents the output feature dimension. $CLS()$ can be any function with the above domain and range.)
>
> Sorry that we did not explain this explicitly. We will include an explicit definition in our revised version.
>
> ___
> >**Q2**: $F$ enhanced the power of $R^2-GNN$ classifiers but it never relates the $R^2-GNN$. Can we state that
>  also enhances the expressiveness of $FOC_2$ classifiers? What will Figure 3 be if we include $FOC_2 \circ F$ (or with proper modification of $FOC_2$ with new predicates introduced)?
>
> **A2**: Your perspective is entirely accurate!  We used the class $FOC_2\circ F$ in line 230-232 as an auxiliary class.
>
> Although R$^2$-GNN can be boosted by $F$, it is not true for $FOC_2$. A counter-intuitive fact is $FOC_2\circ F \subsetneq FOC_2$, which means $FOC_2$ becomes strictly weaker after transformation. Since $FOC_2\circ F$ is just an unimportant bridge class, we don't write the above as a theorem in our paper. We can add the detailed proof in the appendix afterwards. In the following, I will briefly explain it.
>
> First, following the similar idea of proof of Theorem 9 combined with a slightly modified version of Lemma 26 (in the appendix), we can prove $FOC_2\circ F\subseteq FOC_2$. We can't give the specific proof idea due to space constraints, so please refer to our appendix or wait for our detailed proof in later version if interested.
>
> Then, consider the logical classifier $\varphi$ that "classifies a node $v$ to be true iff there are at least 5 nodes that aren't neighbors of $v$". $\varphi$ can be easily represented as a $FOC_2$
>  formula. However, it is impossible to construct an equivalent $FOC_2\circ F$ classifier! Thinking about this example, one may gain some intuition about why the transformation actually makes $FOC_2$ weaker. Combined with the above, we know $FOC_2\circ F \subsetneq FOC_2$.
>
> In fact, we have $FOC_2\circ F= FOC_2$ for any bounded graph class. Moreover, if we introduce a brand new relation type and modify the definition of graph transformation a bit, $FOC_2\circ F=FOC_2$ would be true for arbitrary universal graph classes. However, since this modification will introduce more edges and predicates to our transformation and make $F$ more costly and complex, we give in to the current weaker transformation in our paper. After all, $FOC_2\circ F$ is not important, so we don't have an incentive to boost it.
>
> ___
> > **Q3**: For the temporal graph, the author draws two lines of hierarchies of expressiveness. What are the relationships between R$^2$-TGNN and time-and-graph and R$^2$-GNN$\circ H$ and time-and-graph?
>
> **A3**: First, it must be noted that the original definitions of time-and-graph and time-then-graph only consider localized message-passing. Therefore, in order to compare them fairly with our framework, we have to enable them to do global readout as R$^2$-TGNN do. This is just a slight modification from the original definition of time-and-graph.
>
> Relationships among time-and-graph, R$^2$-TGNN and R$^2$-GNN $\circ H$ are as follows: We can add formal proof in appendix of later version. Here, we just briefly describe some key points.
>
> 1. R$^2$-TGNN  $\nsubseteq$ time-and-graph
>
> That's because time-and-graph can not capture a chain of information that is continuously scattered in time intervals. Specifically, $\varphi(x):=\exists y,\Bigl( r_1^2(x,y)\wedge \bigl(\exists x,r_1^1(y,x)\bigl)\Bigl)$ can't be captured by time-and-graph but is in R$^2$-TGNN.
>
> 2.  time-and-graph $\nsubseteq$ R$^2$-GNN$\circ H$
>
> That's because the static GNN defined in [1] is stronger than R$^2$-GNN. However, Their definition is kind of too ideal and doesn't exactly match most practical models (We can elaborate why this is the case in further discussion if you are interested). That is also why we didn't include a comparison between time-and-graph and R$^2$-TGNN/R$^2$-GNN$\circ H$ in this version of our paper.
>
> 3. Whether time-and-graph $\bigcap$ R$^2$-GNN $\circ H\subseteq$ R$^2$-TGNN ?
>
> Regrettably, we currently do not have a definitive answer to this question, which remains as an open problem. On an intuitive level, however, we are inclined to believe in the validity of *time-and-graph $\bigcap$ R$^2$-GNN $\circ H\subseteq$ R$^2$-TGNN*.
>
> ___
> [1] On the equivalence between temporal and static equivariant graph
> representations ICML2022

---

> > ### Comment · Reviewer_KFPi · 2023-08-12
> >
> > Thanks for your justification. Based on the new results and the improved representation. I will increase the score to 7.

---

> > > ### Author Response · Authors · 2023-08-15
> > >
> > > Dear Reviewer KFPi:
> > >
> > > Thanks again for your constructive review and score increase. We are happy to see your confusion resolved. A kind reminder: Could you please edit your original review to reflect your current evaluation?
> > >
> > > Authors

---

### Official Review · Reviewer_a55p · 2023-07-08

**Soundness:** 4 excellent
**Presentation:** 3 good
**Contribution:** 4 excellent
**Rating:** 8
**Confidence:** 4

**Summary:**

The paper introduces a new connection between the power of the relational GNNs (working on multi-relational graphs) with a class of first-order logic functions. This class of functions named $\mathcal{FOC}_2$ is a subset of first-order logic functions constrained to having only two variables, but includes counting quantifiers. The theoretical analysis considers three classes of multi-relational graphs: universal graph class, bounded size multi-relational graphs, and simple graphs that two nodes can have just one type of connection. Based on these classes paper proves in general none of $R^2-GNN$ and $\mathcal{FOC}_2$ class of functions is a subset of each other, but adding a preprocess to the graph can make a model that is at least as powerful as the union of both classes. They also extend their results to the temporal graphs, where time series graphs can be collapsed into a static graph or considered as separate graphs. The paper also compares two different classes of models, time-and-graph versus time-then-graph, and proves second class is absolutely more powerful. Empirical results on synthetic and real-world graphs have been also conducted for supporting theoretical results.

**Strengths:**

1. Paper for almost all parts is very well-written.
2. Paper finds the exact class of functions that can be learned by most common relational GNNs.
3. Theorems show the theoretical superiority of time-then-graph functions on temporal graphs over the time-and-graph models.
4. Paper finds a hierarchal classification of the power of different models on temporal graphs.
5. Having supporting experiments for showing that theoretical implications have real effects on real-world datasets.


**Weaknesses:**

1. Intuition behind the functions used for synthetic graphs has not quite been explained. A little more explanation on why these functions and if they are just a random selection or one of the most simple functions that could convey a point will be helpful.
2. The paper relies a lot on definitions of time-then-graph and time-and-graph models and refers to previous work for their definitions. These definitions are better to be briefly explained at least as a section in the Appendix.
3. Section 6 starts with the Experiments and it is hard at first to understand that all synthetic are temporal. Particularly, superscripts on the relations were confusing at first without knowing they are temporal indices. It would have been better to clearly indicate that before starting and also include synthetic experiments for the general multi-relational graphs part.
4. Real-world experiment on the Brain-10 dataset is a little inconsistent with theoretical results.

Minor errors:
1. Inconsistent use of $FOC$ and $\mathcal{FOC}$ in the notations.
2. Line 110, a space missing between "vectors" and "$\mathbf{x}_v^{(i)}$".
3. Line 339, "somorphism" -> "isomorphism"


**Questions:**

1. Node-level binary classification tasks are limited in the real world. Theoretical results are insightful, however, I was wondering if these results can imply anything about any other type of tasks on the graphs, e.g. multi-class node classifications or graph-level tasks.
2. In Corollaries 4.1 and 9.1, it is interesting to see that both classes have precisely same power. However, this power seems to appear because of $\mathcal{FOC}_2$ being able to enumerate all possible combinations of relation counts. Is there any complexity measurement, e.g. time-complexity of evaluating a function on a graph, on how complex can the function under $\mathcal{FOC}_2$ be as the bound on the number of nodes in the graph or number of possible relations in the class increases?



**Limitations:**

The paper clearly defines the class of functions analysis. There are limitations on what types of graphs and tasks theoretical results work, but these limitations are clearly indicated in the paper.

---

> ### Author Rebuttal · Authors · 2023-08-05
>
> We sincerely thank Reviewer a55p for acknowledging the merits of our paper in both theoretical and experimental aspects. In particular, we thank you for your interesting and insightful reviews, which motivated us to formally prove and add the following results (see below). We will add these results with rigorous proof in a later version of the paper.
>
> ___
> > **Q1**: Node-level binary classification tasks are limited in the real world. Theoretical results are insightful, however, I was wondering if these results can imply anything about any other type of tasks on the graphs, e.g. multi-class node classifications or graph-level tasks.
>
>
> **A1**: In general, for a multi-class node classification task with $n$ labels, we can reduce it to $\lceil \log{n} \rceil$ separate binary node classification tasks by predicting each binary bit of label's index. A more natural way is to just tackle it as $n$ different binary node classifications. Both of the two ways described above may lead to a generalization of this paper to multi-class scenario. However, we don't know how to directly model a multi-class classifier as a unified logical classifier. This may be an interesting future direction.
>
> ___
> > **Q2**: In Corollaries 4.1 and 9.1, it is interesting to see that both classes have precisely same power. However, this power seems to appear because of  $FOC_2$
>  being able to enumerate all possible combinations of relation counts. Is there any complexity measurement, e.g. time-complexity of evaluating a function on a graph, on how complex can the function under $FOC_2$
>  be as the bound on the number of nodes in the graph or number of possible relations in the class increases?
>
> **A2**: We can indeed get a (rather loose) bound: Suppose there are $P$ unary predicates and $R$ relation types, and we are considering a bounded graph class with no more than $N$ nodes. For any classifier $c$, suppose $c$ can be represented as an R$^2$-GNN with depth (layer number) $L$. Then there is a $FOC_2$ classifier $\varphi$ equivalent to $c$ such that the following bounds hold:
>
> - The quantifier depth of $\varphi$ is no more than $L$.
>
> - The size of $\varphi$ (size of parse tree) is no more than $2^{f(L)}$, where $f(L)=2^{2^{2Nf(L-1)}}, f(0)=2^{2^{2(P+R)}}$.
>
> The key idea is the following: First, by Lemma 25 in our appendix, $c$ can be represented as a $FOC_2$ formula $\varphi$ with quantifier depth no more than $L$. Then by Proposition 24 in our appendix (This is a key point of this bound; please refer to our appendix). We know the number of intrinsically different bounded-depth $FOC_2$ formulas is finite, so we only need to get an upper bound on this number. Finally, we can get the bound by iteratively using the fact that a boolean combination of a set of formulas can always be written as DNF (disjunctive normal form). The tower of power of two comes from L rounds of DNF enumerations. It is a rather loose bound. We can formulate it in the appendix in later version.
>
> ___
> >**Q3**: Real-world experiment on the Brain-10 dataset is a little inconsistent with theoretical results.
>
> **A3** : We think maybe that's because our real-world dataset has the following two drawbacks when used as logical expressiveness benchmark.
>
> - its labels cannot be modeled as first-order-logic classifier. For example, maybe two isomorphic graphs (nodes) have different labels in the dataset. This negative fact about real-world datasets has also been observed and illustrated in Gao and Ribeiro's work (Figure 6 of [1]). As a result, permutation-equivariant GNNs can not get correct answers, and transformation increases the chaos. It means the real-world dataset may contain non-logical rules.
>
> - The intrinsic logical rules in the dataset are too complicated. In this scenario, maybe the transformation sometimes makes these rules more complex. That's because the transformation changes the predicate set and graph pattern. As a result, the transformed intrinsic logical rules become too complicated to capture by our model with bounded size.
>
> As we mentioned in the paper, [2] and [3] also observe the phenomenon, and they think these commonly used benchmarks are inadequate for testing advanced GNN variants.
>
> ___
> > **Q4**: Intuition behind the functions used for synthetic graphs has not quite been explained. A little more explanation on why these functions and if they are just a random selection or one of the most simple functions that could convey a point will be helpful.
> The paper relies a lot on definitions of time-then-graph and time-and-graph models and refers to previous work for their definitions. These definitions are better to be briefly explained at least as a section in the Appendix.
> Section 6 starts with the Experiments and it is hard at first to understand that all synthetic are temporal. Particularly, superscripts on the relations were confusing at first without knowing they are temporal indices. It would have been better to clearly indicate that before starting and also include synthetic experiments for the general multi-relational graphs part.
>
>
> **A4** : Thanks for pointing it out! We've presented an explanation from line 308-316. Due to space constraints, the explanation may be too brief to be perfectly understood. We will revise this part in later version. Other suggestions on writing will be considered, too.
>
> ___
> [1]On the equivalence between temporal and static equivariant graph representations ICML2022
>
> [2]The logical expressiveness of graph neural networks ICLR2020
>
> [3]Are powerful graph neural nets necessary? a dissection on graph classification arXiv

---

> > ### Comment · Reviewer_a55p · 2023-08-15
> >
> > Thanks for your thorough responses and I am thrilled to see that my questions have sparked new ideas to enhance the paper's theory. I am persuaded by the answers provided, and I'll maintain my current score.

---

### Official Review · Reviewer_Yror · 2023-07-23

**Soundness:** 4 excellent
**Presentation:** 3 good
**Contribution:** 3 good
**Rating:** 7
**Confidence:** 4

**Summary:**

The paper extends the logical characterization of GNN --- node classification, by investigating the relational case i.e. multi-relational graphs. The paper generalizes the work of Barcelo et al. [ICLR 2020], who provided the logical characterization of GNNs for node classification in terms of First Order Logic with two variables and counting quantifiers (FOC2). The paper discuss the R2-GNN model, that extends R-GNN --- a GNN architecture for multi-relational graphs --- with a global read out function. The global readout adds aggregation over node features to the combination function in each layer. Some of the key results of the paper are the following:

- R2-GNN are not captured in FOC2 and FOC2 is not captured by R2-GNN (for unbounded graphs)
- R2-GNN are  captured in FOC2 for bounded graphs (given a known upper-bound on the number of nodes) but not vice versa
- R2-GNN = FOC2 for simple graphs

The paper then extends the expressivity of R2-GNNs to full FOC2 , by introducing a transformation F that runs in linear time w.r.t the multi-graph size (O(|V|+|E|)).

Finally, the authors extend there framework to temporal graphs. This is achieved by simply adding additional predicates indexed by time. The authors then discuss the expressivity of this framework with the previously introduced transformation  F applied to each time stamp  and another transformation comprising union of each time stamped graph.

Post-rebuttal: I have read author's rebuttal, and I am more confident about my rating and will keep it, i.e. acceptance.

**Strengths:**

The paper extends logical characterization of GNNs in an interesting direction i.e. over relational structures. The theory presented is rather intuitive and the authors try to convey the key proof ideas. They also exploit their theoretical investigation to expand the expressivity of GNNs over relational structures and temporal graphs. They  provide convincing experiments (although slightly redundant, see weaknesses) over synthetic and real-world data.

**Weaknesses:**

- Quite terse for non-experts
- Comparison of all the aggregation functions in tables are not much discussed in the paper and are not really important to the main message of the paper. Maybe this could be removed/shortened to give space for more proof intuitions?

**Questions:**

When using many predicates, it seems that one could encounter some form of curse of dimensionality --- Do you think R2-GNNs suffer such issues? Specially in the temporal/transformation case, when number of predicates are further increased.

In general, some discussion on how the complexity of learning changes with number of predicates could be interesting

**Limitations:**

The authors discuss the fact that R2-TGNN*F^{T} do not improve performance over R2-TGNN on real-world dataset, whereas on their synthetic dataset they do observe improvement. This ambivalence between theory and practice is quite interesting and further discussion of this--- as to what exactly is different in real-world data that makes more expressive models inferior --- could be very interesting

---

> ### Author Rebuttal · Authors · 2023-08-05
>
> We sincerely thank Reviewer Yror for the recognition of our original and theoretical results analyzing the expressivity of graph neural networks in the multi-relational and temporal setting. Below, we would like to give detailed responses to each of your comments.
>
> > **Q1**: When using many predicates, it seems that one could encounter some form of curse of dimensionality --- Do you think R2-GNNs suffer such issues? Specially in the temporal/transformation case, when number of predicates are further increased. In general, some discussion on how the complexity of learning changes with number of predicates could be interesting
>
> **A1**: Yes. The more predicates there are, the more dimensions are needed in the feature initialization function $I()$ defined in line 103. However, it also means the graph itself is more complex, and reasoning on it requires heavier logic formulas. In some sense, dimensionality is a necessary sacrifice when dealing with graphs under huge semantic systems. In fact, one can indeed avoid this sacrifice by changing the initialization function $I()$. As an extreme example, one can define $I()$ as $I(v):=\prod_{i}{p_i^{c_i}}$ where $p_i$ is the $i-$th prime and $c_i$ is a $0/1$ that indicates whether node $v$ satisfies the $i-$th predicate. This initialization function also distinguishes nodes with different properties but outputs only one-dimension-features, no matter how large the predicate set is! However, we think these settings are unnatural and inapplicable for practical GNN models, so we don't use them in our paper. However, it might be worth noting that some theoretical work, such as the proof of Theorem5.2 of [1] indeed uses such settings for theoretical analysis convenience.
>
> Graph transformation will only introduce $3$ new predicates, so I think this additional cost is rather small and acceptable.
>
> In temporal case, if one uses collapse function H (definition 11 in the paper) to transform a temporal graph to a static graph and then run static R$^2$-GNN on it, this truly introduces $|P|\times T$ predicates and thus huge dimensionality, where $P$ is the unary predicate set and $T$ is the number of timestamps. However, R$^2$-TGNN won't introduce new predicates. It just runs different GNN model instances on different timestamps while preserving the original predicate set on each timestamp. Therefore, R$^2$-TGNN has a size proportional to the timestamp number $T$, but its dimensionality is still $|P|$. Temporal graph transformation $F^T$ (definition 12 in the paper) only introduces $3$ new predicates. In conclusion, R$^2$-TGNN$\circ F^T$ introduces $3$ new predicates in total, which is acceptable. In Theorem 14 of our paper we've proven its superior expressiveness, so there is no need to use the costly $H$ transformation in temporal graphs.
>
> > **Q2**: The authors discuss the fact that R$^2$-TGNN$\circ F^{T}$ do not improve performance over R$^2$-TGNN on real-world dataset, whereas on their synthetic dataset they do observe improvement. This ambivalence between theory and practice is quite interesting and further discussion of this--- as to what exactly is different in real-world data that makes more expressive models inferior --- could be very interesting
>
> **A2**: We think maybe that's because our real-world dataset has the following two drawbacks when used as a logical expressiveness benchmark.
>
>  1. Its labels cannot be modeled as a first-order-logic classifier. For example, maybe two isomorphic graphs (nodes) have different labels in the dataset. This negative fact about real-world datasets has also been observed and illustrated in Gao and Ribeiro's work (Figure 6 of [2]). As a result, permutation-equivariant GNNs can not get correct answers, and transformation increases the chaos. It means the real-world dataset may contain non-logical rules.
>
> 2. The intrinsic logical rules in the datasets are too complicated. In this scenario, maybe the transformation sometimes makes these rules more complex. That's because the transformation changes the predicate set and graph pattern. As a result, the transformed intrinsic logical rules become too complicated to capture by our model with bounded size.
>
> As we mentioned in the paper, [1] and [3] also observe the phenomenon. Their comments also say these commonly used benchmarks may be inadequate for testing advanced GNN variants.
>
> > **Q3**: Comparison among different aggregation functions should be removed/shortened to give space for more proof intuitions__
>
>  **A3**: Thanks for your suggestions. We will consider your suggestions and add some more necessary proof intuition in our revised version.
>
> [1]The logical expressiveness of graph neural networks ICLR2020
>
> [2]On the equivalence between temporal and static equivariant graph representations ICML2022
>
> [3]Are powerful graph neural nets necessary? a dissection on graph classification arXiv

---

### Official Review · Reviewer_p5D8 · 2023-07-24

**Soundness:** 4 excellent
**Presentation:** 4 excellent
**Contribution:** 3 good
**Rating:** 7
**Confidence:** 4

**Summary:**

In this paper the authors provide a theoretical analysis of graph neural network (GNN) expressivity on multi-relational and temporal graphs. Specifically, they extend the multi-relational graph convolution network R-GCN to a class of architectures they call R-GNN, which allows for arbitrary aggregation (pooling neighbors with respect to a particular relation) and combination functions (pooling these aggregations with the node representation itself). They further extend this to a class of architectures they call R²-GNN, which adds a global "readout" function which aggregates the feature vectors of all nodes in the graph. They analyze the capability of R²-GNN to classify nodes and compare it to that of $\mathcal{FOC}_2$, a restricted subset of first-order logic which only allows formulas with at most 2 variables but also allows counting quantifiers. They find that neither is a subset of the other, in particular R²-GNN is not able to distinguish whether two neighbors of a given node with different relations are, in fact, the same node. However by first augmenting the graph with a particular transformation $F$ they are able to overcome this restriction, and ultimately show that $R^2-GNNs \circ F = \mathcal{FOC}_2$ on any graph class with a bounded number of nodes.

All of this is then extended to temporal graphs, which can also be converted to a multi-relational graph (with a unique set of relations per time-step). A hierarchy of expressivity with respect to current temporal graph frameworks is presented. Finally, experimental results validating the theoretical expressivity on synthetic and real-world datasets are presented.

**Strengths:**

The paper is, to the best of my knowledge, original, and theoretical results analyzing the expressivity of graph neural networks in the multi-relational setting are certainly of significant interest to the graph neural network community.

The work is of a high quality, and presented clearly. Despite the highly technical nature of the results, the authors do a good job in motivating their work and providing intuition for the proofs.

**Weaknesses:**

In general, a point of caution with any strongly theoretical work is whether it applies in practice. In this setting the largest concern comes from the graph transformations. It is not unusual for a graph transformation to extend capabilities in the way the authors have done. A classic example is to extend a method which works on undirected graphs to the setting of directed graphs by creating a new graph with twice as many nodes, one representing "head" and "tail" for each node, however this can make some relationships (eg. transitivity) very difficult to observe. In the author's setting, while the graph augmentation may afford the ability to prove theoretical expressivity results, one might be concerned that the resulting graph is augmented to the point that a GNN architecture may require greater depth to perform the same sort of tasks it did previously. Of course the authors have provided empirical evidence that this augmentation is not problematic in at least three graphs, which is quite reasonable to support their approach, however it is something to bear in mind.

**Questions:**

1. What is the initialization used for the nodes in $V' \setminus V$? To the best I can tell, they do not have any unary predicates, so they would seem to be initialized with a vector of all zeros.
2. Not a question as much as a comment - the R²-GNN architecture is equivalent to the R-GNN architecture if we first augment the graph to have a new binary relation which forms a complete graph on all nodes. Given that the work already includes a graph augmentation step, an alternative presentation could potentially avoid the R²-GNN architecture altogether, and simply express the entire thing as R-GNN on an augmented graph. I'm not sure it is clearer to take this approach, but it might be worth noting.

Most of the paper was very well written, but I did find the following typos (mostly in the intro):
L37: authors -> the authors
L45: authors -> the authors
L46: in -> in the
L47: edges in graphs -> edge in the graph
L48: of -> (delete)
L48: graph needs -> graphs need
L51: as -> as a
L57: be -> can be
L60: to -> to the
L60: Besides -> Moreover
L61: extensively -> extensively in
L69: in -> in the
L70: power -> the power
L71: such -> such a
L72: mutli -> multi
L91: to -> to denote
L96: node Boolean -> Boolean node
L111: vectorsx -> vectors x
L338: do somorphism -> perform an isomorphism

A minor suggestion: move the sentence in lines 170-171 to around line 85, to motivate the definition of bounded and simple graph classes.

**Limitations:**

I did not see much discussion of the limitations of the proposed model. Some of the concerns outlined in the weaknesses section above are good candidates for this.

---

> ### Author Rebuttal · Authors · 2023-08-05
>
> We sincerely thank Reviewer p5D8 for the recognition of our original and theoretical results analyzing the expressivity of graph neural networks in the multi-relational setting as well as their potential contribution to the graph neural network community. Below, we would like to give detailed responses to each of your comments.
>
> ___
> > **Q1**: What is the initialization used for the nodes in $V' \backslash V$. To the best I can tell, they do not have any unary predicates, so they would seem to be initialized with a vector of all zeros.
>
> **A1**: Yes, your understanding is right. These new nodes are just constructed as auxiliary "bridges", so they don't own any substantial properties themselves. Their node features become meaningful only after messages from primal nodes pass to them.
>
> ___
> > **Q2**: In general, a point of caution with any strongly theoretical work is whether it applies in practice. In this setting, the largest concern comes from the graph transformations. It is not unusual for a graph transformation to extend capabilities in the way the authors have done. A classic example is to extend a method that works on undirected graphs to the setting of directed graphs by creating a new graph with twice as many nodes, one representing "head" and "tail" for each node, however this can make some relationships (eg. transitivity) very difficult to observe. In the author's setting, while the graph augmentation may afford the ability to prove theoretical expressivity results, one might be concerned that the resulting graph is augmented to the point that a GNN architecture may require greater depth to perform the same sort of tasks it did previously. Of course the authors have provided empirical evidence that this augmentation is not problematic in at least three graphs, which is quite reasonable to support their approach, however it is something to bear in mind.
>
> **A2**: Indeed, your observation is on point. In essence, a model's complexity is contingent upon its depth (number of layers) and width (feature dimensions). With the incorporation of graph transformation, a single new unary predicate is introduced, thereby necessitating an additional feature dimension. This augmentation contributes to a minor increase in the model's size, which remains quite acceptable. However, a more intriguing aspect surfaces when we delve into the process of graph transformation, depicted in Figure 2 of the paper. Here, the progression from one-hop to three-hop message passing elongates the process, potentially demanding a tripling of the model's depth to achieve similar task outcomes. Furthermore, the graph transformation introduces two novel relation types, although this aspect is generally less consequential in terms of complexity.
>
> ___
>
> > **Q3**: Typos and the suggested adjustment of the paper structure.
>
> **A3**: We greatly appreciate your feedback regarding the identified typos and your valuable suggestion for enhancing the readability of our paper. We will address these aspects diligently in our revised version.

---

> > ### Comment · Reviewer_p5D8 · 2023-08-20
> > **Thank you and brief follow-up**
> >
> > Thanks for your clarifications.
> >
> > Could you provide any comment related to my comment related to the equivalent interpretation of R²-GNN as R-GNN with an augmentation? I have copied it below:
> >
> > > Not a question as much as a comment - the R²-GNN architecture is equivalent to the R-GNN architecture if we first augment the graph to have a new binary relation which forms a complete graph on all nodes. Given that the work already includes a graph augmentation step, an alternative presentation could potentially avoid the R²-GNN architecture altogether, and simply express the entire thing as R-GNN on an augmented graph. I'm not sure it is clearer to take this approach, but it might be worth noting.
> >
> > Is my interpretation here correct? If so, would it have been perhaps simpler to present the entire thing as simply R-GNN with a graph augmentation?
> >
> > As some feedback on the comments from (at this time) the remaining reviewer providing a negative score:
> > Reviewer auoR has some concerns with the term "multi-relational" graph, and proposes to use "real world graph" or just "graph". The term "multi-relational" is quite standard in the literature, and I see no particular issue with the author's definition of it. Using "real world graph" would be highly ambiguous, and using just "graph" would be incorrect, as a graph is a tuple $G = (V, E)$ where $E \subseteq V \times V$ but a multi-relational graph is a triple $G' = (V, E, R)$ where $E \subseteq V \times R \times V$. The authors here have chosen to split out binary and unary predicates, which is fairly standard and can be shown to be isomorphic to the multi-relational structure presented here.

---

> > > ### Author Response · Authors · 2023-08-20
> > >
> > > Thanks for your reply. Sorry that we missed one of your issue since we previously misunderstood it in some sense.
> > >
> > > Your alternative augmentation is correct, but not efficient. Suppose there are $n$ nodes, adding a new complete graph would require $n^2$ edges, which is too costly in many cases. Additionally, it may cause more cost after combined with graph transformation.
> > >
> > > A more efficient way is the following: We first add a brand new node called ‘Agg’ (abbreviation of aggregation). Then we only need to connect all original nodes to ‘Agg’ with a special new binary relation. This method     only introduces $n$ new edges, which is linear. The reason why it works is that you can first integrate messages from all original nodes to ‘Agg’ by this new relation. Then in the next round (layer) you distribute the integrated messages from ‘Agg’ to all original nodes. This two rounds message-passing is equivalent to one-round global readout. In fact, it is almost the same as the standard technic used in definition of WL-test (Please refer to page 31 of [1] if you are interested). However, you may notice that this alternative method may require more depth or feature dimension to do the same task. In this work our theory doesn’t focus on depth and feature dimension so it’s fine, but empirical performances of this method and the current method may differ due to depth/dimension requirement.
> > >
> > > As for brevity of presentation, there are two possible use of ‘the new relation augmentation’ (abbreviated as augmentation in the following).
> > >
> > > 1. If we combine the augmentation and graph transformation in the presentation, our original motivation may be hidden: R$^2$-GNN has been studied on single-relation graphs in [2], and they prove R$^2$-GNN is very powerful in single-relation scenario where it can capture $FOC_2$. However, we find this inclusion relationship fails in multi-relation scenario. That’s why we want to calibrate and boost its expressiveness in this new scenario. Therefore, as we want to illustrate our original motivation, we choose to define R$^2$-GNN separately and show its failure in multi-relational graphs, which can’t be combined with the next step. Another reason is that we think calibration of expressiveness of R$^2$-GNN in multi-relational graphs (Section 3) counts for one contribution. We won’t have a chance to show this if we combine augmentation (alternative of R$^2$-GNN) and graph transformation.
> > > 2. In our opinion, if we separate augmentation from graph transformation and only use it as an alternative of R$^2$-GNN, the paper length won’t change much: We simply write a short paragraph to describe transition from R-GNN to R$^2$-GNN. If we use augmentation separately, it would be like replacing all line 116-123 with definition of augmentation. Other parts would remain the almost same.
> > >
> > > Still, we are truly thankful for your interest in our work, as well as the careful reading and review with your own thoughts.
> > >
> > >
> > > [1] Sandra Kiefer. Power and limits of the Weisfeiler-Leman algorithm. PhD thesis, Dissertation, RWTH Aachen University, 2020
> > >
> > > [2] The logical expressiveness of graph neural networks ICLR2020

---

### Official Review · Reviewer_Hbfa · 2023-07-25

**Soundness:** 3 good
**Presentation:** 3 good
**Contribution:** 3 good
**Rating:** 6
**Confidence:** 3

**Summary:**

This work calibrate the logic expressiveness of R2-GNNs as node classifiers on multi-relational graphs. Motivated by some negative results, authors boost R2-GNNs with a graph transformation, which enables R2-GNN to capture FOC2 formulas. They further extend expressiveness results and graph transformation to temporal settings and derive an expressiveness hierarchy of temporal GNNs.

**Strengths:**

1. Clear theoretical results on static and temporal settings.

2. Proposed transformation is straightforward and scalable.

**Weaknesses:**

1. Nearly no related work. There is no related work section. Only the closest related work is mentioned in introduction and some related works are missing. For example, besides [1], some works also analyze GNN's expressivity as boolean classifier. Weifeiler-Leman test have been connected to boolean graph classifier for a long time [2]. [3] also analyze boolean link classifier.

2. The experiments are only conducted on small datasets. You can use larger DGS and AM in RGCN paper [4].


[1] Pablo Barceló, Egor V. Kostylev, Mikael Monet, Jorge Pérez, Juan Reutter, and Juan Pablo Silva. The logical expressiveness of graph neural networks. ICLR, 2020

[2] Jin-yi Cai, Martin Fürer, Neil Immerman, An Optimal Lower Bound on the Number of Variables for Graph Identification. FOCS 1989: 612-617.

[3] Xingyue Huang, Miguel A. Romero Orth, Ismail Ilkan Ceylan, Pablo Barceló, A Theory of Link Prediction via Relational Weisfeiler-Leman. CoRR abs/2302.02209 (2023).

[4] Michael Sejr Schlichtkrull, Thomas N. Kipf, Peter Bloem, Rianne van den Berg, Ivan Titov, Max Welling: Modeling Relational Data with Graph Convolutional Networks. ESWC 2018.

**Questions:**

1. There is no scalability comparison between the proposed method and baselines. Please add it.

**Limitations:**

Limitation is not addressed.

---

> ### Author Rebuttal · Authors · 2023-08-05
>
> We sincerely thank Reviewer Hbfa for the recognition of our theoretical results on both static and temporal settings as well as our proposed novel transformation. Below, we would like to give detailed responses to each of your comments.
> ___
> > **Q1**: There is no scalability comparison between the proposed method and baselines. Please add it. The experiments are only conducted on small datasets. You can use larger DGS and AM in RGCN paper.
>
> **A1**:  We have followed your advice and tested our method and baselines on larger datasets DGS and AM. The results on the two bigger datasets are as follows. From the results, we can see that R$^2$-GNN$\circ F$  outperforms the other baselines, which confirms the scalability of our method.
>
> |              | AIFB | MUTAG | DGS  | AM   |
> | ------------------ | ---- | ----- | ---- | ---- |
> |# of nodes|8285|23644|333845|1666764|
> |# of edges|29043|74227|916199|5988321|
> | R-GNN              | 91.7 | 76.5  | 81.2 | 89.5 |
> | R$^2$-GNN          | 91.7 | 85.3  | 85.5 | 89.9 |
> | R$^2$-GNN$\circ F$ | **97.2** | **88.2**  | **88.0** | **91.4** |
> | R-GCN              | 95.8 | 73.2  | 83.1 | 89.3 |
> | R-GAT              | 96.9 | 74.4  | 86.9 | 90.0 |
>
>
> > **Q2**: Nearly no related work. There is no related work section. Only the closest related work is mentioned in introduction and some related works are missing. For example, besides [1], some works also analyze GNN's expressivity as boolean classifier. Weifeiler-Leman test have been connected to boolean graph classifier for a long time [2]. [3] also analyze boolean link classifier.
>
> **A2**: We didn't have a separate section for related works in this preliminary version due to space constraints. It will be added in later version. Thanks for your proposal.
>
> It might be worth noting that [2] focuses more on distinguishing ability, which means whether a logic segment can distinguish two graphs. It is slightly different from classification ability. Specifically, consider that we have a graph (node) property $A$, and a set of logical classifiers $B$ (such as $B=FOC_2$). Distinguishing ability of $B$ over $A$ just guarantees that for each graph (node) $G$ with $A$ and each graph (node) $H$ without $A$, there exists a formula $\varphi_{G,H}\in B$ such that $G$ and $H$ get different labels on $\varphi_{G,H}$. However, Classification ability requires stronger expressiveness in the sense that classification ability of $B$ over $A$ means that there exists a **fixed** formula $\varphi\in B$, such that for every graph (node) $G$ with $A$ and every graph (node) $H$ without $A$, it satisfies $G\models \varphi, H\models \neg \varphi$. This difference appears often in related theoretical analysis, and it has been implicitly and briefly mentioned in the introduction of this preliminary version.
>
> [1]The logical expressiveness of graph neural networks ICLR2020
>
> [2]An Optimal Lower Bound on the Number of Variables for Graph Identification FOCS1989
>
> [3]A Theory of Link Prediction via Relational Weisfeiler-Leman arXiv

---

> > ### Comment · Reviewer_Hbfa · 2023-08-10
> >
> > Thank you for the detailed reply. It solves my concerns. I am willing to raise my score to 6.

---

### Official Review · Reviewer_auoR · 2023-07-25

**Soundness:** 3 good
**Presentation:** 2 fair
**Contribution:** 1 poor
**Rating:** 4
**Confidence:** 3

**Summary:**

This paper proposes the R^2-GNN that captures the logic features in the graph. The proposed methodology largely resembles the previous work of ACR-GNN, where a GNN variant is proposed to capture the logic feature for graphs with no relation types. The proposed R^2-GNN extends the previous method by considering the graphs that involve a set of unary and binary predicates and potentially temporal information. To do so, the authors apply the same readout function to the base R-GNN model and show that it can capture the full logic classifiers with an additional graph transformation F.  Also an temporal variant R^2-TGNN is proposed to handle the temporal graphs.

**Strengths:**

The paper is overall easy to read, however, I didn't check all the proof in detail.

**Weaknesses:**

## Novelty

As mentioned in summary, the authors follow the same method in ACR-GNN in modifying R-GNN into R^2-GNN to enable the model to capture the FOC_2 logic family. It seems to me the main methodological differences are the additional graph transformation F and the discussion on temporal graphs.
- The former helps the model to distinguish nodes with different relations, while it seems to work empirically, it is more or less an incremental modification to the framework.
- For the temporal graphs, the author propose to "temporalize" the graph by mapping the graph snapshots into one single static graph and process it with a slightly modified model, namely R^2-TGNN.

That said, this paper is not very novel methodology-wise.


## Quality

The definition of a multi-relational graph is more or less redundant and confusing: this is effectively the graph with both unary and binary predicates, and many existing KGs already contain facts of both these types. One can refer to it as a real-world graph or just an ordinary graph instead of as "multi-relational".

Some model and design choices lack justification:
- It is unclear why the authors pick the R-GNN as the base model, provided that: (1) most existing GNNs can handle graphs with different relations and (2) the proposed modifications such as readout function and transformation F and H are orthogonal to the choice of the model. The authors may want to consider evaluating their method on different base models empirically to better demonstrate the difference
- It is also unclear why the task is limited to node classification only: this leads to a narrow choice of the benchmarks in the experiments (L329-L330) and leaves some popular datasets untested such as GDELT and ICEWS.

I'm also concerned about the real-world datasets experiments. The authors show the R^2-(T)RGNN achieves better performance than other GNN variants. While empirically this does have some merits, it does not provide any insights into the proposed model. Since the model focuses on learning the latent FOC_2 classifiers. One should first inspect the datasets and see what are the learnable FOC_2 rules in there. And if the model performs better, the authors should take the opportunity and analyze if this indeed resulted from the extra capabilities introduced to the model.


## Clarity

The paper is overall easy to read, however, I didn't check all the proof in detail.


## Significance


Apart from the lack of novelty, this paper also seems to lack significance. The FOC_2 logic is a small subset of first-order logic. Being able to capture these classifiers leads to minor benefits to the GNN literature: it does not provide sufficient explanation of the GNN model, nor does it tackle the difficult logical reasoning problems on graphs such as multi-hop reasoning, induction, and so on. In the experiments, the model is tested on only synthetic and three real-world datasets, and with no comparison to the SOTA models, so the empirical significance is also unclear.

**Questions:**

See above

**Limitations:**

Yes

---

> ### Author Rebuttal · Authors · 2023-08-05
>
> We sincerely thank Reviewer auoR for the positive feedback and insightful comments. Below, we would like to give detailed responses to each of your comments.
>
> > **Q1**: Why the authors pick the R-GNN as the base model. The authors should evaluate their method on different base models.
>
> **A1**: It is worth noting that R-GNN is not a specific model architecture; it is a framework that contains a bunch of different GNN architectures. In the paper, we just said it's generalized from R-GCN [1], but our real goal is to define a generalized framework as an abstraction of most Message-Passing GNNs (MPGNN). We feel sorry for not pointing this out explicitly and perfectly in the paper. In the definitions (line 112 and 118), the functions can be set as any functions you like, such as matrix multiplications or QKV-attentions. Most commonly used GNN such as R-GCN[1] and R-GAT[2] are captured (upper-bounded) within our R-GNN frameworks. Many other related works, such as [3], [4] and [5] also use intrinsically the same frameworks as our R-GNN if you read the definitions, so R-GNN is a kind of abstraction framework for MPGNN that has been approved widely and studied a lot in the community. Therefore, we think analyzing these frameworks leads to common results for many existing GNNs.
>
> We've added a series of experiments that use R-GAT [2] as base model. The following results show the generality of our results on different base models within the framework. (Here, We'd like to **re-emphasize** that R-GAT is actually another specific architecture under R-GNN.)
>
> |  $FOC_2$ classifier  | $\varphi_1$ | $\varphi_2$ | $\varphi_3$ | $\varphi_4$ |
> | ------------------------------ | ----------- | ----------- | ----------- | ----------- |
> |R-GAT $\circ H$|100|61.4|88.6|82.0|
> | R-GAT+readout $\circ H$| 100| 93.5        | 95.0        | 82.2        |
> | R-GAT+readout $\circ F\circ H$ | **100**         | **98.2**        | **100**         | **95.8**  |
>
> > **Q2**: It is also unclear why the task is limited to node classification only
>
> **A2**: We consider it as a future work because we are are not sure whether these results can be generalized to other tasks such as link prediction.
>
>  Take link prediction as an example. If one wishes to define logics for link prediction, the considered logical formulas have to contain two free variables rather than one in node classification. It may influence some of our theoretical results, but maybe we can use graph transformation to reduce the two-variable case to some one-variable case. We need further research to check whether it is true, and whether our results can be generalized.
>
> > **Q3**: We focuses on learning the latent $FOC_2$ classifiers. One should first inspect the datasets and see what are the learnable $FOC_2$ rules in there.
>
> **A3**: Many real-world datasets may not be perfectly modeled as first-order-logic classifiers. Sometimes two isomorphic graphs (nodes) have different labels in the dataset. This negative fact about real-world datasets has also been observed and illustrated in [6] (Figure 6).
>
> Even if these datasets can be logically modeled, the challenge lies in extracting intricate latent logical rules from real-world datasets. This very challenge underscores the necessity of employing both synthetic and real-world datasets in our experimentation, as elaborated in this paper. Synthetic datasets serve to showcase enhanced logical expressiveness, whereas real-world datasets demonstrate the tangible performance enhancements stemming from this heightened expressiveness.
>
> Considering the intricacy of the datasets, we hold reservations about the efficacy of existing logical extraction algorithms, such as those outlined in [7], for our specific objectives. Given the dataset's complexity, a pragmatic approach seems more viable. We envisage a dataset that not only originates from real-world contexts but also employs explicit logical classifiers. Regrettably, our search for such a benchmark dataset remains unfruitful.
>
> > **Q4**: Being able to capture $FOC_2$ leads to minor benefits to the GNN literature. the model is tested on only synthetic and three real-world datasets, and with no comparison to the SOTA models
>
> **A4**:  While $FOC_2$ is a weak fragment within the realm of first-order logic, our **impossibility result** as exemplified in Proposition 2 has shown that R$^2$-GNN, a rather powerful framework, remains **inadequate** in capturing the entirety of $FOC_2$. Consequently, the framework also falters in apprehending more intricate logical architectures such as multi-hop and induction, which inherently belong to the broader scope of first-order logic. This realization prompts us to emphasize the imperative of enhancing the logical expressiveness inherent to GNNs. Our goal is to empower them to encompass even the seemingly modest scope of $FOC_2$. A closer examination of the capabilities of MPGNNs reveals that $FOC_2$ is indeed more formidable than initially perceived.
>
> In terms of comparison with SOTA, we have added comparison with R-GAT as above. We have to point out that we didn't compare with a few SOTA models, such as [8] because they are not permutation-equivariant and therefore inconsistent for classification tasks.  These non-equivariant GNNs won't be equivalent to any logical classifier, and talking about their logical expressivenesses is meaningless. (See [6] for more detail).
>
> [1]Modeling relational data with graph convolutional networks ESWC2018
>
> [2]Relational graph attention networks arXiv
>
> [3]The logical expressiveness of graph neural networks ICLR2020
>
> [4]A Theory of Link Prediction via Relational Weisfeiler-Leman arXiv
>
> [5]Logical Expressiveness of Graph Neural Network for Knowledge Graph Reasoning arXiv
>
> [6]On the equivalence between temporal and static equivariant graph representations ICML2022
>
> [7]Explainablegnn-based models over knowledge graphs ICLR2021
>
> [8]The surprising power of graph neural networks with random node initialization IJCAI2021

---

> > ### Comment · Reviewer_auoR · 2023-08-21
> > **Thanks for the response**
> >
> > After reading the response and other reviews, I decided to keep my score and I'm still inclined to reject, as it does not address my concerns about the novelty and the significance of this work.
> >
> > **Technical novelty**. I'm a bit frustrated that the authors did not attempt to defend the technical novelty issues I raised in the initial comment, and my concerns that (1) *"the proposed method largely resembles the previous work of ACR-GNN"* and (2) *"the additional graph transformation F is more or less an incremental modification"* remain unaddressed. Still, I'm happy to be proven wrong in this regard.
> >
> > **R-GNN**. I appreciate the clarification and I strongly recommend authors revise the draft according to make clear that this is for a family of GNN instead of a particular example.
> >
> > **Limited significance**. Apart from the novelty issue, my biggest concern is the limited significance of this work, and the author's response seems to acknowledge it rather than rebut it:
> > - **The authors confirmed that R^2-GNN is limited to the node classification task only**. This significantly limits the scope of the work, as other tasks such as link prediction are critical to most of the graph-based systems and applications such as graph reasoning, entity resolution, QA, and recommendation.
> > - **The authors also acknowledge that real-world graph datasets are difficult to be modeled with FOL classifiers**. While I fully understand the difficulty here, this limitation inevitably undermines the empirical evaluation and consequentially the claims that are supposed to be supported. As far as I'm concerned, the aim of this work is to enable GNN learning FOC2 classifiers rather than pursuing SOTA performance, then if real-world datasets are inherently bad for showcasing this, then why would one want to include them in the first place? Showing results on real-world datasets that are not SOTA distracts the audience and does not add to the main claim of the work.
> > - **Finally, the authors acknowledge that even though FOC2 is already a fragment of FOL, R^2-GNN still cannot fully capture it**. I appreciate the authors' motivation and understand this is a challenging task, but the fact is that such a framework is indeed far from being practically useful and has yet to advance the explainability of GNN by a significant degree.

---

> > > ### Author Response · Authors · 2023-08-21
> > >
> > > We truly appreciate the time and effort you’ve invested in reviewing our paper.
> > >
> > > Briefly, you have concerns about the technical novelty and the significance .With the utmost respect, we respectfully hold a contrary viewpoint on these matters. In the ensuing discussion, we will provide detailed elucidations for each of these aspects.
> > >
> > > > Resemblance of our proposed method and the previous work of ACR-GNN
> > >
> > > We  want to re-emphasize that these two frameworks work in two totally different scenarios, where ACR-GNN focuses only on the simple **single-relational** scenario while our proposed method targets at the more complex **multi-relation** scenario.
> > > Besides, as you can see in the Section 3 and Section 4, the formulation and theoretical analysis of them are totally different, too.
> > >
> > > > Graph transformation is only an insignificant increment.
> > >
> > > The logical expressiveness of GNN over multi-relational graph is **unexplored**, so after large amounts of theoretical analysis in Section 3, we found that direct extension of ACR-GNN **fails** at capturing $FOC_2$ in multi-relational scenarios. Hence, in Section 4, we innovatively proposed the graph transformation strategy, which is the key strategy to surpass $FOC_2$ and break the expressiveness barrier in multi-relational graphs. In particular, we have provided the detailed theoretical analysis in Section 4 (together with the proof in the Appendix).  Experimental results also empirically confirm its superiority in both synthetic and real-world datasets.
> > >
> > > > The authors confirmed that R^2-GNN is limited to the node classification task only, which limits its significance.
> > >
> > > Node classification (as well as graph classification) is itself a quintessential task boasting numerous real-world applications. In line with prior researches that also only focus on node classification such as  [1], we embrace this task in the context of our paper.
> > >
> > > Tasks such as link prediction, as elaborated upon in our response (Answer 2),  may also be analysed and resolved using our technic, but it demands kind of different formations and proofs for conducting theoretical analyses. After all, other tasks are not focus of this work.
> > >
> > > > The authors also acknowledge that real-world graph datasets are difficult to be modelled with FOL classifiers. Therefore, real-world dataset experiments are unnecessary.
> > >
> > > We respectfully hold a differing perspective on this assertion. Our stance is rooted in the recognition that real-world datasets inherently embody noise and complexity, often lacking explicit logical rules. Consequently, the real-world datasets are difficult to be modelled with FOL classifiers. We firstly propose this point in rebuttal just in order to show that your original proposal of logically analysing real-world datasets is unrealistic.
> > >
> > > Yet, we think experiments on real-world datasets are still important. We want to reiterate the fundamental importance of incorporating both synthetic and real-world datasets in our experimental framework: Synthetic datasets, characterized by well-defined logical rules, serve to show the augmented logical expressiveness facilitated by our approach. On the other hand, the inclusion of real-world datasets demonstrates the practical performance enhancements attributed to this heightened expressiveness. Besides, the inclusion of both synthetic and real-world datasets with similar reason is a common practice in many previous related works, exemplified by [1], [2], and [3].
> > >
> > > > Finally, the authors acknowledge that even though $FOC_2$ is a fragment of FOL, R^2-GNN still cannot fully capture it. I think this work is not practical, and doesn’t advance the explainability of GNN by a significant degree.
> > >
> > > We wish to underscore that while R^2-GNN, in its original form, fails to capture $FOC_2$ within the multi-relation context, our contribution lies in proving that the integration of graph transformation empowers R^2-GNN to overcome this limitation, marking a pivotal advancement of GNN explainability.
> > >
> > > Besides, to showcase the practicality of our proposed method, we do experiments over the real-world datasets, which exactly **refutes** your aforementioned second limitation saying ‘real-world datasets don’t add to main claim of your work’. In fact, showing the practicality of a methodology certainly holds substantial importance and counts for a main claim.  As suggested by other reviewers, we have extended our experiments to encompass larger-scale real-world datasets. These additional results consistently corroborate the applicability, practicality and real-world relevance of our work.
> > >
> > > Thanks again for your time  in writing your comments, and about **R-GNN**, we will take your suggestion and update the description in our  revised  version.
> > >
> > > [1] The logical expressiveness of graph neural networks ICLR2020
> > >
> > > [2] Rethinking the Expressive Power of GNNs via Graph Biconnectivity ICLR2023
> > >
> > > [3] On the equivalence between temporal and static equivariant graph representations ICML2022

---

### Comment · Area_Chair_C2nq · 2023-08-13
**Please respond to the rebuttal**

Dear reviewers

Please read the rebuttals and respond, even if only briefly. I would specifically expect a detailed response from the more negative reviewers who are in the minority here.

Thanks

---

### Decision · Program_Chairs · 2023-09-21

**Decision:**

Accept (poster)

**Comment:**

In this submission, the authors explore the expressivity of GNNs in a multi-relational context. The paper is original and offers valuable theoretical insights that are sure to be interesting to the graph neural network community. The authors propose for the first time a rigorous logical foundation for relational GNNs.

While the authors acknowledge limitations in real-world dataset validation (and should make these limitations more visible and explicit in the final manuscript), the theoretical insights offered here outweigh this concern, rendering the paper valuable even without strong empirical results on real-world datasets. Given its contribution to the field and potential for sparking further research, I recommend the acceptance of this paper.